# ExoSloNano: multimodal nanogold labels for identification of macromolecules in live cells and cryo-electron tomograms

Lindsey N. Young [1,9], Alice Sherrard[2,9], Huabin Zhou[3], Farhaz Shaikh[1,7], Joshua Hutchings[1,8], Margot Riggi [4], Mythreyi Narasimhan[1], W. Alexander Flaherty[1], Eric J. Bennett [1], Michael K. Rosen [3,5], Antonio J. Giraldez [2] ✉ & Elizabeth Villa [1,6] ✉

In situ cryo-electron microscopy (cryo-EM) enables the direct interrogation of structure–function relationships by resolving macromolecular structures in their native cellular environment. Recent progress in sample preparation, imaging and data processing has enabled the identification and determination of large biomolecular complexes. However, the majority of proteins are of a size that still eludes identification in cellular cryo-EM data, and most proteins exist in low copy numbers. Therefore, novel tools are needed for cryo-EM to identify macromolecules across multiple size scales (from microns to nanometers). Here we introduce nanogold probes for detecting specific proteins using correlative light and electron microscopy, cryo-electron tomography (cryo-ET) and resin-embedded electron microscopy. These nanogold probes can be introduced into live cells, in a manner that preserves intact molecular networks and cell viability. We use this ExoSloNano system to identify both cytoplasmic and nuclear proteins by room-temperature electron microscopy, and resolve associated structures by cryo-ET. By providing high-efficiency protein labeling in live cells and molecular specificity within cryo-ET tomograms, ExoSloNano expands the proteome available to electron microscopy.

Cellular cryo-ET is a powerful method to study the structure and interactome of macromolecules directly in the cell. To access the cell interior at near-native conditions, cells are vitrified on grids and micromachined using focused ion beam (FIB) milling to generate 80–250-nm-thick lamellae. The lamella is imaged by transmission electron microscopy (TEM) at different tilt angles to produce a tilt series, which is then aligned. By back-projection one can then generate a three-dimensional (3D) tomogram at high spatial resolution that directly represents the structures of intact molecular networks. Recent reviews on the cryo-FIB-ET workflow include refs. 1–4.

Cellular cryo-ET allows structural biology and cell biology to be performed within the same experiment by leveraging preserved structural information in the context of their supramolecular organization in cells. So far, complex macromolecular assemblies such as nuclear pore complexes, proteasomes and ribosomes have been resolved in situ, their native environment, in various cell lines and conditions.

[1]School of Biological Sciences, University of California, San Diego, La Jolla, CA, USA. [2]Yale University, New Haven, CT, USA. [3]University of Texas Southwestern Medical Center, Dallas, TX, USA. [4]Max Planck Institute for Biochemistry, Munich, Germany. [5]Howard Hughes Medical Institute, Dallas, TX, USA. [6]Howard Hughes Medical Institute, La Jolla, CA, USA. [7]Present address: School of Medicine, University of California, San Francisco, San Francisco, CA, USA. [8]Present address: Chan Zuckerberg Imaging Institute, Redwood City, CA, USA. [9]These authors contributed equally: Lindsey N. Young, Alice Sherrard. ✉e-mail: antonio.giraldez@yale.edu; evilla@ucsd.edu

These assemblies represent a small class of macromolecular complexes that are large enough to be unambiguously identified in tomograms. However, approximately 10,000 different proteins were identified by mass spectrometry to be present in cells at any given time[5]. Specific macromolecular complexes of interest are possible to locate and identify using molecular labels, fluorescence microscopy and image recognition algorithms. But no method is applicable to image a broad range of proteins embedded in the molecular networks in which they operate. Here, we present a new method to Exogenously deliver probes via pore forming Streptolysin O (Slo) using functionalized Nanoparticles (ExoSloNano) to expand the proteome that is accessible to cryo-ET.

One of the major benefits of cryo-ET, that the sample is vitrified and directly visualized without stain[6], prevents introduction of labels through standard fixation and permeabilization approaches, and is incompatible with powerful oxidative probes such as APEX and APEX2, miniSOG and chromatin electron tomography (ChromEMT)[7–9]. Antibodies have been a common choice for immunolabeling in TEM, but they cannot be introduced without fixation and permeabilization, are large and have limited nuclear delivery. Moreover, a growing consensus argues that poorly vetted commercial antibodies contribute to the reproducibility crisis in basic science[10,11]. Currently available labeling strategies in cellular cryo-EM utilize large macromolecules ranging from 20 nm to 70 nm. While such labels are beneficial for visual identification, they may disrupt complex molecular networks within the crowded cellular environment, or affect their formation[12–14].

Wide-field cryo-fluorescence and super-resolution microscopy can be used for correlated target identification of cells or regions of cells on an electron microscopy (EM) grid or molecules within a cryo-tomogram[15]. A major challenge with wide-field cryo-correlative light and electron microscopy (cryo-CLEM) is that the resulting fluorescence signal can be of a size in the same order of magnitude of the field of view of a tomogram (~300-nm diffraction limited data with a 500-nm tomogram's field of view). To date, cryo-CLEM is mostly used to locate regions within the cell for FIB milling or tilt-series acquisition, but not to resolve discrete macromolecules[16]. Exciting development in these areas will continue to provide higher precision and accuracy in localization of macromolecules[15,17]. However, we expect that a purely fluorescence-based tagging system to correlate single molecules with cryo-ET data will have limitations and will benefit substantially from labels that can be directly visualized in cryo-ET data to find and label molecules ranging from cells to molecules (that is, tens of microns to the angstrom scale, five orders of magnitude).

It would be ideal to complement experimental methods to identify novel targets with the growing computational and artificial intelligence programs for particle identification. Template-matching programs include two-dimensional (2D) template matching[18], a computationally exhaustive search of high-resolution information from a 3D reference to match to the high-frequency information acquired on a close-to-focus 2D micrograph. A similar method has been developed for 3D, although acquired at standard defocus values for cellular cryo-ET[19]. As of yet, the matching of high-resolution signatures (2D template matching) is still restricted to molecules of sufficient molecular weight[18,20].

The potential of machine learning for image pattern recognition is undoubtable. However, currently, most machine learning algorithms act as robust particle pickers—identifying molecules that could be identified by eye, but in an automated way. Machine learning programs for particle identification include crYOLO, TomoTwin and DeepFinder[21–23]. The added challenge is insufficient priors or ground-truth knowledge with which to evaluate template matching and machine learning outcomes for novel targets. Template matching, machine learning and artificial intelligence are image classifiers, and the low signal-to-noise ratio of cryo-ET limits their application to the entire proteome. Finally, there are homologous macromolecules inside the cell that possess high structural similarity to each other. It would be beneficial to have tools to differentiate between these in cryo-ET data.

Highly homologous structures often have small differences that are difficult to discern, even when considering high-resolution features as with template-matching procedures[18]. For instance, a core nucleosome and nucleosomes containing histone variants have a backbone root-mean-square deviation < 1 Å, yet their incorporation elicits vastly different functions within the genome (maintenance of active-versus-inactive chromatic regions, transcriptional activation, centromeric regions, and so on).

When imaging vitrified biological material by TEM, the pixel intensity is roughly proportional to the mass of the object[24]. Biological samples contain macromolecules composed of elements of low and similar atomic number (carbon, oxygen, nitrogen, phosphorus), which leads to low contrast in imaging. In contrast, gold nanoparticles (Au NPs) provide useful high-contrast markers because of gold's high atomic number and the condensed structure[25,26]. Immunogold labels from colloidal gold have been used extensively in resin-embedded TEM[27]. However, colloidal gold is large and polydisperse, and relies on nonspecific interactions for binding[28]. In contrast, Au NPs are monodisperse, and nanogold labeling of macromolecules has been demonstrated in biochemical systems[29].

Recent work using nanogold in cryo-conditions includes external nanogold on the cell surface[30,31], endocytosis of 2.2-nm Au NPs into internalized membrane compartments[32,33] and in postsynaptic neurons[34]. Similarly, nanogold-mediated labeling of macromolecules in live cells required endocytosis of the Au NPs[26,33,35], which may not afford access to all subcellular compartments. However, fully generalizable nanogold labeling for cytosolic and nuclear targets has not yet been demonstrated, and would be a powerful tool for the cellular EM field.

To visualize endogenous proteins, we developed a generalizable EM labeling strategy, ExoSloNano. Once inside the cell, these functionalized nanogold particles become covalently conjugated to a target protein that is endogenously tagged, all within live cells. We use small Au NPs, 1.4 nm and 5 nm, with an attached HaloLigand to provide molecular specificity to the protein of interest, which carries an endogenous HaloTag. In the presence of the monovalent nanogold-HaloLigand, the modified bacterial haloalkane dehalogenase, HaloTag, forms a covalent, irreversible linkage to its conjugate HaloLigand[36], yielding a covalently attached nanogold particle directly next to a target of interest (Fig. 1a). HaloTag is relatively small (33-kDa molecular weight) and has been used extensively as a genetically encoded molecular tag[36]. For visualization by live-cell fluorescence microscopy and CLEM methods, the 1.4-nm-HaloLigand-nanogold probes were also conjugated with Alexa fluorophores (Alexa Fluor 488 or 594), yielding 1.4-nm-Halo-Alexa-nanogold-488 (1.4-nm-HAN-488) or 1.4-nm-Halo-Alexa-nanogold-594 (1.4-nm-HAN-594). Thus, these small probe designs possess (1) molecular specificity, (2) a high-contrast nanogold moiety and (3) are suitable labels for imaging from microns to nanometers (Fig. 1b).

For these Au NPs to be useful molecular labels in live cells, we sought to demonstrate that the probes can be robustly delivered into live cells, detected by both fluorescence and EM, recognize their targets of interest, and that the method is compatible with subtomogram analysis. Our method is applicable to both room-temperature TEM and cryo-EM, leveraging the scale and statistical power of the former with the high resolution of the latter. This method should be broadly useful and was developed with challenging targets in mind, macromolecules that are <200 kDa, have high structure similarity to other targets and have a specific suborganellar localization.

## Results

### Live cells recover from bacterial toxin treatment

Exogenous entry of nanogold particles into live cells was achieved by using the bacterial endotoxin streptolysin O (SLO), which forms small pores in the plasma membrane by binding to cholesterol[37]. Approximately 35–50 subunits oligomerize to form a pore ranging from 15 nm to

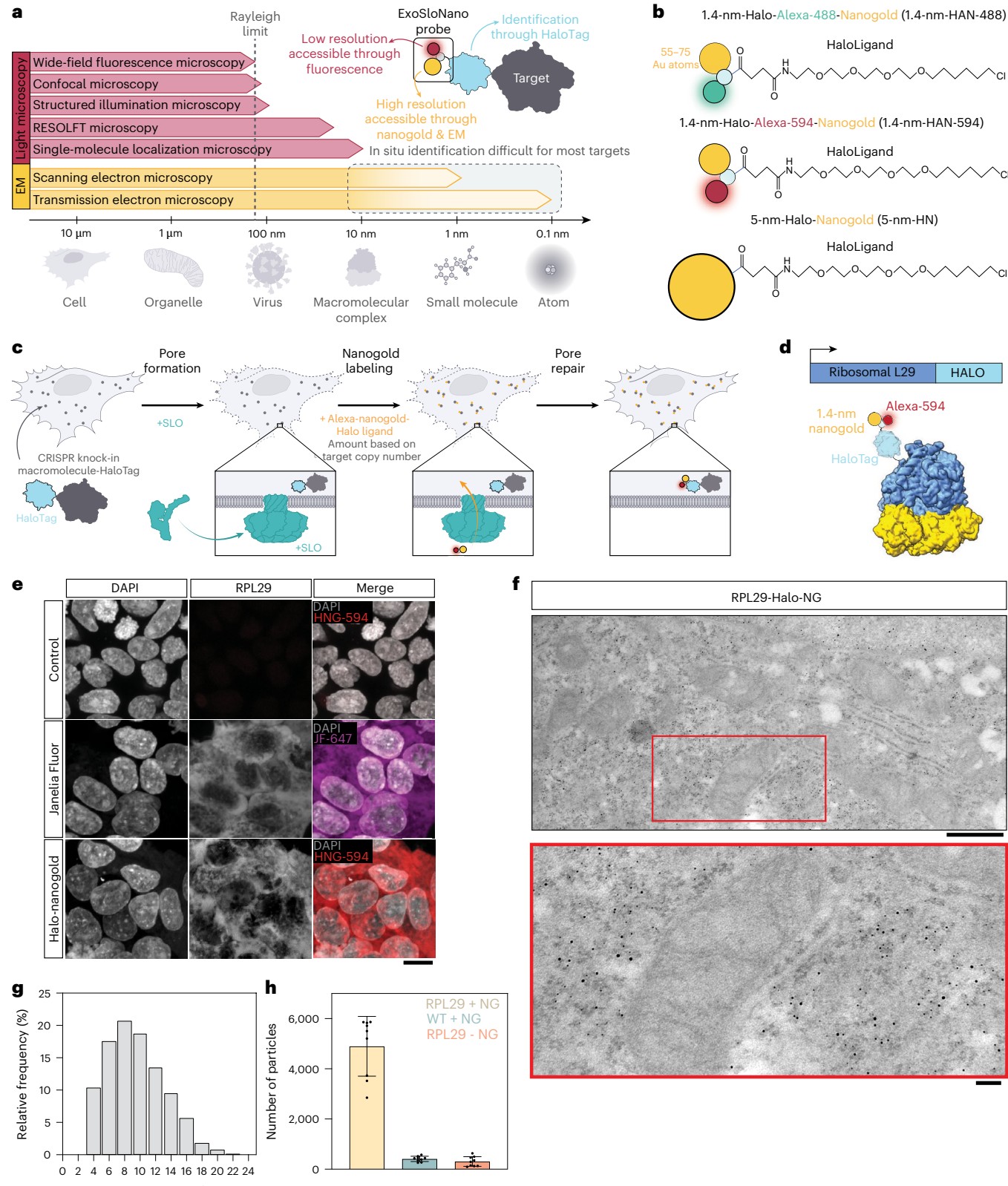

**Fig. 1 | Nanogold probes and room-temperature TEM. a,** There is a need for tools that span both the mesoscale (tissue and cellular localization afforded through fluorescence microscopy) and nanoscale (cellular and subcellular localization afforded by EM) to enable target localization from microns to nanometers. **b,** Nanogold probes used in this study. 1.4-nm-HAN-488, 1.4-nm-HAN-594 and 5-nm-HN. **c,** Schematic of ExoSloNano: live-cell delivery of exogenous nanoparticles by SLO. **d,** Schematic of a covalent linkage of 1.4-nm-HAN-594 labeling of ribosomal protein L29-Halo. **e,** Live-cell delivery of 1.4-nm-HAN-594 into RPL29-Halo cells: WT HEK 293T cells with nanogold delivery (top). RPL29-Halo cells treated with cell-permeant Janelia fluorophore (middle). Delivery of 1.4-nm-HAN-594 into RPL29-Halo cells (bottom). **f,** TEM micrograph of RPL29-Halo cells with gold enhancement of 1.4-nm-HAN-594. **g,** Histogram of the size distribution of gold particles after gold enhancement. $N = 956$ particles across two cells. **h,** Plot of the number of nanogold particles found in the indicated conditions. Data shows the mean ± s.d. $N = 9$ cells. Scale bars, 10 μm (**e**); 600 nm (top), 150 nm (bottom) (**f**).

30 nm in diameter[38,39] (Fig. 1c). As a molecular tool, these bacterial pores have been used for exogenous delivery of membrane-impermeant proteins into the cell[40,41], as well as probes for super-resolution microscopy studies[42]. Following pore formation, the plasma membrane repairs through an ESCRT-mediated mechanism[37,43]. SLO treatment and recovery is not cell-type specific and has been shown in monocytes (THP-1)[40], COS-7 cells[40], Chinese hamster ovary (CHO) cells[42], National Institutes of Health (NIH) 3T3 fibroblast cells[44], human osteosarcoma (U2OS) cells[42] and Henrietta Lacks (HeLa) ovarian cells[42,44]. We monitored SLO-treated cells up to 100 h after treatment to confirm that cells recover (Extended Data Fig. 2b). We observed a slight reduction in cell growth rate in RPL29-Halo cells compared to wild-type (WT) HEK cells, with SLO treatment having no effect on cell growth, and detected only a mild reduction by subsequent addition of nanogold (Extended Data Fig. 2a–c). These changes are minor, and cells ultimately recover and reach maximum confluency. Moreover, we found low levels of apoptosis that are comparable to untreated cells (Extended Data Fig. 2d). Finally, we found by room-temperature EM that the cell's ultrastructure is maintained after SLO treatment (Extended Data Fig. 2e). Other toxin-mediated delivery includes cargo delivery of nanobodies into the cytosol of primary rat cortical neurons using a chimeric toxin-based platform based on botulinum neurotoxins[45]. To determine if SLO could be used for more sensitive cell types, iNeuron cells were cultured and treated with an SLO and anti-MAP2-Alexa-647. Cells recovered following treatment and showed fluorescence labeling along the neuronal dendrites from anti-MAP2-Alexa-647 (Extended Data Fig. 10). Together, these results suggest that SLO treatment provides a broadly applicable method for probe delivery to live cells.

### Delivery of 1.4-nm-HAN-594 and enhancement with room-temperature TEM

To test the ability of our probe to label specific proteins, we first targeted the ribosome, a large and abundant macromolecular machine that has been a well-characterized and beloved target for cellular cryo-ET studies[18,46,47]. To ensure that all ribosomes contained a HaloTag, we used a HEK 293T knock-in cell line with an endogenous C-terminal HaloTag7 on ribosomal protein L29 (RPL29)[48]. This RPL29-Halo cell line had previously been used as a marker to monitor ribosome turnover and degradation, as RPL29 is stably integrated, guaranteeing that each ribosome carries a HaloTag[48]. We utilized this cell line to validate that our nanogold probes successfully labeled the intended target using room-temperature TEM and cryo-ET.

We first demonstrated nanogold labeling using resin-embedded room-temperature TEM. To this end, we delivered Halo-Alexa-594-1.4-nm nanogold (1.4-nm-HAN-594) to HEK 293T L29-HaloTag cells (RPL29-Halo; Fig. 1d). WT cells treated with SLO and 1.4-nm-HAN-594 do not show fluorescence (Fig. 1e). In contrast, RLP29-Halo cells treated with SLO and 1.4-nm-HAN-594 show fluorescence within the cytosol (Fig. 1e) that is comparable in intensity and spatial distribution to the fluorescence of a commonly used cell-permeable Janelia fluorophore that contains a HaloLigand (Fig. 1e). This shows that nanogold labeling is specific to the presence of the HaloTag, and suggests that live nanogold tagging may not perturb protein localization. Following nanogold delivery, cells were fixed, exposed to silver, which binds to and enhances 1.4-nm-HAN-594, which subsequently increases in diameter (Fig. 1f). Visualized by room-temperature EM, the enlarged 1.4-nm-HAN-488 have a mean diameter of 9.6 nm, and show a broad, uniform distribution of nanogold throughout the cytoplasm and endoplasmic reticulum (Fig. 1f). Nanogold enhancement was consistent with the presence of RPL29-Halo, yielding robust cellular recruitment in the presence of the Halo-tagged target; $5 \times 10^4$ particles were detected in a 10.1-um³ volume (Fig. 1g and Extended Data Fig. 3a). When 1.4-nm-HAN-594 was supplied in cells without a Halo-tagged protein (WT), nanogold enhancement and detection was negligible (Fig. 1h). Similarly, nanogold

enhancement and detection was minimal in RLP29-Halo cells when 1.4-nm-HAN-594 was not supplied (Fig. 1h and Extended Data Fig. 3a). Using an established protocol to determine cellular copy number[49], we determined that there are approximately $1.7 \times 10^6$ ribosomes per HEK RPL29-Halo cell (Extended Data Fig. 3e). Extrapolating the number of enhanced nanogold particles detected from a 10.1-μm³ volume to the entire volume of a HEK L29-Halo (5282.5 μm³) cell yields $2.6 \times 10^6$ ribosomes per cell (Extended Data Fig. 3b–d). Overall, this is consistent with the saturation of L29-Halo binding sites due to the delivery and enhancement of the 1.4-nm-HAN-594 probes (see next section for a quantitative analysis of labeling efficiency).

### Detection of 5-nm-HN by cryo-ET

We next sought to visualize individual ribosomes with bound nanogold by cryo-ET. The 5-nm-Halo-nanogold (5-nm-HN) particles are readily visible in cryo-ET data, and were utilized to optimize parameters for data collection, tilt-series alignment and subtomogram analysis of in situ tomograms by cryo-ET. The 5-nm-HN particles were delivered to live RPL29-Halo cells as described above. Cells recovered overnight, and then were seeded onto EM grids, vitrified by plunge freezing, subjected to FIB milling and and tilt series were collected (Table 1). Individual 5-nm-HN particles could be readily identified directly adjacent to ribosomes (Fig. 2c,d). Automated particle-picking of 13,748 ribosomes across 46 tomograms and subsequent subtomogram alignment and averaging yielded a bright mass (Fig. 2c) with a size consistent with a 5-nm nanogold particle. Unsurprisingly, this indicated that the high-intensity region in the subtomogram corresponding to the high-mass density of the nanogold particle drives the alignment of the nanogold-labeled ribosomes.

To determine labeling efficiencies, the pairwise distances between these 13,748 particles were determined between nanogold coordinates and the center of the refined ribosome coordinates. The nanogold coordinates were obtained based on the initial alignment including the nanogold signal, whereas the ribosome coordinates were obtained from the latter alignment without it. The distribution of distances yielded a Gaussian distribution in which the average distance is 16.8 nm (±3 nm) between the center of mass of the ribosome and the center of mass of the 5-nm nanogold. Then, 12.9 nm, the distance from the center of a ribosome to the last ordered residue of L29, was subtracted, and those values were plotted. Of the 13,748 picks, 9,567 particles, or approximately 69.5%, were determined to be within a distance of 5.1 nm (±1.9 nm; Fig. 2e,f). These distances agree with the manual identification of the 5-nm-HN distance as well as the position relative to the Halo-tagged ribosomal protein (Extended Data Fig. 4a). The labeling efficiency was calculated using the labeling efficiency formula used in single-molecule experiments[50]—the number of labeled particles divided by the combined number of labeled and unlabeled particles. Applying this to the number of nanogold-labeled ribosomes divided by the number of nanogold-labeled ribosomes and unlabeled ribosomes yields a labeling efficiency of 0.695 (69.5%; Fig. 2f). It is important to note that labeling efficiency is a convolution of several independent factors, which include efficiency of SLO pore formation, probe delivery including access to the target and binding affinity of for example, HaloTag-HaloLigand, ability to saturate all targets and rate of new protein synthesis after probe delivery. A further measure of labeling efficiency was determined by purification of ribosomes from SLO-treated cells and delivery of nanogold, as presented below.

### Nanogold signal randomization for subtomogram analysis

To recover the ribosomal signal and mitigate the effects of the high-intensity pixels preventing ribosomal alignment during subtomogram analysis, the nanogold signal was removed by randomizing the values of the pixel corresponding to the nanogold on a per-particle basis using a custom MATLAB script (Extended Data Fig. 5). Then, those nanogold-randomized subtomograms were aligned in

**Table 1 | In situ cryo-FIB-ET data collection and reconstruction statistics**

| | Labeling the ribosome | | | | | Labeling H2AFY |
|---|---|---|---|---|---|---|
| | In situ labeled 5-nm-HN | | In situ labeled 1.4-nm-HAN-488 | | In situ unlabeled | In situ labeled 1.4-nm-594 |
| **Data collection** | | | | | | |
| Magnification | ×81,000 | ×81,000 | ×81,000 | ×81,000 | ×81,000 | ×105,000 |
| Voltage (kV) | 300 | 300 | 300 | 300 | 300 | 300 |
| Defocus range (µm) | −3 to −5 | −3 to −5 | −3 to −5 | −3 to −5 | −3 to −5 | −3 to −5 |
| Pixel size (Å) | 1.068 | 1.068 | 1.068 | 1.068 | 1.068 | 0.876 |
| High Dose 0° | 20 | 0 | 20 | 0 | 20 | 0 |
| Total Electron exposure (e–/Å$^2$) | 180 | 170 | 180 | 160 | 180 | 170 |
| Tilt-range/step (°) | ±36 /2 | ±48/2 | ±36/2 | ±48/2 | ±36/2 | ±40/2 |
| Tilt-scheme | Dose-symmetric, grouping 2 | | | | | |
| **Processing** | | | | | | |
| Symmetry imposed | C1 | | C1 | | C1 | C1 |
| Initial particle images (no.) | 13,748 | | 77,336 | | 17,662 | 6,603 |
| Final particle images (no.) | 1,743 | | 10,759 | | 2,7689 | 4,955 |
| Map resolution (Å) | 15 | | 11 | | 11.2 | 18Å |
| Fourier shell correlation threshold | 0.143 | | | | | |

RELION-3.1.4 yielding a ribosome-centered subtomogram average (Extended Data Fig. 5). Particles were then re-extracted and a soft low-pass-filtered mask from the ribosome average was used during further subtomogram analysis.

### Detection of 1.4-nm-HAN-488 by cryo-ET

To test a smaller probe, optimization of data collection conditions and preprocessing steps using the 5-nm-HN was applied to directly visualize ribosomes labeled with 1.4-nm-HAN-488 by cryo-ET. Briefly, 1.4-nm-HAN-488 was delivered to RPL29-Halo cells via SLO, and 18 h later, cells were seeded onto EM grids. Samples were then vitrified after adhesion for 4–6 h, subjected to FIB milling, and tilt series were collected (Fig. 3a and Table 1). Data were collected at sufficient magnification (1.068 Å per pixel) to ensure that there would be enough pixels to represent an individual nanogold moiety (14 Å, ~13 pixels). As with 5-nm-HN, 1.4-nm-HAN-488 could be visualized directly adjacent to the ribosome within a tomogram (Fig. 3e).

### Whole-tomogram-level detection of 1.4-nm-HAN-488 probe from in situ tomograms

To demonstrate that this method will allow the detection of particles within tomograms, we located gold particles independent of the macromolecules they may be bound to. For this, we detected high-intensity pixels corresponding to the 1.4-nm-HAN-488 particles directly at the whole-tomogram level (Extended Data Fig. 6) by filtering for coordinates corresponding to voxels consistent with the volume of a 1.4-nm-HAN-488-nanogold particle (Methods). Then, we calculated the distance to the nearest ribosome (Methods); the average pairwise distance between the centers of the nanogold and the labeled ribosomal subunit was 4.5 nm (±2.4 nm; Fig. 3g). Across 14 tomograms, the ribosomal labeling efficiency was determined to be 0.87 (87%; Fig. 3h). Of the particles that were consistent with the dimensions of the 1.4-nm-HAN-488-nanogold, 15% were determined to be unbound (Extended Data Fig. 6j). Notably, these data suggest that nanogold particles of multiple sizes can be used within in situ cryo-ET experiments to facilitate future multiplexing, that is, the tagging of multiple cellular components simultaneously. To demonstrate that these particles can be distinguished and thereby enable multiplexing, we deposited nanogold particles on EM grids and quantified their diameter. This showed clearly separated size distributions with a mean diameter of 1.52 nm (±0.359 nm) and 5.62 nm (±0.911 nm) for 1.4 nm and 5-nm-HN, respectively (Extended Data Fig. 1a–d).

### Labeling efficiency determined by isolation of 1.4-nm-HAN-488-labeled ribosomes from SLO-treated L29-Halo cells

As mentioned above, labeling efficiency is a combination of several factors, efficacy of SLO pore formation, probe delivery including access to the target and binding affinity of, for example, HaloTag-HaloLigand, ability to saturate all targets and rate of new protein synthesis after probe delivery. To further validate in-cell labeling efficiency of the exogenously delivered 1.4-nm-HAN-488 probe to target the ribosomal L29-HaloTag, ribosomes were purified from HEK 293T L29-Halo cells treated with SLO and labeled with 1.4-nm-HAN-488 (Methods). The purified ribosomes were visualized by in vitro cryo-ET. The labeled and unlabeled populations were separated into two classes during 3D classification, and resulted in a labeling efficiency of 0.92 (92%; Extended Data Fig. 7).

### Expanding the proteome to challenging targets

We developed this method generally to expand the proteome available to cellular EM. However, our ultimate goal was to tackle targets that have proven challenging to available methods. Notably, the nucleus has largely remained an uncharted territory for in situ structural cell biology. Even for ultrastructure studies using room-temperature EM, the nucleus has been referred to as 'the dark side of the cell', largely due to the challenge with antibodies penetrating into nucleus[51]. Great progress has been made in imaging chromatin by room-temperature EM[9] and cryo-ET[52], but it has so far not been possible to discern histone variants and nucleosome subpopulations, that is, we currently cannot distinguish heterochromatin versus euchromatin beyond inferring it from the degree of compaction. To our knowledge, no exogenous, for instance antibodies, or endogenous, for example small genetically encoded proteins visible in cryo-EM, methods have been able to specifically target and label histone-specific nucleosomes with minimal disruptions to the chromatin landscape.

Understanding how chromatin architecture regulates gene expression is critical to understand how cell lineages are specified in development and disease. Powerful, but indirect, methods such as chromosome conformation capture assays, including Hi-C[53], map DNA interactions at a range of scales (0.1–1 Mb), from an individual

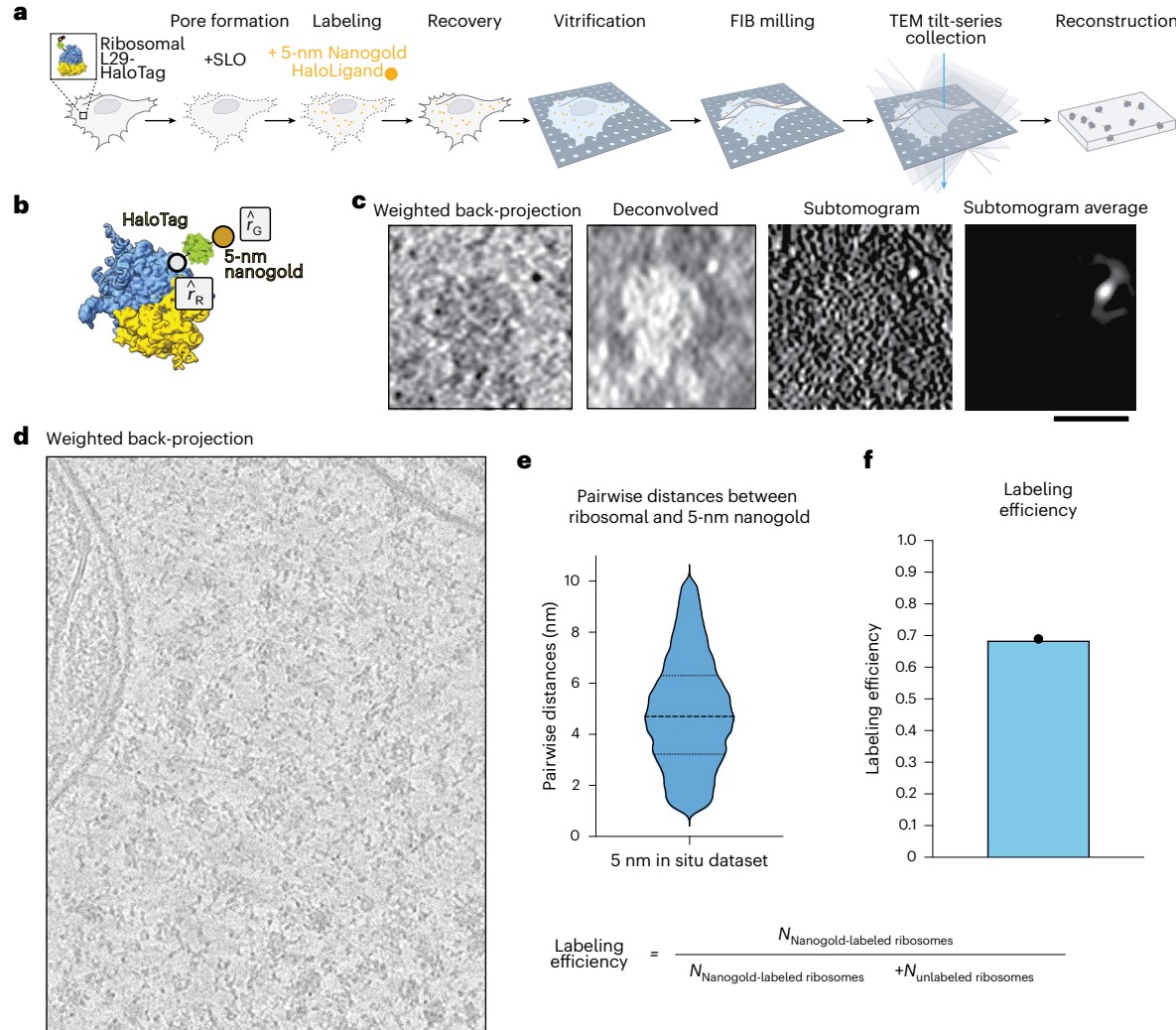

**Fig. 2 | Cellular cryo-ET of 5-nm-HN-labeled ribosome. a,** Schematic of live-cell delivery of 5-nm-HN, vitrification, cryo-FIB milling, cryo-ET and tomographic reconstruction to generate 3D volumes. **b,** Schematic of a 5-nm-NH-labeled ribosome. The white dot represents the C terminus of L29 that defines $\hat{r}_R$, and the center of the nanogold defines $\hat{r}_G$. **c,** Central slice (8 Å per pixel) of an individual ribosome labeled with 5-nm-NH by weighted back-projection (contrast is as in original data, where dark pixels correspond to high occupancy, that is, black-on-white), the same subtomogram deconvolved, its corresponding normalized subtomogram and the subtomogram average from 13,748 ribosomal picks, dominated by alignment of nanogold intensity (contrast inverted, that is, white-on-black). Experiments were repeated twice independently with reproducible results. **d,** Central slice of a weighted back-projection of the whole tomogram, averaged over ten slices. **e,** Violin plot of the pairwise distances between ribosomal subunit L29 ($r_R$) and 5 nm-HN ($r_G$), mean = 5.1 nm, s.d. = 1.9. $N$ = 9,567 pairwise distances. **f,** Bar graph of the labeling efficiency (0.695) and how the labeling efficiency was calculated. Scale bars, 25 nm (**b**); 25 nm (**c**); 20 nm (**d**).

cell to a population of cells[54]. A major limitation is that chromatin capture experiments only capture genomic DNA interactions; that is, nuclear macromolecular complexes are lost. To complement chromatin capture experiments, in situ structural biology has the potential to map spatial relationships within native chromatin landscapes.

Genome organization relies on the incorporation of canonical and noncanonical nucleosomes and their associated epigenetic markers, among other factors[55]. Canonical nucleosomes are octamers containing two copies of replication-coupled histone proteins H2A, H2B, H3.1 and H4, wrapped by ~145–147 base pairs of DNA[56]. Variants of H2A, H2B and H3 allow greater complexity of epigenomic regulation, as their incorporation is specific to cell type, developmental and functional state, among other aspects[57]. In mammals, there are 19 variants of H2A alone[58]. Histone variants play important roles in genome maintenance—for example, the phosphorylation of H2A.X labels sites of double-stranded DNA breaks for DNA repair processes[59], H3 variant CENP-A localizes to centromeric DNA[60], and H2A.Z is incorporated

around transcription start sites[61,62]. Nucleosomes from different species and nucleosomes containing histone variants have high structural homology[63], with <1 Å root-mean-square deviation when the backbone of the structures are compared. The largest variation in their structure is due to the 'breathing' of the linker DNA near to the entry and exit of the nucleosome[64]. Thus, even with the advent and progress of template-matching methods for cryo-ET data, it will be difficult to discern nucleosome subpopulations within the nucleus. As a test for our method, we chose macroH2A (mH2A, gene name *H2AFY*), a histone with broad spatial distribution across the nucleus (Fig. 4b), but enriched at heterochromatin regions[65], lamin-associated domains[66] and, most notably, the inactive X chromosome[67].

Although mH2A plays an important role in chromatin architecture and maintenance, we currently do not have a way to identify and spatially resolve mH2A-containing nucleosomes within the nucleus. We reasoned that 1.4-nm-HAN-594 and cryo-ET would be ideal for probing histone variants, as the small probe elicits minimal perturbations.

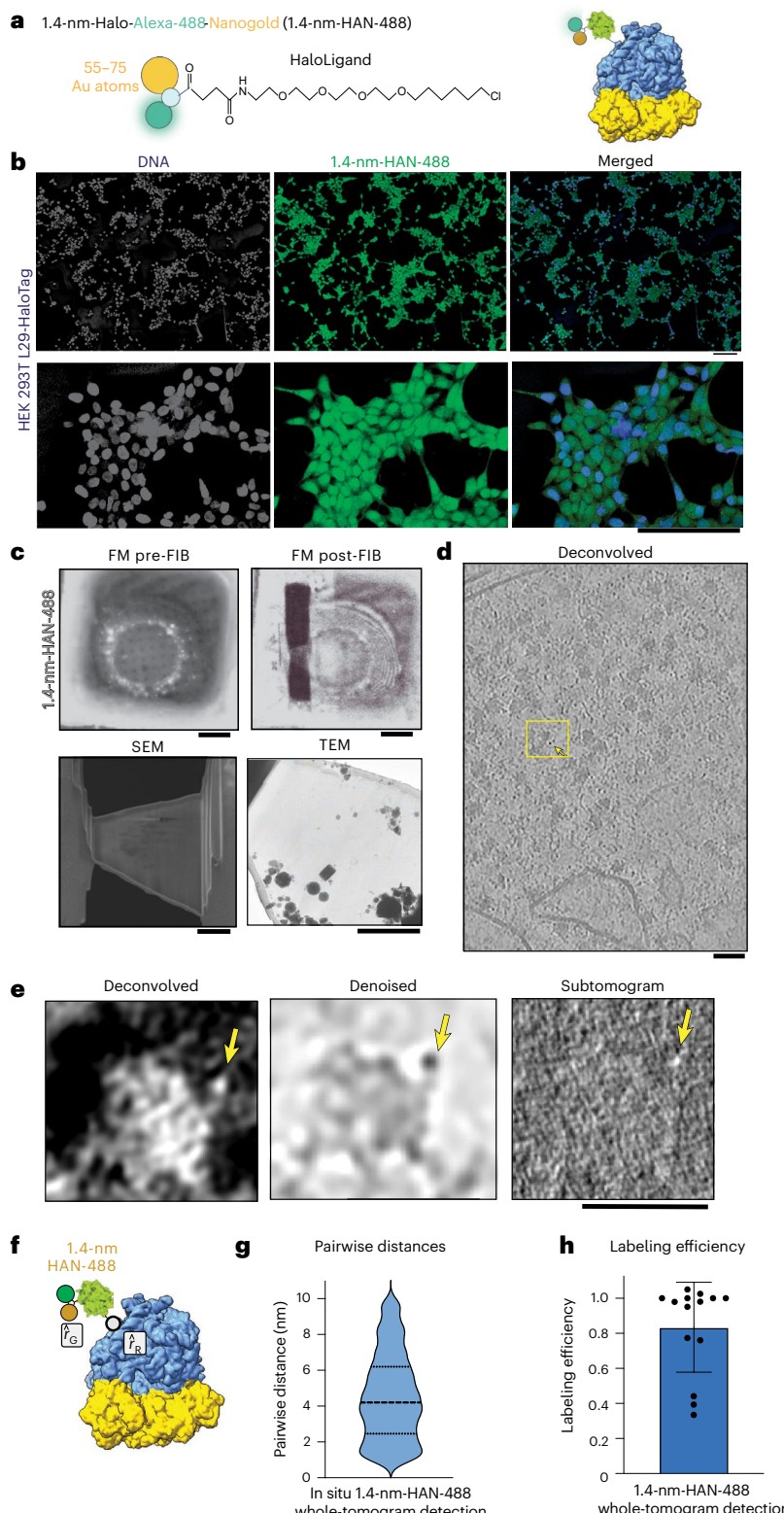

**Fig. 3 | Ribosomal labeling with 1.4-nm-HAN-488. a**, Schematic of probe used to label the ribosome. **b**, HEK 293T RPL29-Halo cells treated with SLO and labeled with 1.4-nm-HAN-488. Left, DNA (DAPI stain); middle, 1.4-nm-HAN-488 (green); right, merged, DNA (blue), 1.4 nm-HAN-488 (green). **c**, Fluorescence microscopy (FM) of a grid before FIB and after FIB milling (scale bar, 10 μm). Scanning electron microscopy (SEM) view of the lamella (scale bar, 1 μm) and TEM view of the lamella. **d**, Central slice of a deconvolved tomogram (contrast is black-on-white). **e**, Central slice of an individual ribosome labeled with 1.4-nm-HAN-488 shown as a deconvolved, denoised and a normalized subtomogram (4 Å per pixel). **c**–**e**, Experiments were repeated three times independently with

reproducible results. Scale bar, 25 nm. **f**, Schematic of 1.4-nm-HAN-488-labeled ribosome to represent the distance from ribosomal subunit L29 ($\hat{r}_R$) and 1.4-nm-HAN-488 ($\hat{r}_G$). **g**, Violin plot of the pairwise distances between ribosome coordinates and 1.4-nm nanogold coordinates. Mean, 4.5 nm; s.d., = 2.4 nm, from 817 1.4-nm-HAN-488-labeled ribosomes across 14 tomograms. **h**, Labeling efficiency of 1.4-nm-HAN-488-labeled ribosomes from whole-tomogram-level detection from in situ tomograms. Mean, 0.87; s.d. = 0.28; $N$ = 14 tomograms. Scale bars, 100 μm (top), 50 μm (bottom) (**b**); (clockwise) 10 μm, 10 μm, 1 μm, 1 μm (**c**); 25 nm (**d**); 25 nm (**e**).

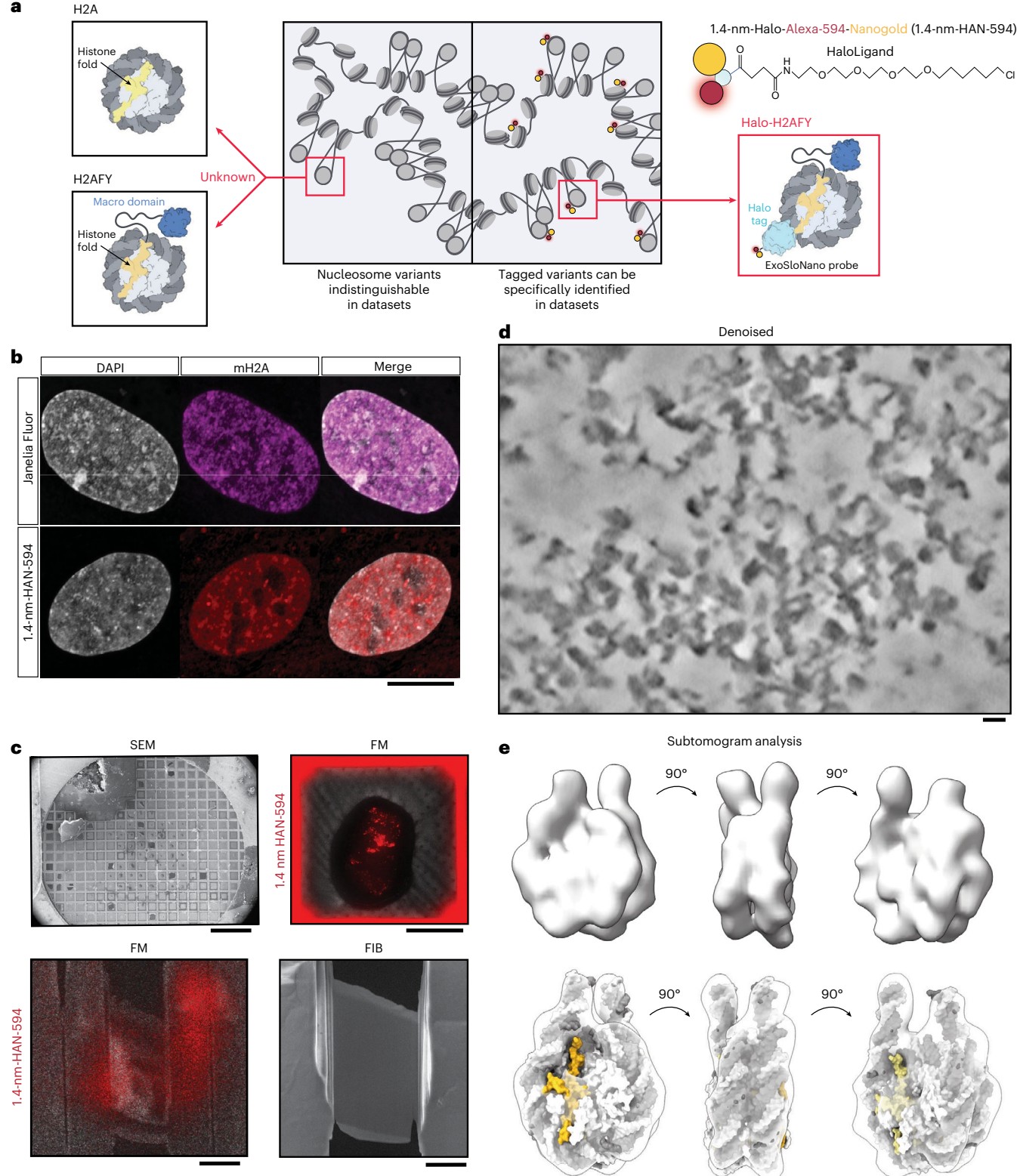

**Fig. 4 | Nuclear delivery of 1.4-nm-HAN-594 to target heterochromatin.**
**a**, Protein domain schematic of relevant histone variants. Within chromatin organization, there are canonical nucleosomes and histone variants. Distinguishing nucleosome histone variants within the nucleus would facilitate an understanding of chromatin organization. The ExoSloNano probe will recognize the HaloTag on histone variant H2AFY/mH2A. **b**, Top, RPE1 Halo-mH2A cells labeled with membrane-permeant Janelia Fluor, which freely diffuses into cells and recognizes the HaloTag. Bottom, RPE1 Halo-mH2A cells labeled with 1.4-nm-HAN-594 delivered via SLO. Experiment was repeated three independent times with reproducible results. **c**, Cryo-FIB-ET workflow: RPE1 Halo-mH2A cells were grown on an EM grid, subjected to cryo-fluorescence imaging before FIB milling, during milling (before final polishing) and after milling. **d**, Denoised and deconvolved in situ tomogram of chromatin. Experiment was repeated three independent times with reproducible results. **e**, Subtomogram analysis of all nucleosomes: opaque rendering (above) and with docked nucleosome structure (below; Protein Data Bank, PDB 2CV5). Scale bars, 10 µm (**b**); 100 µm, 30 µm, 10 µm, 5 µm (**c**); 10 nm (**d**).

To obtain endogenous levels of Halo-tagged mH2A, cultured human retinal pigment epithelium (RPE1) cells, a primary female derived cell line, were engineered using CRISPR to add an N-terminal HaloTag and a V5 Tag epitope at the endogenous *H2AFY* locus (Fig. 4a and Extended Data Fig. 8c). Endogenous integration of the HaloTag was verified by sequencing (Extended Data Fig. 8c).

## Nanogold labeling is restricted to the nucleus in RPE1 Halo-mH2A cells

To investigate 1.4-nm-HAN-594 selectivity inside the cell, WT RPE1 and RPE1 Halo-mH2A cells were treated with SLO for labeling with 1.4-nm-HAN-594 and visualized by fluorescence microscopy (Extended Data Fig. 8e). Fluorescence imaging shows that 1.4-nm-HAN-594 is restricted to the nucleus in RPE1 Halo-mH2A cells, while it is dispersed throughout the nucleus and cytoplasm in WT RPE1 cells (Extended Data Fig. 8e), indicating that specificity is achieved through the HaloTag on the target.

## Quantification of an endogenous target and nanogold delivery

To match the internal cellular concentration of the target and mitigate unlabeled targets and unbound nanogold, appropriate exogenous nanogold concentrations must be delivered to the cell. This requires knowledge of the target's cellular copy number and quantification of probe delivery. We used the earlier-mentioned established protocol to determine the copy number of mH2A in RPE1 cells[49]. From this, we determined that there are approximately $7 \times 10^5$ copies ($\pm 2.3 \times 10^5$) of mH2A within the nucleus (Extended Data Fig. 8f). To quantify nanogold delivery into the cell, RPE1 mH2A-Halo cells were exposed to SLO and 1.4-nm-HAN-594, and the fluorescence intensity per cell was measured against a dilution series of the 1.4-nm-HAN-594 probe. The internalized concentration of 1.4-nm-HAN-594 was determined to be $6.6 \times 10^5$ copies ($\pm 1.05 \times 10^5$) per cell (Extended Data Fig. 8g).

## Nanogold labeling does not perturb chromatin architecture

Cultured RPE1 Halo-mH2A cells were exposed to SLO for live-cell labeling with 1.4-nm-HAN-594. The fluorescence of 1.4-nm-HAN-594 shows nuclear localization, comparable to a control cell labeled with a cell-permeant fluorophore containing a HaloLigand (Janelia Fluor). In both samples, there is an enrichment in the nucleus, with a heterogeneous signal partially overlapping with peaks of DAPI intensity, consistent with the presence of mH2A in compact chromatin (Fig. 4b and Extended Data Fig. 9a,b). To confirm that 1.4-nm-HAN-594 tagging does not perturb chromatin ultrastructure, chromatin organization was visualized by an established protocol for studying chromatin by ChromEMT, a room-temperature TEM method. Briefly, fixed RPE1 Halo-mH2A cells, in the presence or absence of 1.4-nm-HAN-594, were treated with the DNA dye DRAQ5. This was photo-oxidized using 630 nm of excitation and diaminobenzidine (DAB), leading to the formation of DAB polymers on the chromatin surface that are visualized by EM via binding of osmium tetroxide[9]. Dual-axis tomography datasets were collected, and chromatin diameter and packing density were quantified (Extended Data Fig. 9c–e), showing similar distributions between conditions, suggesting that nanogold tagging does not perturb global chromatin packing or ultrastructure.

## Subtomogram analysis of mH2A-labeled 1.4-nm-HAN-594

RPE1 Halo-mH2A cells were treated with SLO in the presence of 1.4-nm-HAN-594, left to recover overnight, seeded onto EM grids and vitrified. CLEM microscopy was performed, and the cells were exposed to FIB milling and TEM imaging (Fig. 4c,d). Subtomogram averaging from all the nucleosomes was performed (Fig. 4e). As there are two copies of H2A within the nucleosome octamer, a single copy could be replaced with an mH2A variant (creating a hybrid or heterotypic nucleosome), or both copies of H2A could be replaced (Fig. 5a). Single and double incorporations of mH2A are highly structurally homologous to

a canonical nucleosome. Intriguingly, it was found that there is a rearrangement of the L1–L1 interface in mH2A-containing nucleosomes[68]. The L1–L1 interface is the only region where the two H2A–H2B dimers interact[56]. Under in vitro conditions, this results in an increased thermodynamic stability of hybrid nucleosomes[69]. As far as we are aware, how mH2A is incorporated into nucleosomes inside the cell has not yet been determined.

To isolate nanogold-labeled nucleosomes, all nucleosomes were picked by template matching on deconvolved tomograms, and subtomogram averaging was performed, resulting in an 18-Å reconstruction (Fig. 4e). Then, 3D classification with local angular sampling using an inverse mask around the nucleosome was performed. Particles were selected for those containing a globular density consistent with the size of the 1.4-nm nanogold probe (Fig. 5b). The average contains a diffuse density corresponding to a flexible nanogold probe 5.5 nm above a nucleosome. Notably, either zero or one nanogold is present in the nucleosome averages, in agreement with in vitro data suggesting that, in nucleosomes containing mH2A, hybrid mH2A composition is the most stable; that is, we did not observe nucleosomes with two mH2A histones. Using the coordinates and Euler angles, labeled and unlabeled nucleosomes were mapped back to the tomogram (Fig. 5c). Overall, this demonstrates that macromolecules of limited copy number can be recovered using our ExoSloNano approach.

## Delivery of 5-nm fluorescence nanogold into the nucleus

To aid future multiplexing endeavors, we sought to determine if 5-nm Alexa-647-nanogold can enter into the nucleus. WT HEK 293T cells were treated with SLO and labeled with 5-nm-Alexa-647-nanogold and visualized by fluorescence microscopy (Extended Data Fig. 8h). The presence of the 5-nm-Alexa-647-nanogold within the nucleus indicates its nuclear entry.

## Discussion

We present a method for delivering, labeling and visualizing macromolecules in their native context at high spatial resolutions by two different modalities. Fluorescently labeled nanogold can be used to label macromolecules in live cells for visualization by fluorescence microscopy and be further identified in EM studies. This strategy is compatible with live cells and does not alter ribosomal localization or heterochromatin architecture. Importantly, this method only requires the addition of a HaloTag to the protein of interest; the subsequent covalent linkage to a nanogold particle occurs within the cell. Notably, this tagging approach does not require complex molecular assemblies or sensitive in vivo reactions. Finally, it can be visualized in different modalities of in-cell cryo-EM. Here, we show its application for in-cell cryo-ET. We anticipate that for in-cell 2D template matching followed by single-particle analysis[18], the presence of gold labels could dramatically reduce the search space for 2D template-matching algorithms, the current rate-limiting step for the wide use of this method.

We show that 1.4-nm and 5-nm nanogold probes can be detected by cryo-ET and demonstrate the feasibility of our approach for future multiplexed labeling, wherein gold particles of different sizes are conjugated to self-ligands, here shown, but not limited to HaloTag. This method can readily be expanded to use SNAP tags, click chemistry, labels that recognize epigenetic markers or any other biological or chemical moiety that can be adapted for labeling. This nanogold labeling and delivery method should be easily transferable to additional labels in order to further extend the toolkit. The ExoSloNano approach presented here will allow for detailed interrogation of complex interactions within the cell and the nucleus, and likely other organelles. Ideal implementation would utilize cell lines with an endogenously tagged macromolecule of interest of which there is a prior estimation of copy number.

The methods presented here would complement the progress within the chromatin body field and provide more tools for probing chromatin organization, in vitro and in situ, facilitating advancement

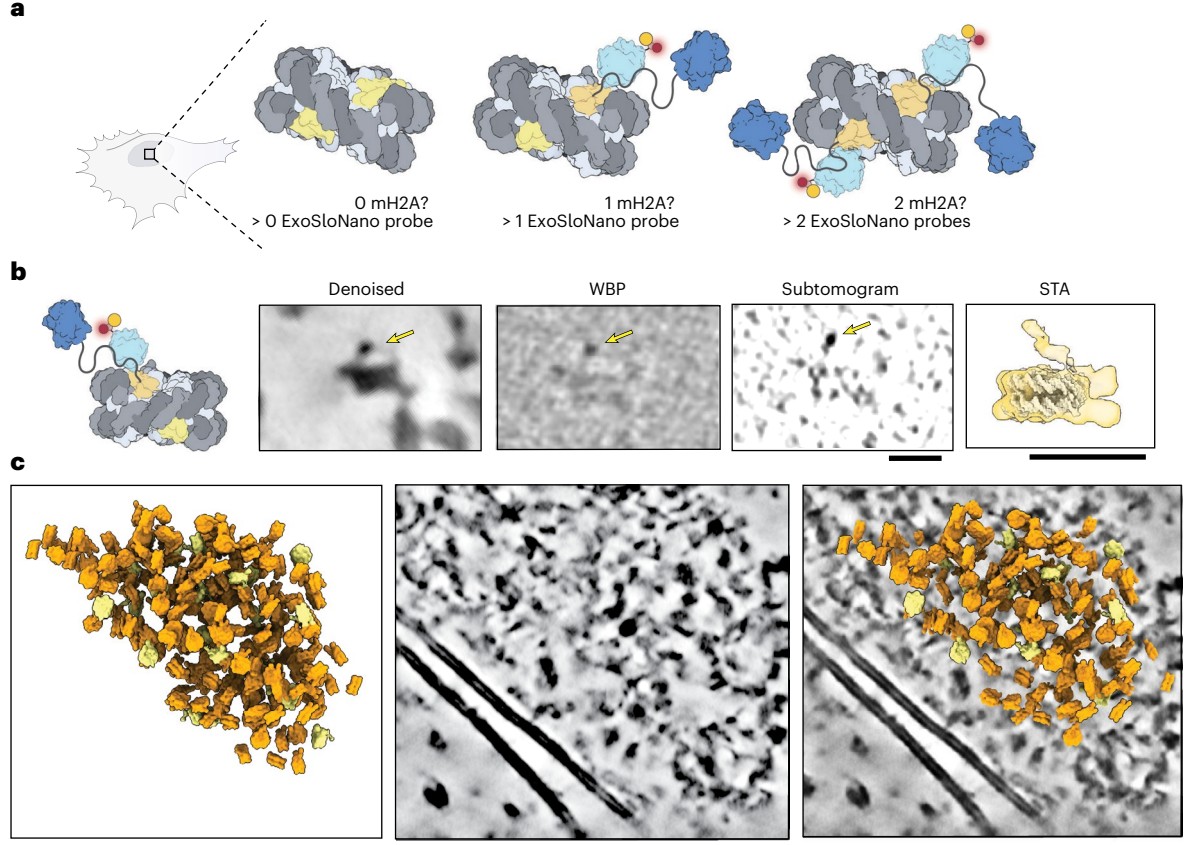

**Fig. 5 | Identifying mH2A-nucleosomes from in situ cryo-ET data. a**, Diagram of the possible mH2A incorporations per nucleosome and the number of ExoSloNano probes. **b**, Single nucleosome from RPE1 mH2A-Halo cells treated with 1.4-nm-HAN-594 of the central slice at 8 Å per pixel from denoised, weighted back-projection (WBP), a normalized subtomogram and from subtomogram analysis (STA). Yellow arrows point to a density consistent with a 1.4-nm nanogold moiety. Subtomogram analysis on nanogold-labeled nucleosomes. **c**, Subtomograms mapped back to the original tomogram containing labeled nucleosomes (yellow) and unlabeled nucleosomes (orange). Scale bars, 10 nm (**b** and **c**). The dataset analyzed represents nine tomograms across three cells.

in studying genome architecture, and genome remodeling events. Combined with recent advances in resolving and analyzing chromatin by cryo-ET[47,52,70,71], our labeling strategy has the potential to directly visualize chromatin regulators as they interface with chromatin in situ.

## Limitations

Treating live cells with a bacterial toxin could be perturbative for the cell phenomena under study, and the extent of the perturbation will need to be assessed to leverage this toxin for delivery of membrane-impermanent nanogold probes[41]. The labeling efficiency for membrane-impermeant probes will be a combination of the efficiency of probe delivery into the cell, saturation of the target and the rate of cellular synthesis of new targets before plunge freezing. Optimization should include known priors (abundance of protein target, cellular localization). In this proof-of-principle study, particles were picked through template-matching algorithms and then the nanogold signal was recovered. As such, this labeling method would be suitable as a label for identifying protein subcomplexes or to distinguish between complexes with high homology inside the cell. We also demonstrated the ability to directly detect nanogold to identify targets. For this to be effective, prior information, for example, cellular localization such as a particular organellar membrane surface or known structural information, can be combined with nanogold presence to localize targets.

Here, we perform subtomogram analysis on particles that, besides being labeled, can easily be picked by template matching or other particle-picking algorithms, or where other priors exist that can facilitate subtomogram location, for example, bound to a membrane or a filament. It is important to note that for any labeling approach including ours, particles that do not satisfy this criterion may be challenging to pick based only on distance to the gold particle since there is a range of orientations and distances that could represent the center of the particle. Ideally, one could generate subtomograms of all these possible particles and process them through subtomogram analysis so that only the particles of interest are kept. However, current subtomogram analysis algorithms depend on well-centered particles. Future developments to overcome this will realize the full potential of ExoSloNano and other labeling methods. As described above, this may not be an issue with 2D template-matching algorithms for in-cell cryo-EM data, where 2D high-resolution template matching is more accurate than the low-resolution 3D modality, and where gold would dramatically reduce the search space for template matching. Alternatives for the precise location of particle centers include adding more than one nanogold particle to the protein of interest and using asymmetric labels.

For extremely low abundant targets, there are more fundamental issues that are inherent to high-resolution in situ tomography—that there is a trade-off between magnification and field of view. In such cases, one must determine how many tilt series must be acquired that would contain the target of interest, and assess if this target is feasible. This could be mitigated by other prior knowledge, such as location of target to subcellular regions or cellular structures[3,72]. Another challenge with low abundance can be that fluorescence localization cannot be used as a guide, that is, low signal for single molecules in

cryogenic microscopy or difficulty in precisely correlating to TEM data. Taken together, if the likelihood of capturing the target in a tilt series is extremely low, this and other methods to label proteins within cells for cryo-ET are unlikely to overcome those preexisting challenges. We hope our recent review may be helpful to design successful in-cell cryo-ET projects[33].

## Outlook

Currently, in situ cryo-EM is far from the theoretical limitations on detection of molecules of sub-100 kDa[73,74]. To truly expand what we can answer by cellular cryo-ET, we need more tools that are orthogonal in their labeling methodologies. This method in conjunction with existing probes such as DNA Origami Signpost, FerriTag and GEM will further empower cell and structural biologists to identify macromolecules across scales[12–14]. We encourage academic and commercial chemistry efforts to lead the development of homogeneous and monovalent nanogold particles of additional sizes and shapes. Eventually, swapping nanogold with a scanning transmission electron microscopy with electron energy loss spectroscopy (STEM-EELS) compatible probe will further expand in situ molecular identification of novel targets inside cells by cryo-ET[75], which will enable future multiplexing of targets using multiple label sizes and EELS signatures. We hope that the method presented here empowers the cellular cryo-ET community to expand the accessible proteome to advance the potential of structural cell biology.

## Online content

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

## Methods

### Cell culture

RPE1 H2AFY and WT RPE1 cells were cultured in DMEM with 4.5 g l$^{-1}$ D-glucose and L-glutamine (Gibco, 11965-092) supplemented with 10% fetal bovine serum (Gibco One Shot, A3160401) and Antibiotic–Antimycotic solution (100 U ml$^{-1}$ penicillin, 100 μg ml$^{-1}$ streptomycin; Gibco, 15240062). U2OS C32 Halo-CTCF cells were cultured in DMEM (low glucose, GlutaMAX Supplement, pyruvate; Fisher Scientific, 10567014) supplemented with Antibiotic–Antimycotic (Gibco, 15240062) and 10% fetal bovine serum (Gibco One Shot, A3160401). HEK 293T L29-HaloTag cells were cultured in DMEM with 4.5 g l$^{-1}$ D-glucose and L-glutamine (Gibco, 11965-092) supplemented with 10% fetal bovine serum (Gibco One Shot, A3160401) and Antibiotic–Antimycotic (100 U ml$^{-1}$ penicillin, 100 μg ml$^{-1}$ streptomycin; Gibco, 15240062). Cells were cultured at 37 °C and 5% $CO_2$. All cell lines were tested regularly for mycoplasma by PCR (Myco-Sniff mycoplasma PCR detection kit).

### Validation of CRISPR knock-in HEK RPL29-Halo

CRISPR knock-in HEK L29-Halo cells were validated through Sanger sequencing of the endogenous *RPL29* gene. The linker connecting RPL29 and HaloTag7 is GSGGSAEIGTGFPFDPHYVEVLGER. The gene product was verified by a western blot against HaloTag, and fluorescence microscopy.

### HaloTag CRISPR–Cas9-mediated endogenous genome editing

To generate the targeting plasmid, Gibson assembly was used to clone a V5 epitope, followed immediately by a HaloTag into pUC19. The V5-Halo tag is upstream of H2AFY and is joined using a GDGAGLIN-linker. To generate the guide RNA plasmid, the guide RNA sequence (5′ CAC-CGCCCACCGCGGCTCGACATGG 3′) was cloned into PX458 using Bbs1 digestion and established protocols[76]. To generate an endogenous H2AFY-HaloTag cell line, 1 million hTERT immortalized RPE1 cells (American Type Culture Collection, CRL-4000TM) were electroporated with 7.5 μg of targeting plasmid and 6 μg of PX458 using the Neon transfection system (Invitrogen). Following electroporation, 50 nM JF-Halo ligand (647, Promega) was added to cells, and fluorescence-activated cell sorting was used to purify cells with a Halo-tagged knock-in. Single-cell colonies were genotyped using the following primers, which were designed to amplify the N-terminal region of H2AFY. PCR fragments were separated by gel electrophoresis, obtaining the expected band shift, and bands were gel purified and sequenced using Sanger sequencing to confirm V5-Halo knock-in.

Forward primer: 5′ TCCAGCGGAGTGCATCACC 3′
Reverse primer: 5′ TTCTTGATGTACCGCAGCATCCG 3′

### Flow cytometry and protein abundance

The absolute protein abundance of H2AFY was determined through flow cytometry when compared to a known standard[49]. U2OS C32 Halo-CTCF cells were cultured in DMEM supplemented with 1 g l$^{-1}$ glucose and 110 mg l$^{-1}$ sodium pyruvate (Gibco, 10567-014). U2OS C32 HaloTag-CTCF cells were a gift from R. Tijan's lab (UC Berkeley). U2OS C32 Halo-CTCF cells and RPE1 H2AFY-Halo cells were cultured in six-well plates to 90% confluency, to which 1 μM HaloTag TMR ligand (Promega, G825A) was supplied for 30 min. Cells were washed twice with 1× DPBS, trypsinized with 0.5% Trypsin (Gibco, 15400054), gently resuspended in HBSS (Gibco, 14025134), spun down and gently resuspended in HBSS with 50 μg ml$^{-1}$ deoxyribonuclease I from bovine pancreas (Sigma, D4263). For each sample, >10,000 events/rates were analyzed on a BD Biosciences LSR II flow cytometer using a 561-nm excitation laser line and a 582/15 bandpass filter cube. Absolute protein abundance was determined using an established protocol[49].

### SLO treatment

SLO treatment was adapted from established methods[77]. HEK 293T WT and RPL29-Halo cells were cultured on fish gelatin-coated 24-well plates. When cells reached 70–80% confluency, they were treated with 40–100 units of SLO from *Streptococcus pyogenes* (25,000–50,000 U, Sigma-Aldrich, S5265). Before use, SLO was activated with 10 mM TCEP, pH 7.0, for 20 min at 37 °C. Cells were exposed to SLO for 5 min, then gently washed two times with 1× DPBS, then exposed to CF Dye 488 Dextran Anionic and Fixable (Biotium, 80110) at a molecular weight of 10 kDa, 40 kDa or 250 kDa. Cells were incubated in recovery media consisting of OptiMEM, 10% FBS, 1 mM CalCl$_2$, 1 mM ATP-MgSO$_4$, 1 mM GTP-MgSO$_4$ and 2 mM glucose. In experimental conditions following SLO treatment optimization, cells were cultured to 70–80% confluency and treated with 40–100 units of SLO (Sigma-Aldrich, S5265) for 5 min, then gently washed two times with 1× DPBS. Cells were exposed to 4 μM 1.4 nm-HAN-488, 1.4 nm-HAN-594 or 5 nm-HN for 8 min on a rotating platform (HEK 293 T L29-Halo cells) or (800 nM for RPE1 Halo-mH2A cells), then gently washed two times with 1× DPBS and left to recover in recovery media consisting of OptiMEM, 10% FBS, 1 mM CalCl$_2$, 1 mM ATP-MgSO$_4$, 1 mM GTP-MgSO$_4$ and 2 mM glucose. The authors later used Corning Matrigel hESC-Qualified Matrix, LDEV (Corning, 354288), overnight at 1:200 dilution, instead of fish gelatin to coat plates before SLO treatment.

### Live-cell imaging after SLO treatment

Cells were imaged for 60 h following SLO treatment to monitor cell survival. Cells were incubated in phenol-free OptiMEM media supplemented with 10% fetal bovine serum, 1% Antibiotic–Antimycotic (Gibco), 1 mM CaCl$_2$, with the addition of propidium iodine (0.5 μg ml$^{-1}$, BD Biosciences), and fluorescently labeled Annexin V-AF647 (BioLegend) was supplied to monitor cell death and apoptosis, respectively. Images were acquired on a Nikon Eclipse Ti2-E equipped with a Qi-2 camera and using Nikon Elements 5.02.02 software. Cell survival and cell growth was imaged over 60 h at 37 °C with 5% CO$_2$ in a stage-top incubator (Okolab H301 Bold Line). Three nonoverlapping 3.125-mm$^2$ zones within each well were imaged by DIC, 554/609 and 618/698 nm using a SpectraX light engine (Lumencor) with individual LFOV filter cubes (Semrock). Images were acquired every 20 min. Cell viability was determined by the ratio of live to dead cells.

### Nanogold-HaloLigand probes

1.4-nm-HAN-594, 1.4-nm-HAN-488 and 5-nm-HN conjugates were custom ordered from Nanoprobes. To generate 1.4-nm-HAN-488 probes, Dextran, Alexa Fluor 488; 10,000 MW, Fixable (Thermo Fisher, D22910) was conjugated with mono-Sulfo-NHS-nanogold (2025-30NMOL from Nanoprobes) and HaloTag Succinimidyl Ester (O4) Ligand was from Promega (P6751). The 1.4-nm-HAN-488 and 1.4-nm-HAN-594 arrived as 30 nmol, lyophilized from 1 ml of 0.02 M sodium phosphate buffer, pH 7.4, with 0.15 M sodium chloride, and was rehydrated in 1 ml of ultrapure water to make 30 μM stocks, and 50 μl aliquots were stored at −80 °C in PCR tubes.

### Quantification of fluorescence nanogold delivery

A dilution series of 1.4-nm-HAN-594 was prepared, and fluorescence intensity was measured using a Molecular Devices SpectraMax i3x plate reader. The fluorescence intensity was plotted against the number of 1.4-nm-HAN-594 molecules, and a linear regression was fit to the data. The number of HAN-594 molecules was calculated by multiplying Avogadro's number by the number of moles, which was obtained from the product of molar concentration and sample volume. Fluorescence intensity was then measured from SLO-treated cells with or without 1.4 nm-HAN-594 delivery. Background fluorescence from SLO-treated cells without 1.4-nm-HAN-594 delivery was subtracted. Using the linear calibration curve, the number of internalized HAN-594 molecules was estimated and normalized to the total number of cells, yielding the average number of 1.4-nm-HAN-594 molecules delivered per RPE1 Halo-mH2A cell (Extended Data Fig. 8g). This bulk measurement provides an approximation of the per-cell delivery efficiency.

## InCuyte imaging

Following SLO treatment and nanogold labeling, cells recovered for 4 h in recovery media, which was then replaced with normal media. Cells were placed in an InCyute microscope (Satorius), maintained at 37 °C and 5% CO₂. Four images were acquired using a ×10 objective in each well (three replicates per condition) every 3 h for a period of 4 days. Growth curves were calculated by cell confluence and were pooled between wells and replicates.

## Culturing cells on micropatterned EM grids

EM Quantifoil carbon on gold grids (R 1/4 Au 200-mesh) were plasma cleaned on a Pelco easiGlow plasma cleaner for 1 min at 20 mA on both sides. Then, 10 μl of PLL-PEG (0.5 μg ml⁻¹) in 10 mM HEPES buffer, pH 7.4, was applied for 45 min to 1 h at room temperature to an EM grid, and then washed four times with 1× PBS and then 3 μl of the photoactivatable reagent PLPP (Avéole) was applied to each grid just before patterning. EM grids were imaged on an Eclipse Ti2-E (Nikon) using an S-Plan Fluor ELWD ×20 0.45-NA objective, and the microscope was equipped with a PCO.edge 4.2 bi sCMOS camera (PCO). Grid squares were automatically detected through the Fiji plugin Aveole Leonardo software using the Experimental Wizard's, a size circular pattern to cover the grid square was micropatterned with 1,000 mJ per mm² of a 360-nm laser. After patterning, the grid was washed four times with DPBS. Micropatterned grids were coated with 50 μg ml⁻¹ of fibronectin for 45 min. Approximately 25,000 RPE1 Halo-mH2A cells labeled with 1.4-nm-HAN-594 were plated onto four micropatterned EM grids in a 35-mm Mattek dish (P35G-1.5-20-C). PDMS stencils of 15 mm in diameter with four wells, each well 4 mm in diameter (Aveole 4W001), were applied to the center of the Mattek dish. An EM grid was placed within each well. Four to six hours after seeding, grids were blotted for 4–6 s, and then plunged into a 50:50 mixture of ethane:propane with a custom manual plunger (MPI Martinsried) and then stored under liquid nitrogen.

## Cryo-FIB-milling and cryo-fluorescence

Grids clipped into FIB-compatible Autogrids (Thermo Fisher) were loaded into an Aquilos 2.0 Dual-Beam cryo-FIB/SEM (Thermo Fisher). A layer of organometallic platinum was applied onto the grid. Lamellae were prepared using a Ga²⁺ ion beam at 30 kV in which the beam current was gradually reduced stepwise from 0.5 nA to 10 pA during the thinning procedure, as described in ref. [77] and performed with Thermo Fisher AutoTEM version 2.4. The milling progress was monitored by SEM at 2 kV and 5 kV. Following automated milling, fine polishing was performed at 10 pA for 5–10 min. The final lamella thickness was between 140 nm and 180 nm. Grids were imaged by fluorescence at the Aquilos equipped with iFLM version 1.2.

## Cryo-ET data collection

**5-nm ribosomal dataset.** Lamellae were imaged on a 300-kV Titan G2 Krios Transmission Electron Microscope (Thermo Fisher) using a K3 Summit direct electron detector equipped with a Biocontinuum energy filter (Gatan) at ×81,000 nominal magnification, and the calibrated pixel size calibrated was 1.068 Å per pixel.

Hybrid tilt series were acquired with SerialEM[78] using PACE-tomo[79] scripts. The 0˚ tilt was exposed to 20 e⁻/Å², while the remaining tilts were exposed to 5 e⁻/Å². The tilt-series range was ±35˚ with a 2˚ increment, with a defocus range from −3 μm to −5 μm. The non-hybrid tilt-series data range was ±48˚ with a 2˚ increment, with a defocus range from −3 μm to −5 μm, and a total dose of 180 e⁻/Å².

**1.4-nm ribosomal dataset.** Lamellae were imaged on a 300-kV Titan G2 Krios Transmission Electron Microscope (Thermo Fisher) using a K3 Summit direct electron detector equipped with a Biocontinuum energy filter (Gatan) at ×81,000 nominal magnification, and the calibrated pixel size calibrated was 1.068 Å per pixel. Tilt series were acquired with

PACE-tomo[78]. The 0˚ tilt was exposed to 20 e⁻/Å², while the remaining tilts were exposed to 4.2 e⁻/Å². The tilt-series range was ±35˚ with a 2˚ increment, with a defocus range from −3 μm to −5 μm. The non-hybrid tilt-series data range was ±48˚ with a 2˚ increment, with a defocus range from −3 μm to −5 μm, and a total dose of 180 e⁻/Å².

**Control ribosomal dataset.** Lamellae were imaged on a 300-kV Titan G2 Krios Transmission Electron Microscope (Thermo Fisher) using a K3 Summit direct electron detector equipped with a Biocontinuum energy filter (Gatan) at ×81,000 nominal magnification, and the calibrated pixel size was 1.068 Å per pixel. Tilt series were acquired with PACE-tomo[79]. The 0° tilt was exposed to 20 e⁻/Å², while the remaining tilts were exposed to 4.2 e⁻/Å². The tilt-series range was ±36˚ with a 2° increment, with a defocus range from −3 μm to −5 μm.

**Nucleosome dataset.** Lamellae were imaged on a 300-kV Titan G2 Krios Transmission Electron Microscope (Thermo Fisher) using a K3 Summit direct electron detector equipped with a Biocontinuum energy filter (Gatan) at a ×105,000 nominal magnification, and the calibrated pixel size calibrated was 0.876 Å per pixel. Tilt series were acquired, and each tilt angle was exposed to 4.1 e⁻/Å². The tilt-series range was ±40˚ with a 2˚ increment, with a defocus range from −3 μm to −5 μm.

## Data preprocessing

**5-nm-HN ribosome dataset.** Individual frames were motion-corrected and gain-corrected in Warp (1.09)[46]. High dose 0˚ images were processed separately and then merged with the rest of the data. Frame stacks were created in Warp and aligned through a batch Etomo procedure. Exposure filtering, 3D-CTF estimation and tomographic reconstruction were performed in Warp (1.09). A custom script was used to account for the pre-tilt of the lamella (https://github.com/dgvjay/EM_Scripts/blob/master/massNormalize_mdoc.py/).

## Nanogold randomization for ribosome subtomogram analysis

Particles were picked on Warp-generated deconvolved tomograms using crYOLO (version 1.8.4)[23]. Subtomograms were generated in Warp at 8 Å per pixel, then imported into MATLAB version R2019b, read in by Dynamo[23]. During subtomogram averaging, images are aligned to the nanogold[80]. To overcome this and to recover the ribosomal signal, high-intensity pixels (values greater than 3.3 standard deviations from the mean) were randomized on a per-particle basis to be within one standard deviation of the mean using a custom MATLAB script (subtomo_randomize.m; 'Code availability'). The purpose was to effectively hide the gold particles from the image. The nanogold-randomized subtomograms were aligned in RELION-3.1.4 yielding a low-resolution (but, now ribosomal aligned) subtomogram average. Particles were then re-extracted and a low-pass-filtered mask was used subsequently during further subtomogram analysis to mask out the nanogold in RELION (3.1.4)[81].

## Pairwise distance measurements

The nanogold coordinates come from the RELION subtomogram average in which the alignment converged on the nanogold particle (Fig. 2c; subtomogram average panel). After randomizing the nanogold signal and obtaining a low-resolution ribosomal average, the ribosomal average was re-centered around the last ordered residue of RPL29 (12.9 nm away), resulting in a new set of coordinates that correspond to RPL29. To determine the distances between the RPL29 and the nanogold, a matrix of all pairwise distances was determined using the Euclidean distance formula between both datasets in MATLAB. Then the minimum value or the minimum pairwise distances between each ribosome and nanogold was determined and a distribution was plotted. To determine the upper bound, a Worm-Like Chain model was used to model the unordered/flexible residues between the last ordered residue of RPL29 and the HaloTag, and the distance was estimated to

be ~3.6 nm. As the distance between the HaloTag and the nanogold is based on (a) the 10,000 MW Dextran (roughly 3.6 nm) in diameter and (b) the HaloLigand itself, which is about 22 Å from end-to-end, but also enters the HaloTag cavity, a pairwise distance of up to 10 nm away was taken as the maximum extended distance. See Extended Data Fig. 4 for a visual explanation.

### Subtomogram analysis on nucleosome dataset

For the nucleosome dataset, the tomograms were initially denoised using Warp's[46] Noise2Noise denoising scheme for 60,000 iterations. The denoised tomograms were then further corrected for the missing wedge using IsoNet[82] with 11 tomograms for 30 iterations. Subsequently, they were segmented with DeepFinder[22] to determine particle positions. Masks were applied in IMOD[83] to exclude particles outside the chromatin region. The remaining particles were processed with a novel context-aware template-matching (CATM) algorithm (https://git.biohpc.swmed.edu/rosen-lab/catm/), specifically designed for crowded environments, to determine nucleosome orientations[83]. Following the CATM workflow, particle coordinates and orientation data were used to reconstruct subtomograms at a resolution of 8 Å per pixel in Warp[46]. These subtomograms were then averaged into an initial model[84], which was low-pass filtered to 40 Å and used as a reference for further classification and refinement in RELION-4 (ref. [85]). Aligned particles were subsequently re-extracted in Warp at 4 Å per pixel and further refined in RELION-4. The resolution was determined using Fourier shell correlation between two half-maps. To classify and refine the nucleosome with nanogold labeling, an inverted mask was applied. This mask down-weighted the nucleosome density to 0.2 and up-weighted the density outside the nucleosome to 0.8. The final subtomogram averaging of nucleosomes with nanogold (Fig. 5b) represents a composite map, combining the nucleosome subtomogram average with the weighted nanogold subtomogram average.

### Whole-tomogram detection of 1.4-nm-HAN-488-labeled ribosomes in situ cryo-tomograms

Fourteen well-aligned tilt series initially collected at 1.068 Å per pixel (tilt-series alignment with mean residual ~1 nm) were reconstructed at 3 Å per pixel by weighted back-projection using Warp (1.10)[46] and analyzed in MATLAB version R2019b. Tomograms were averaged over ten slices and normalized between 0 and 1. High-intensity voxels that were 3.3 standard deviations away from the mean were isolated. To create a binary mask, isolated voxels were set equal to 1 and everything else was set equal to 0. Neighboring voxels of value 1 were connected and grouped and the new center was determined. Background signal was any unpaired voxels equal to 1 and subsequently were removed. In theory, the number of voxels for which a 1.4-nm sphere (radius of 0.7 nm) sampled over a grid of 3 Å per pixel is 53 voxels. Two cutoffs were determined. First, any coordinates greater than 55 voxels were taken to be gallium, ice or membrane and were filtered out. To account for undersampling due to the boundary effect of sampling a spherical object over a grid (see Extended Data Fig. 6 for schematic), the lower bound was set to be 40 voxels. The number of voxels for these final values were converted to nm³ and plotted (Extended Data Fig. 6g).

### Isolation of nanogold-labeled ribosomes from HEK 293T cells

Four 10-cm dishes were coated with 1 ml of Corning Matrigel hESC-Qualified Matrix, LDEV (Corning, 354288) overnight at a 1:200 dilution and then HEK 293T L29-Halo cells were seeded. Cells were treated with SLO at 90% confluency with 4 µM 1.4-nm-Halo-Alexa-488-nanogold. Four 10-cm plates of HEK L29-Halo cells were treated with SLO without 1.4-nm-Halo-Alexa-488-nanogold and used as a control. Cells were harvested 18 h after treatment and lysed in 700 µl of lysis buffer (20 mM Tris-Cl pH 8.0, 150 mM NaCl, 1% Triton-X 100, 15 mM MgCl₂, 1 mM TCEP, 40 mM NEM, 40U Turbo DNaseI, EDTA-free protease inhibitor cocktail in DEPC treated water) by vigorous pipetting, followed by incubation on

ice for 15 min. Cell lysates were clarified by centrifuging at 15,000 rpm for 10 min at 4 °C and total RNA content was quantified by nanodrop (Thermo Scientific). Lysate containing 1,000 µg RNA was loaded onto a 10–50% sucrose gradient (supplemented with 150 µg ml⁻¹ cyclohexamide, 80 U SuperaseIn) prepared using Gradient Master 108 (BioComp). Gradients were centrifuged in a SW-41 rotor at 41,000 rpm for 2 h at 4 °C and 1-ml fractions were collected using a PGFip piston gradient fractionator (Biocomp). The 80S ribosomal fraction was pooled and the sucrose was removed by dialysis in 30 mM HEPES pH 7.6, 150 mM NaCl, 2 mM MgCl₂, 1 mM TCEP overnight at 4 °C and then concentrated to 514 nM with a Gibco 100-kDa molecular-weight-cutoff filter.

### In vitro cryo-ET grid prep and data collection

From SLO-treated and 1.4-nm-HAN-488 delivered RPL29-Halo cells (ribosomal concentration 514 nM), 4 µl was applied to plasma-cleaned (Pelco easiGlow plasma cleaner for 1 min at 20 mA on each side) Quantifoil 400-mesh 1.2/1.3 copper grids and plunge frozen into a 50:50 mix of ethane and propane. Micrographs were collected on a 300-kV Titan G2 Krios Transmission Electron Microscope (Thermo Fisher) using a K3 Summit direct electron detector equipped with a Biocontinuum energy filter (Gatan) at ×81,000 nominal magnification with a calibrated pixel size of 1.068 Å per pixel, acquired with PACE-tomo[79]. Each tilt was exposed to 4 e⁻/Å², the tilt-series range was ±54° with a 3° increment, with a defocus range from −1 to −3 µm, and the total dose was 150 e⁻/Å².

### Subtomogram analysis of isolated ribosomes

Individual frames were motion-corrected and gain-corrected in Warp (1.10)[46]. Frame stacks were created in Warp and aligned through a batch Etomo procedure. Exposure filtering, 3D-CTF estimation and tomographic reconstruction were performed in Warp (1.10). Particles were picked in crYOLO, and subtomograms were extracted in Warp (1.10) and averaged in RELION (3.14).

### Labeling efficiency

The labeling efficiency was determined by nanogold-labeled ribosomes divided by the combined number of nanogold-labeled ribosomes and unlabeled ribosomes[85].

$$\text{Labeling efficiency} = N_{\text{nanogold labeled ribosomes}} /$$
$$(N_{\text{nanogold labeled ribosomes}} + N_{\text{nanogold unlabeled ribosomes}})$$

### iPS cells to iNeurons and SLO treatment of iNeurons

Human induced pluripotent stem (iPS) cells engineered to express mNGN2 under a doxycycline-inducible system at the *CLYBL* locus (kind gift from M. Ward, NIH)[50] were cultured in a 10-cm tissue culture dish (Genesee Scientific, 25-202), precoated with KnockOut DMEM/F-12 medium (Gibco, 12660-012), supplemented with a 1:100 dilution of Matrigel human embryonic stem cell-Qualified Matrix (Corning, 354277) for 24 h. The iPS cells were cultured in Essential 8 Basal Medium (Gibco, A1517001) supplemented with 50 nM Chroman 1 (MedChemExpress, HY-15392) for 24 h, after which the medium was replaced with fresh Essential 8 Basal Medium to remove Chroman 1. After reaching 70–90% confluency, cells were detached with StemPro Accutase (Gibco, A1110501), centrifuged for 5 min at 300 rcf, and then resuspended to induce differentiation in induction media consisting of KnockOut DMEM/F-12 (Gibco, 12660-012) supplemented with 1:100 dilution of N2 supplement (Gibco, 17502048), 1:100 GlutaMAX (Gibco, 35050-061) and a 1:100 dilution of MEM NEAA (Gibco, 11140050) and 2 µg ml⁻¹ doxycycline (Clontech Labs 3P, 631311) and supplemented with 50 nM Chroman 1 to initiate neuronal differentiation in a six-well plate (Genesee Scientific, 25-105MP) at 500,000 cells per well. On day 4, the medium was replaced with fresh induction media without Chroman 1 and was incubated at 37 °C for 72 h. Induced

iPS cells were detached from the six-well plate with StemPro Accutase (Gibco, A1110501) and 20,000 cells per well were seeded into an ibidi u-Slide eight-well glass-bottom plate (ibidi, 80827-90), pretreated with 0.1 mg ml⁻¹ poly-L-ornithine hydrobromide (Sigma-Aldrich, P3655-10MG) for 72 h at 37 °C and 25 µg ml⁻¹ of laminin-1 at 37 °C, 2–6 h before seeding, and cultured in BrainPhys neuronal medium (STEMCELL Technologies, 05790) supplemented with 2 µg ml⁻¹ doxycycline (Clontech Labs 3P, 631311), B27 supplement (1:50 dilution; Thermo Scientific, 17504044), 10 ng ml⁻¹ BDNF (Thermo Scientific, 450-02), 10 ng ml⁻¹ NT-3 (Thermo Scientific, 450-03) and 1 µg ml⁻¹ laminin-1 (R&D Systems, 3446-005-01) for 7 days, with a half media change done every 4–5 days. iNeurons were treated with 40–100 units of SLO from *Streptococcus pyogenes* (25,000–50,000 U, Sigma-Aldrich, S5265). Before use, SLO was activated with 10 mM TCEP pH 7.0 for 20 min at 37 °C. Cells were exposed to SLO and 0.5 nM Anit-MAP2 antibody (Abcam, EPR19691) for 5 min. Cells were incubated in recovery media consisting of OptiMEM, 10% FBS, 1 mM CalCl₂, 1 mM ATP-MgSO₄, 1 mM GTP-MgSO₄ and 2 mM glucose.

### Room-temperature TEM
Cells were treated with SLO as described using 4 µM 1.4 nm Alexa-594-Halo-nanogold and fixed in 2.5% glutaraldehyde/0.1 M sodium cacodylate buffer for 5 min at room temperature, and then 2 h on ice. Cells were washed several times in 0.1 M cacodylate buffer and transferred to a dark room using only a red lamp. HQ silver enhancement kit (2012-45 ML, Nanoprobes) was prepared as described and incubated with cells for 10 min. After this time, cells were placed on ice and washed 5×/0.1 M cacodylate buffer. Cells were next stained with 0.7% osmium tetroxide (19150, Electron Microscopy Sciences) diluted in 0.1 M cacodylate for 30 min on ice rocking. Cells were washed in water, dehydrated and embedded in Durcupan resin (Sigma A-D) using standard protocols. Then, 60-nm sections were obtained using a diamond knife (Diatome), and collected on formvar slot grids (VWR, FF2010-Cu-25). Images were acquired on a Biotwin TEM operating at 120 kV using a CCD camera (AMT). For ribosome quantification, images were acquired at ×1,900.

### Ribosome quantification in resin-embedded TEM images
Ribosome quantification was performed using Imaris v10.2.2. To quantify the number of ribosomes, a spot detection feature was applied, which, through thresholding, reliably detected gold particles due to their high contrast. To determine the number of ribosomes detected per cell, we first measured the average volume imaged in our TEM images, using the cell membrane to create a surface. This provides the number of ribosomes detected per cell. To extrapolate this number to the volume of the cell, the average HEK cell volume was next calculated by fluorescence imaging. Phalloidin was used to label actin filaments and thus the cell boundary, thereby allowing a surface to be created and the total HEK cell volume to be calculated. Finally, to quantify the size of gold particles, a surface feature was applied, allowing the edges of gold particles to be masked and their diameter measured.

### ChromEMT and plastic section tomography
Cells were labeled with nanogold via SLO delivery as described. Cells were fixed in 2.5% glutaraldehyde (Ted Pella, 18420) diluted in 0.1 M sodium cacodylate buffer (Ted Pella, 18851) for 5 min at room temperature, and then 2 h on ice. To allow for DNA staining, embryos were incubated with DRAQ5 (Thermo Fisher Scientific, 62251) diluted at 1:500 in 0.1 M sodium cacodylate buffer for 1 h on ice. For photo-oxidation, 12 mg of 3,3′-DAB (Sigma, D8001) was dissolved in 1 ml of 0.1 M HCL by vortexing. This was diluted by adding 19 ml of 0.1 M cacodylate buffer, which was passed through a 0.2-µm filter. Sodium cacodylate buffer was exchanged for cold DAB solution immediately before photo-oxidation. This was performed using a ×63 oil-immersion objective and custom-built wide-field fluorescence microscope with a high-powered light source (642-nm CW fiber laser, MPB Communications) and square-core multimode fiber optics (Thorlabs) to deliver the laser light into the illumination path. We experimentally determined that illuminating the sample with 100 mW of laser power for 30 s–1 min provides optimal oxidization.

### Room-temperature TEM
For tomography, 300-nm sections were obtained and deposited on copper slot grids (Luxel, SKU: C-S-M-L) that were first glow discharged for 30 s. For reconstruction, the grid was placed on a drop of 15-nm gold protein A gold (PAG) fiducial particles (diluted at 1:50 in double-distilled water) for 30 s, before being rinsed in double-distilled water and removing excess liquid using filter paper. These steps were repeated for the opposite side of the grid, which was subsequently left to air-dry. Dual-axis tilt series were acquired on a 200-kV Tecnai F20 Transmission electron microscope (Thermo Fisher Scientific), using an Eagle 4k × 4k CCD camera (Thermo Fisher Scientific). The tilt series were acquired using SerialEM[2] at a tilt range of ±60° using 1° increments at a magnification of ×13,500, corresponding to a pixel size of 1.62 nm at a binning size of 2.

### Room-temperature TEM reconstruction and analysis
Dual-axis tilt series were aligned in IMOD[83] eTomo and reconstructed with a SIRT filter. Chromatin was quantified from the final tomographic z-stacks using the MIA modularized image analysis workflow plugin (v1.2.6) for ImageJ[86].

### Reporting summary
Further information on research design is available in the Nature Portfolio Reporting Summary linked to this article.

## Data availability
Electron Microscopy Data Bank (EMDB) accession codes are: 1.4-nm nanogold-labeled ribosomes STA (EMD-71202); 5-nm nanogold-labeled ribosome (EMD-71205); the nucleosome STA (EMD-71113); and the nanogold-labeled nucleosome (EMD-71211). Electron Microscopy Public Image Archive (EMPIAR) accession codes are: 1.4-nm-HAN-488-nanogold-labeled ribosomal dataset (EMPIAR-12907); the 1.4-nm-HAN-594 nucleosome dataset (EMPIAR-12908); the 5-nm-HN dataset (EMPIAR-12919); and the in vitro 1.4-nm-HAN-488 purified ribosomes from SLO-treated cells dataset (EMPIAR-12918). Source data are provided with this paper.

## Code availability
Code and scripts are available via GitHub from the lab of E.V. (https://github.com/villalab/exoslonano/), the lab of M.K.R. (https://git.biohpc.swmed.edu/rosen-lab/catm/) and https://github.com/dgvjay/EM_Scripts/. Cell lines are available upon request.

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

## Acknowledgements

We thank P. Selvin for insightful initial conversations about using SLO for probe delivery, N. Grigorieff, B. Ren, T. Laughlin and members of the lab of E.V. for helpful discussions. We thank the following imaging facilities at UCSD: Neuroscience Imaging Center, the Nikon Imaging Center at UC San Diego, and the UCSD Cryo-Electron Microscopy Facility, which was built and equipped with funds from UCSD and an initial gift from the Agouron Institute. We thank the UCSD Physics Computing for computational support. M.K.R. and E.V. are Howard Hughes Medical Institute Investigators. This work was supported by NIH grants NCI K00 CA223029 (to L.N.Y.), K99 HD112607-02 (to A.S.), R35GM148339 (to E.J.B.), R35-GM141736 (to M.K.R.), NIH U54 AI170856 (to E.V.), an EMBO long-term postdoctoral fellowship ALTF 902-2019 (to A.S.), a Pew Scholar Award (to E.V.) and an NSF DBI 1920374 (to E.V.). Molecular graphics and analyses were performed in part with UCSF ChimeraX, developed by the Resource for Biocomputing, Visualization, and Informatics at the University of California, San Francisco, with support from NIH R01-GM129325 and the Office of Cyber Infrastructure and Computational Biology, National Institute of Allergy and Infectious Diseases. Computational support from the PCF. HEK 293T L29-Halo was a kind gift from H. An. Human iPS cells engineered to express mNGN2 under a doxycycline-inducible system at the *CLYBL* locus were a kind gift from M. Ward, NIH.

## Author contributions

L.N.Y., A.S. and E.V. conceived the research. L.N.Y. and A.S. designed the experiments. A.S. generated the RPE1 H2AFY cell line. L.N.Y. and F.S. prepared the samples for cryo-FIB-ET. L.N.Y. and F.S. prepared cryo-lamellae. L.N.Y. collected the cryo-ET data. J.H. established PACE-tomo at UCSD. M.R. created the schematics. L.N.Y. carried out tomographic reconstruction, subtomogram averaging and image analysis. H.Z. helped analyze the chromatin data. A.S. prepared, imaged and analyzed resin-embedded samples. W.A.F. assisted with cell culture of iPS cells to iNeurons and HEK 293T cells. M.N. and E.B. provided guidance and resources for the sucrose gradient ribosomal purification from SLO-treated cells. L.N.Y., A.S. and E.V. wrote the paper with contributions from all authors.

## Competing interests

The authors report no competing interests.

## Additional information

**Extended data** is available for this paper at https://doi.org/10.1038/s41592-025-02928-4.

**Correspondence and requests for materials** should be addressed to Antonio J. Giraldez or Elizabeth Villa.

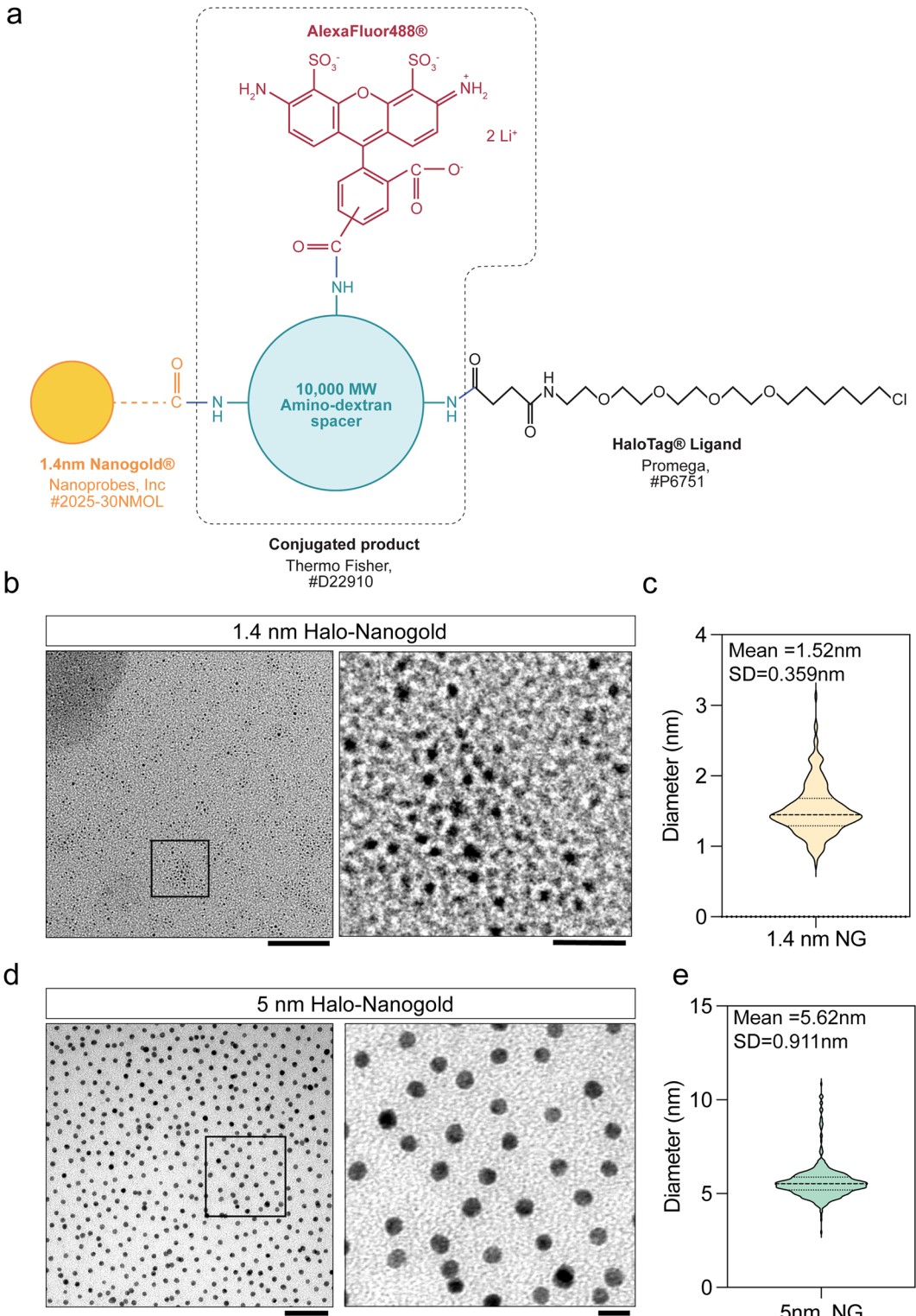

**Extended Data Fig. 1 | Size distribution of Halo-nanogold particles. (a)** 1.4 nm-HAN-488 probe design with catalog numbers. **(b)** TEM image of 1.4 nm Halo nanogold, black inset shows zoomed region (right). **(c)** Violin plot showing quantification of particle diameter of the image shown in a. N = 224 particles. Experiment was performed once. **(d)** TEM image of 5 nm Halo nanogold, black inset shows zoomed region (right). **(e)** Violin plot showing quantification of particle diameter of the image shown in c. N = 505 particles. Experiment was performed once. Scale bars: **(b)** 100 nm (left), 5 nm (right); **(d)** 50 nm (left), 10 nm (right).

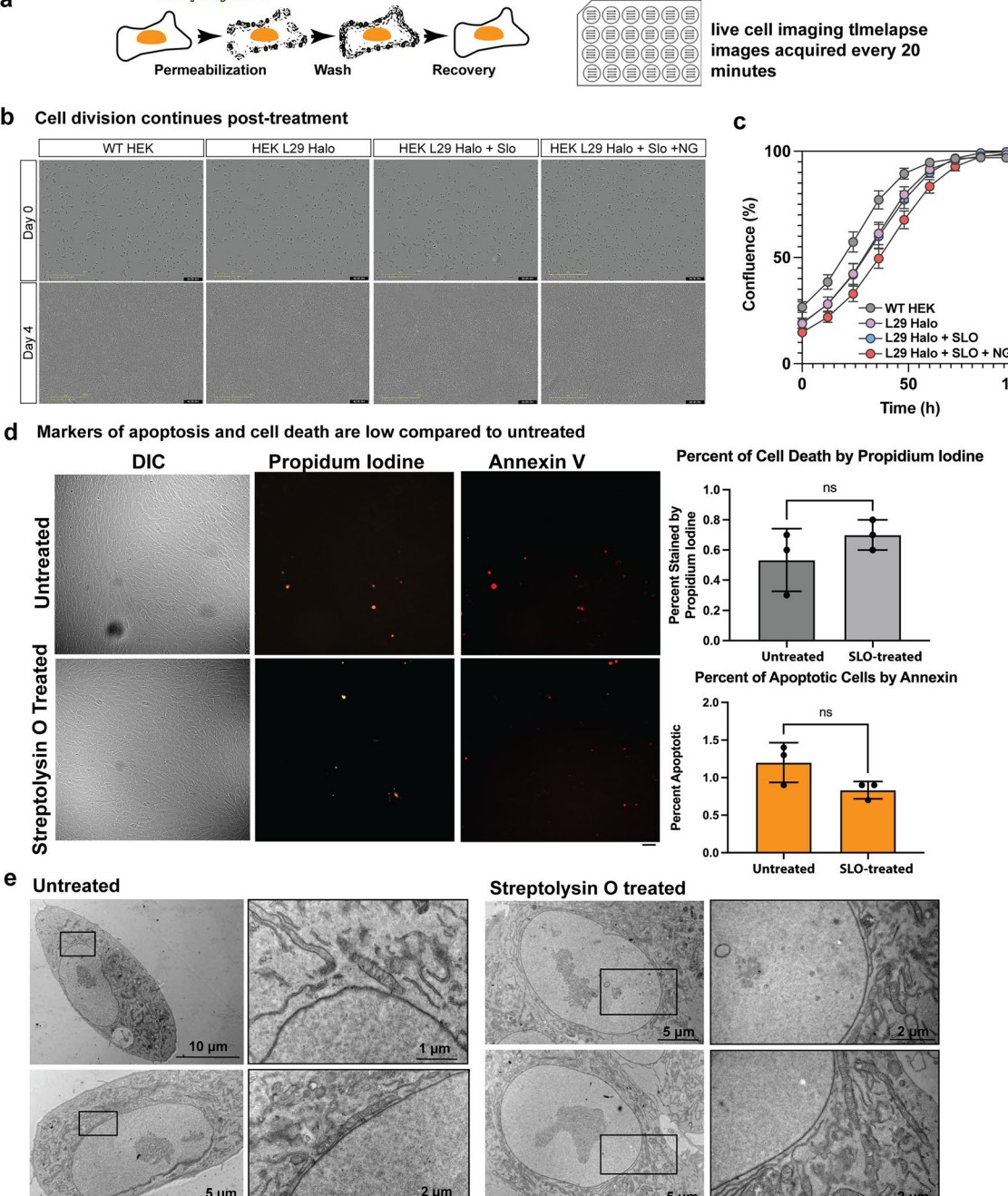

**Cells recover from Streptolysin O (bacterial toxin) treatment**

**Extended Data Fig. 2 | Cells recover from Streptolysin O (SLO) treatment.**
(a) Schematic of pore formation mediated by SLO treatment to mammalian cells in cell culture. (b) InCuyte images of HEK cells in the indicated conditions. Images show cell confluence at time 0 (day 1) and 100 h (day 4). Scale bar is 400um. (c) graph showing the group rate of the indicated conditions. Data are shown as mean ± SD, error bars indicate SD, p = 0.906 (Kruskal–Wallis test, n.s.). N = 3 replicates. (d) Following SLO treatment, cell viability was assessed using propidium iodine (PI), and annexin V (marker of apoptosis). Quantification across three wells of cells in treated and untreated samples shows cell death is low. Control: Mean = 0.5, SD = 0.21. SLO-treated: Mean = 0.7, SD = 0.1. p = 0.5 (two-sided Mann-Whitney U Test, ns). and apoptosis is low. Control: Mean = 1.2, SD = 0.3. SLO-treated: Mean = 0.8, SD = 0.1. p = 0.3 (two-sided Mann-Whitney U Test, ns.). (e) The ultrastructure of the cell is maintained following SLO treatment, as seen by room temperature TEM. Boxes are zoomed-in regions.

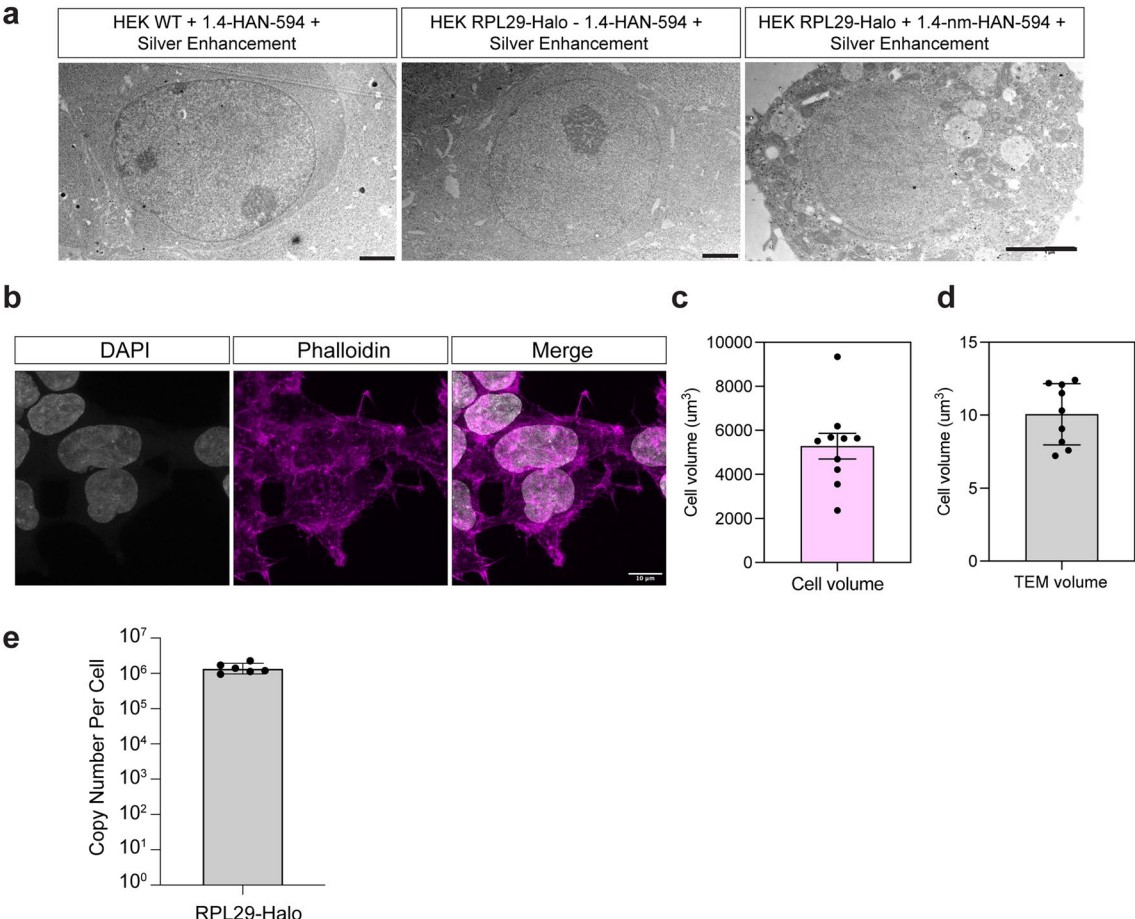

**Extended Data Fig. 3 | Nanogold enhancement is specific to HEK 293 T RPL29-Halo. (a)** Transmission electron microscopy (TEM) images of HEK 293 T cells that were enhanced for 10 min. Left: WT cells without Halo-nanogold, middle: WT cells treated with 3 uM 1.4 nm-HAN-488; right: RPL29-Halo cells treated with 3 uM 1.4 nm-HAN-488. The scale is 2 um. **(b)** Immunofluorescence images of HEK 293 T cells with DAPI and phalloidin staining as a maker for the cell edge.

**(c)** Quantification of cell volume from the cells shown in A. N = 10 cells. Mean =5283, error bars show SD, SD = 1.85. **(d)** Quantification of cell volume in TEM images. N = 9. Mean =10.05, error bars show SD, SD = 2.1. **(e)** Quantification of ribosomal copy number in HEK 293 T RPL29-Halo cells, Mean=1.4×10⁶, SD = 4.8×10⁵, N = 6 independent experiments.

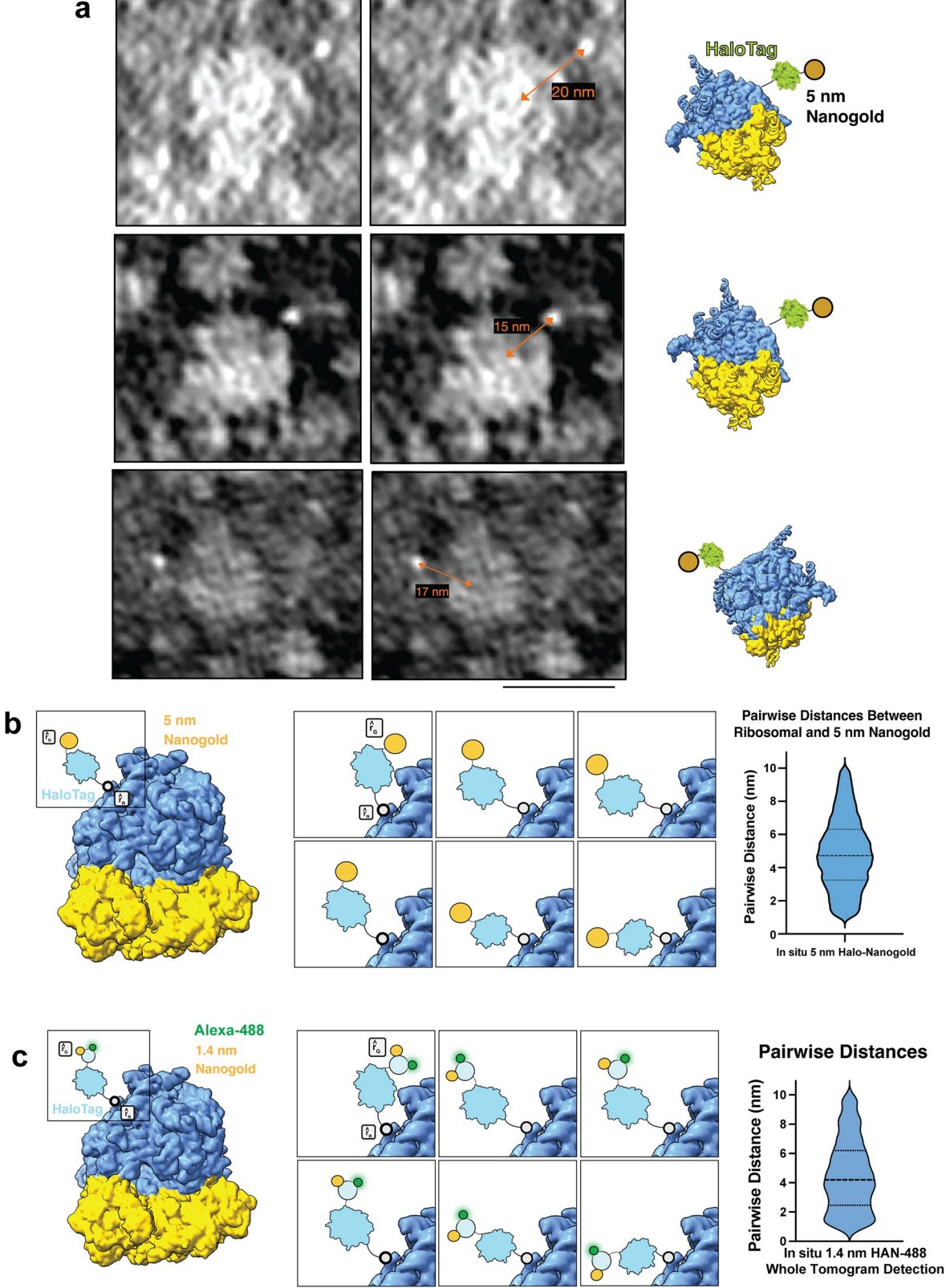

**Extended Data Fig. 4 | Pairwise distance between the label and the target vary in accordance to construct design. (a)** Central slice of three examples of 5-nm-HN-labeled ribosomes from denoised tomograms. Contrast-inverted, that is white-on-black. On the right are ribosome models showing expected location of gold nanoparticle. Scale bar: 20 nm. **(b)** Left: schematic of a 5 nm Halo-nanogold labeled ribosome. Middle: insets show potential arrangements of the different components of the label (linker, Halo in cyan, gold in yellow) and resulting pairwise distances. Right: violin plot of measured pairwise distances. **(c)** Left: schematic of a 1.4 nm HAN-488 labeled ribosome. Middle: insets show potential arrangements of the different components of the label (linker, Halo in cyan, Dextran in baby blue, fluorophore in green, gold in yellow). Right: violin plot of the measured pairwise distances.

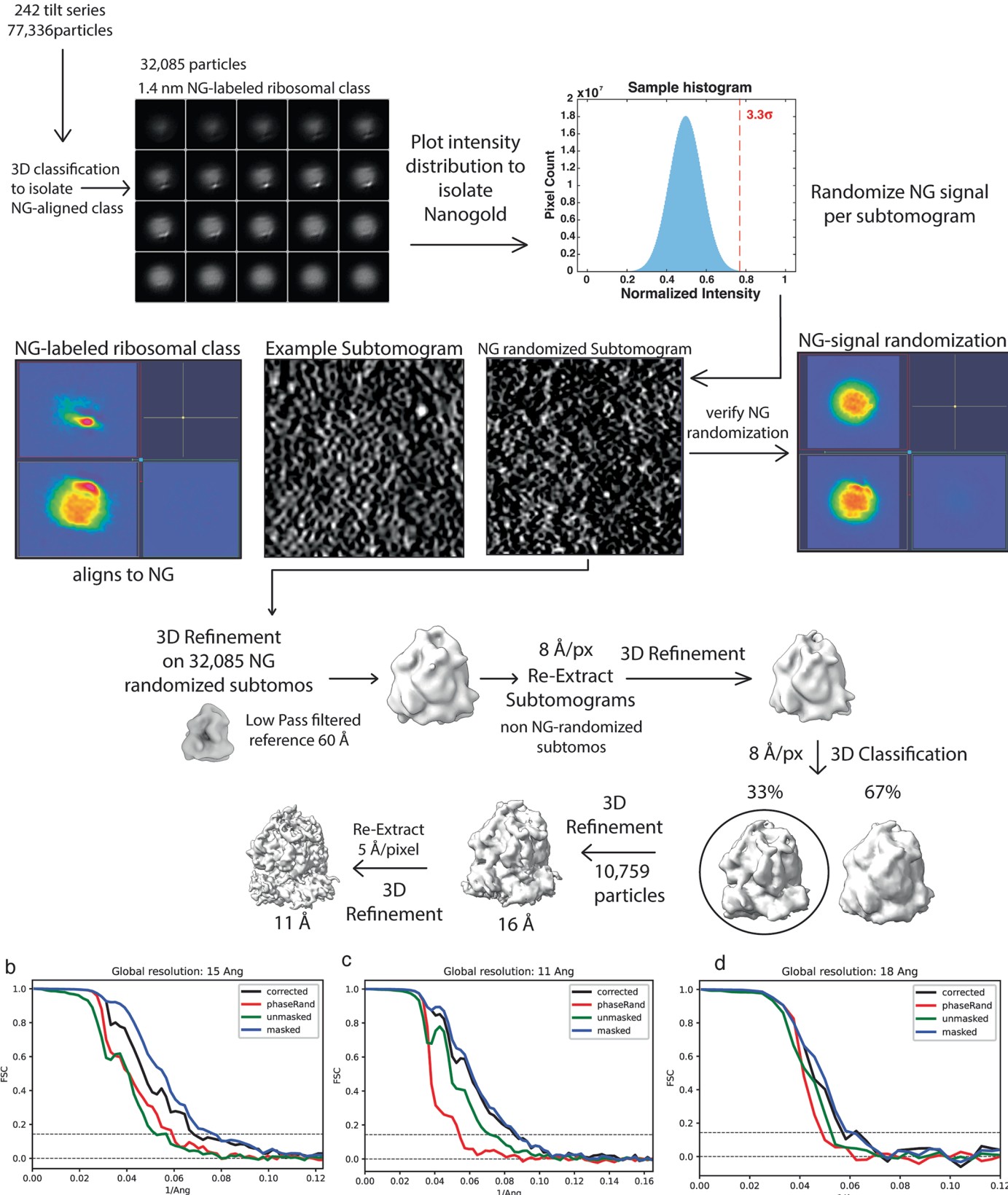

**Extended Data Fig. 5 | Data processing and NG-signal randomization on 1.4 nm-HAN-488 labeled ribosomes for subtomogram analysis. (a)** Data processing and NG-signal randomization on 1.4 nm-HAN-488 labeled ribosomes for subtomogram analysis **(b)** FSC curve from 5 nm nanogold-labeled ribosomal subtomogram average. **(c)** FSC curve from 1.4 nm HAN-488 labeled ribosomal subtomogram average. **(d)** FSC curve from all nucleosomes from SLO-treated nuclear delivery of 1.4 nm-HAN-594.

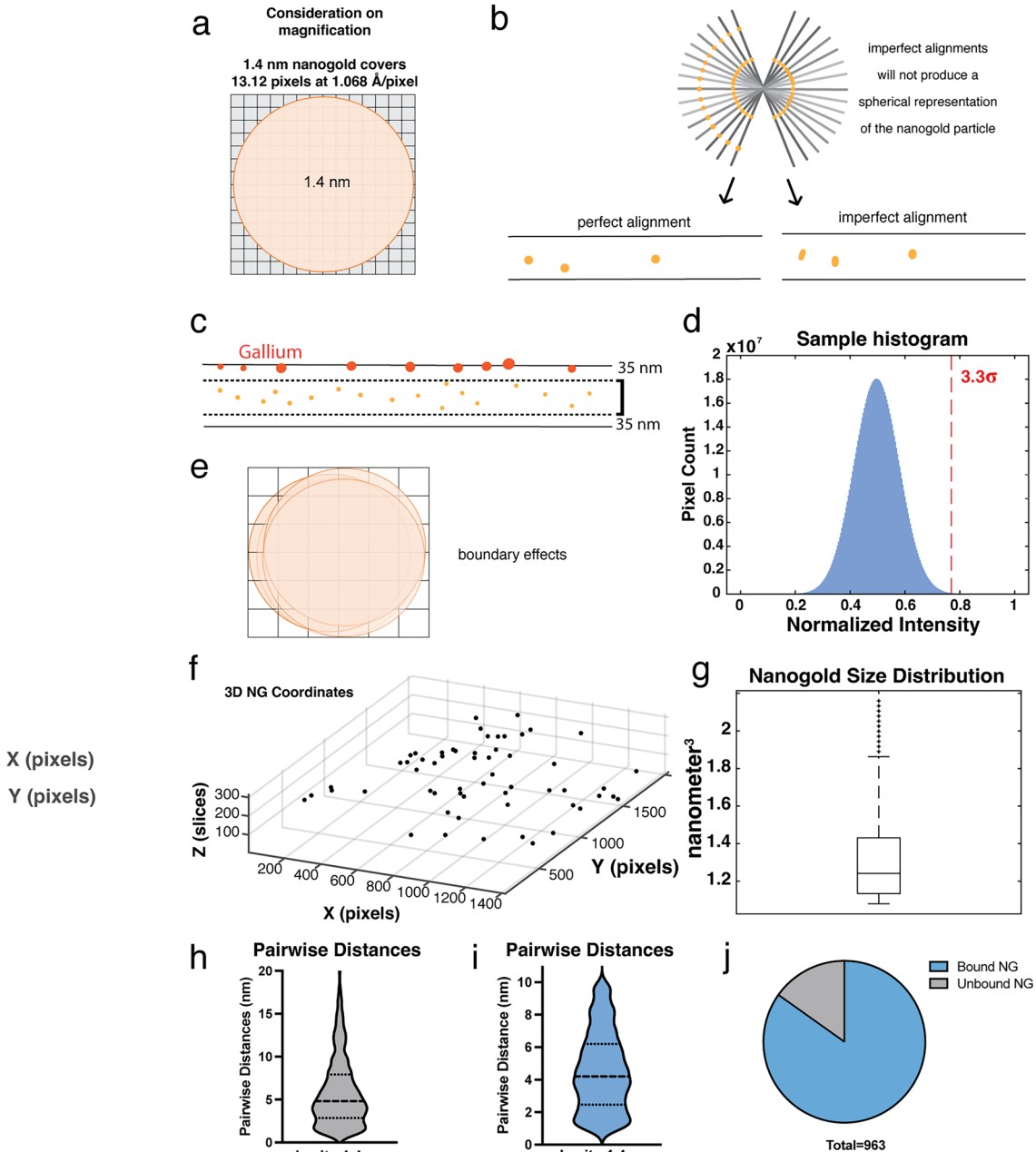

**Extended Data Fig. 6 | Whole Tomogram level Isolation of 1.4-nm-HAN-488.**
**(a)** Schematic of considerations for data collection regarding pixel size and
nanogold size. **(b)** Considerations of how the accuracy in tilt alignment affects
the nanogold particle within the reconstructed tomogram. **(c)** Tomograms were
first reconstructed by Weighted-Back Projection, the Gallium layers tomogram
were measured and then the tomograms were reconstructed again, but this time
without those slices, in order to mitigate any high intensity signals coming from
surface deposition of the gallium to affect the intensity histograms. The
pixel intensity distribution from those resulting tomograms were plotted.
**(d)** A sample normalized histogram distribution, high intensity nanogold pixels
were taken to be values 3.3 σ away from the mean. **(e)** Considerations on how a
sphere sampled over a grid results in boundary effects. **(f)** Sample 3D coordinates

from isolated nanogold. **(g)** Boxplot of size distribution of isolated 1.4 nm-
HAN-488 nanoparticles extracted from the whole tomogram level,
mean = 1.3 nm$^3$, std = 0.3, from 963 nanogold particles from 14 tomograms.
Box plot elements are defined as follows: the central red line indicates the
median; the bounds of the box mark the 25th and 75th percentiles (interquartile
range, IQR); whiskers extend to data points within 1.5 × IQR from the box; and
individual points beyond the whiskers are plotted as red crosses. **(h)** Violin plot
of the pairwise distance distribution from all extracted nanogold coordinates
and ribosomal coordinates, mean = 4.9 nm, STD = 1.9, n = 963 nanogold particles.
**(i)** Violin plot of a subset of the pairwise distances within 10 nm, mean = 4.5 nm,
STD = 2.4, n = 817 coordinates. **(j)** Pie chart of distribution of ribosomal-bound
(85%) and ribosomal-unbound nanogold particles (15%).

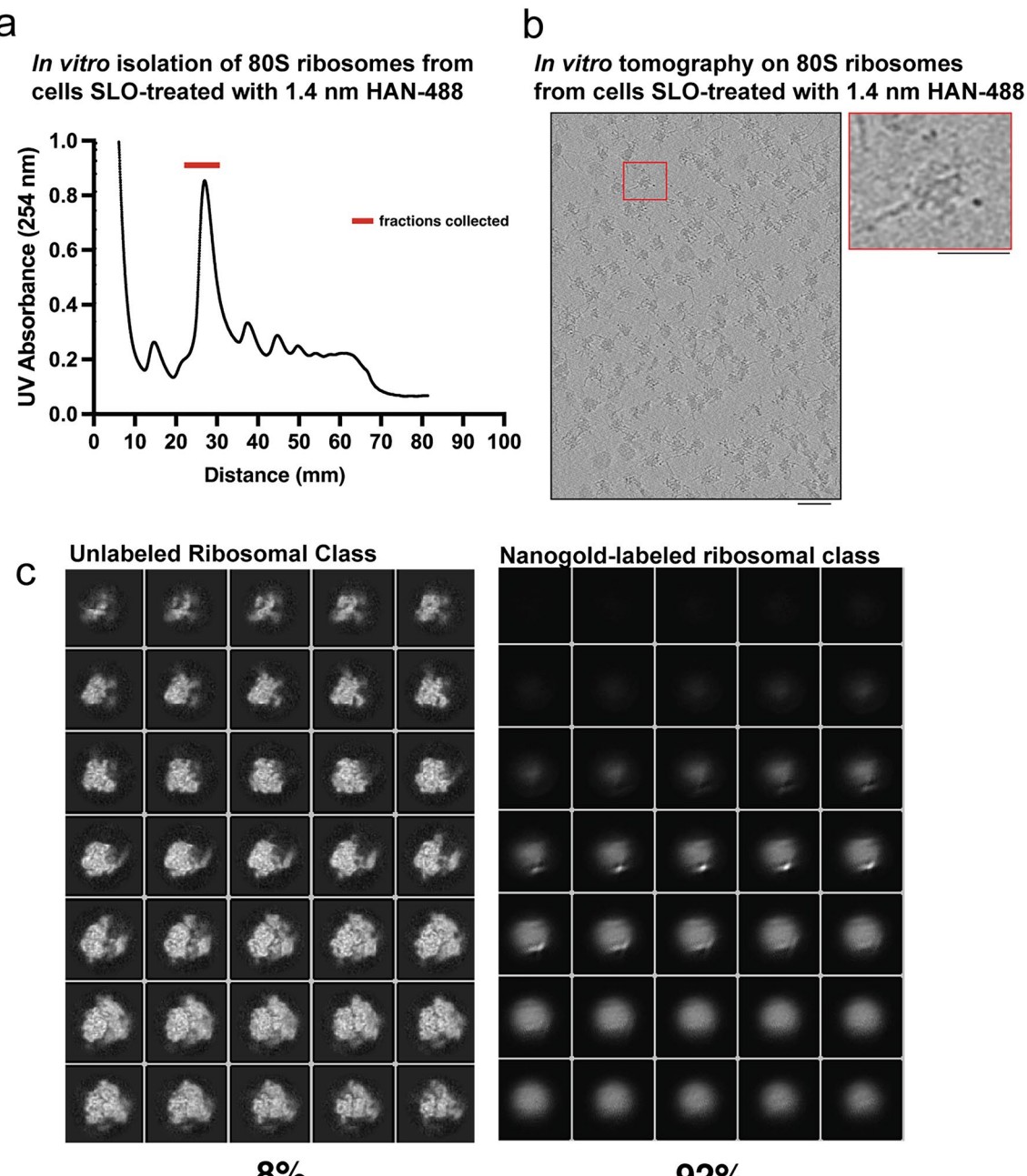

**a**

*In vitro* isolation of 80S ribosomes from cells SLO-treated with 1.4 nm HAN-488

**b**

*In vitro* tomography on 80S ribosomes from cells SLO-treated with 1.4 nm HAN-488

**c**

**Unlabeled Ribosomal Class**

**Nanogold-labeled ribosomal class**

8%

92%

**Extended Data Fig. 7 | Isolation of 1.4 nm-HAN-488 Labeled ribosomes from HEK 293 T L29-Halo cells. (a)** UV Absorbance profile following sucrose gradient of ribosomal particles isolated from approximately 3.2 ×107 HEK 293 T L29-Halo cells treated with SLO and delivery of 4 uM 1.4 nm-HAN-488 probes. Experiment was performed once. **(b)** Central slice of a representative reconstructed tomogram from 1.4 nm HAN-488 labeled ribosomes, right inset to show an individual ribosome, from 167 tomograms. **(c)** Orthoslices from unlabeled ribosomal class, and the nanogold-labeled ribosomal class with classification distribution, from 26,413 picked particles. Scale bars: b, 50 nm (left), 25 nm (right).

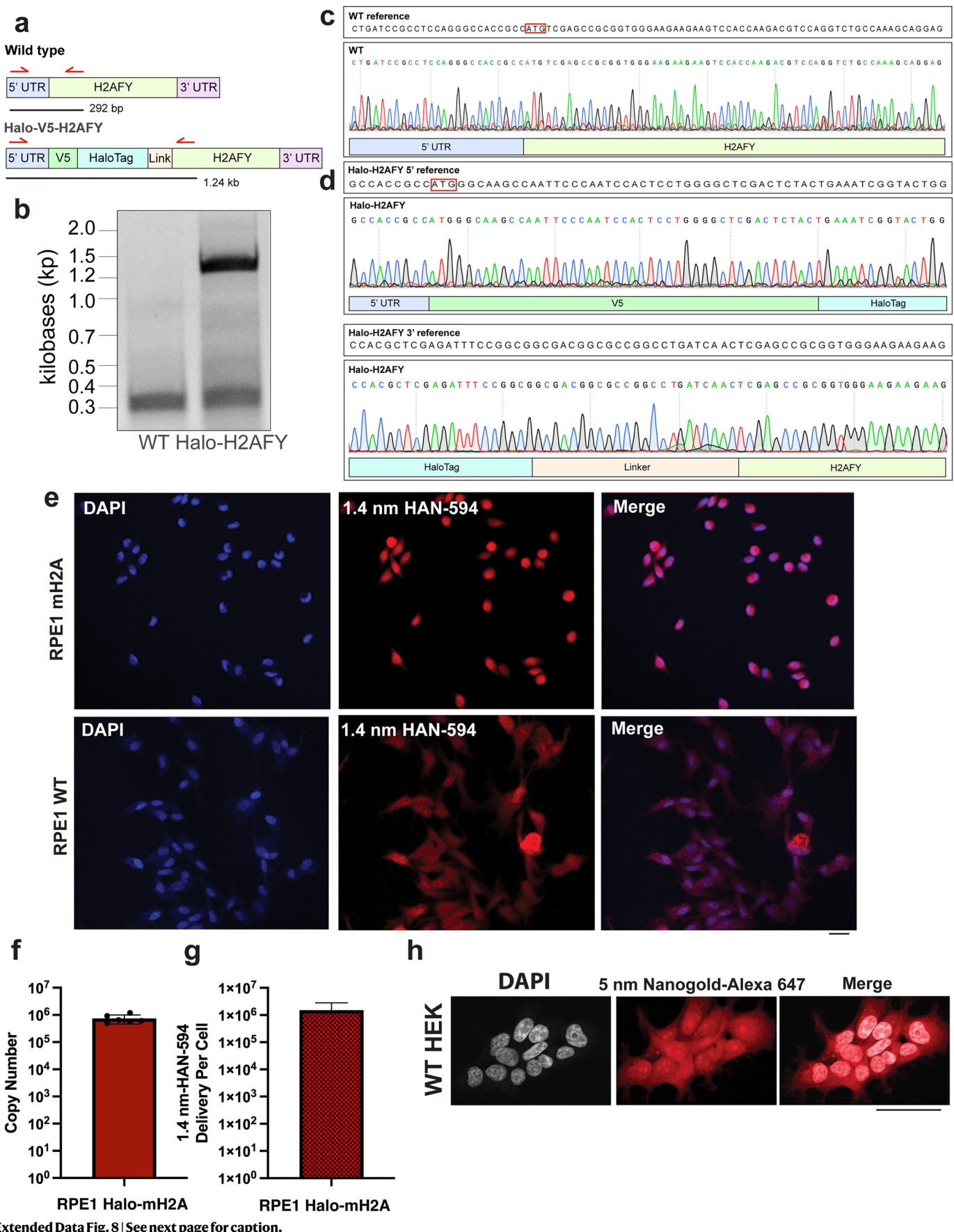

**Extended Data Fig. 8 | See next page for caption.**

**Extended Data Fig. 8 | Quantification of mH2A copy number in CRISPR knockin HaloTag-H2AFY RPE1 cells and quantification of 1.4 nm HAN-594 delivery. (a)** Diagram of the knock-in and PCR strategy. A V5 epitope and Halo tag are knocked into the N terminus of the human H2AFY (coding the histone variant MacroH2A) locus in retinal pigment epithelial (RPE1) cells. Top shows wild type (WT), bottom shows Halo-knock-in. Red arrows highlight PCR primers used for genotyping with the expected fragment size depicted as a black bar below. **(b)** DNA gel of the PCR described in A. N = 1. **(c)** Sanger sequencing of WT and V5-Halo-H2AFY RPE1 cells. **(d)** Alignment to the indicated genomic sequences.

**(e)** Localization of 1.4 nm-HAN-488 is restricted to the nucleus in RPE1 H2AFY-Halo cells but is dispersed in RPE1 WT cells. Experiment was repeated three times independently with reproducible results. Scale bar: 20 microns. **(f)** Absolute copy number of macroH2A molecules per RPE1 Halo-mH2A cell, mean=$7 \times 10^5$, SD = $2 \times 10^5$, N = 4 independent experiments. **(g)** Quantification of number of molecules of 1.4 nm-HAN-594 delivered to RPE1 Halo-mH2A cells, mean = $6.6 \times 10^5$, SD = $1.05 \times 10^5$, N = 4 independent experiments. **(h)** 5 nm Alexa 647-nanogold can enter the cell nucleus. Scale bar: 50 microns. N = 12 cells.

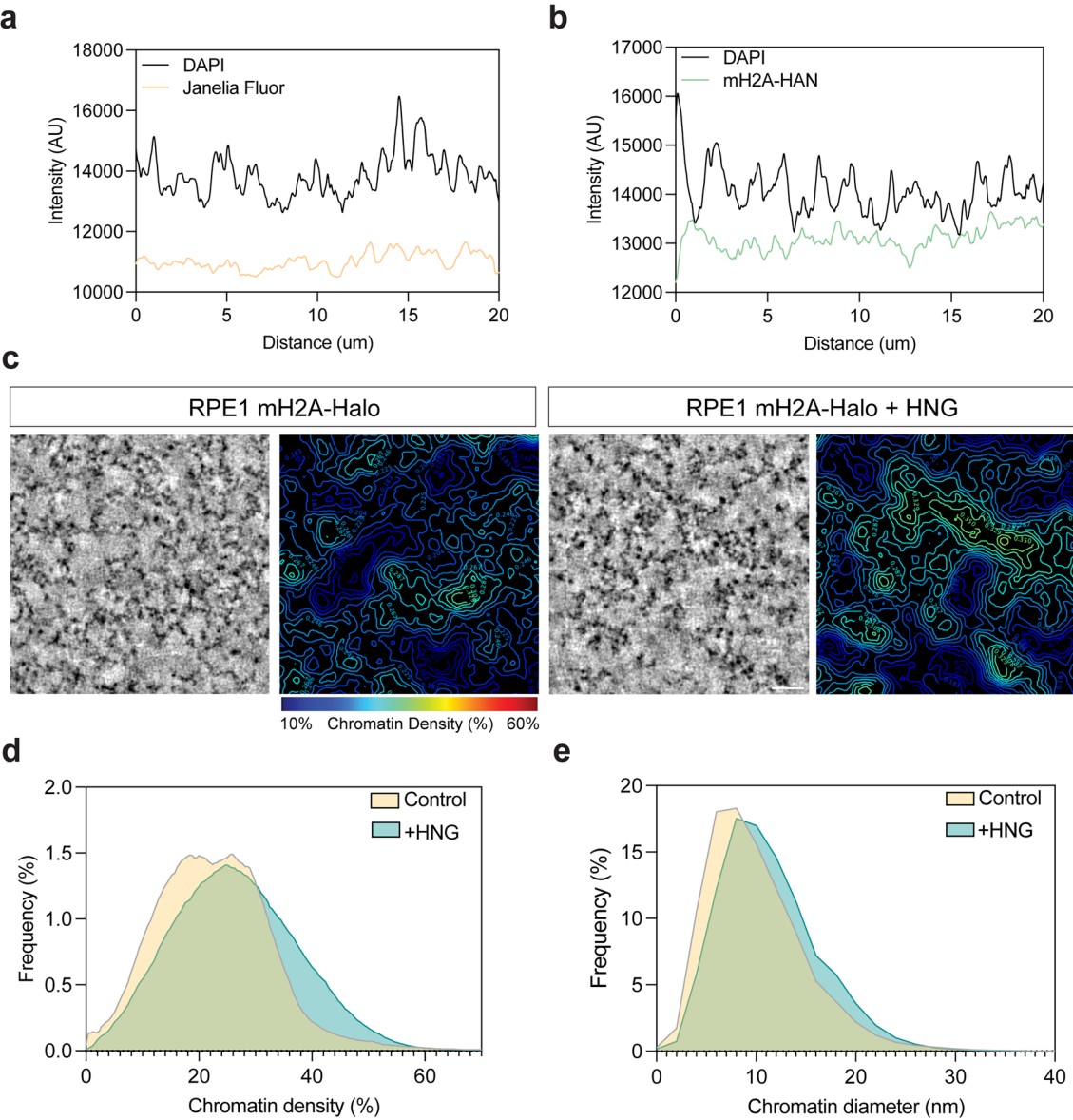

**Extended Data Fig. 9 | Tagging mH2A by ExoSloNano does not affect chromatin structure. (a)** line profile of DAPI and mH2A-Halo intensity using a Janelia Fluor ligand (JF). N = 5 nuclei. **(b)** line profile of DAPI and mH2A-Halo intensity using 1.4 nm Halo-nanogold (HAN). N = 5 nuclei. **(c)** Tomographic slices of control or 1.4 nm HAN-594 treated RPE1 mH2A cells strained using ChromEMT (left). Right, shows a chromatin density map where chromatin packing is color coded based on the scale bar. The scale bar is 100 nm. **(d)** Histogram of chromatin density of the cells shown in A. N = 4 tomograms. Control: median =21.2%, Treatment with 1.4 nm HAN-594 ( + HNG): median = 26.7%. p = 0.343 (two-sided Mann-Whitney U Test, ns.). **(e)** Histogram of chromatin diameter of the cells shown in A. Control: Median = 9 nm, Treatment with 1.4 nm HAN-594 ( + HNG): Median = 10 nm. p = 0.657 (two-sided Mann-Whitney U test, ns.), N = 4 tomograms.

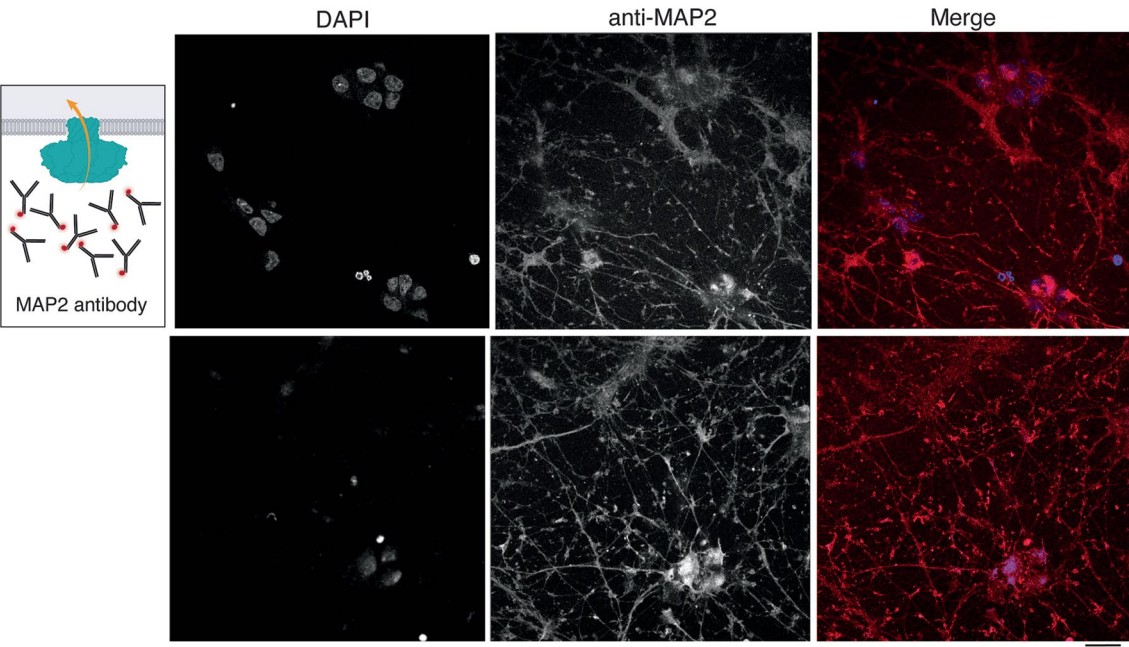

**Extended Data Fig. 10 | SLO-mediated delivery of Alexa-647 anti-MAP2 antibody into cultured iNeurons.** Experiment was repeated twice independently with reproducible results. Scale bar: 20 microns.

NMETH-A58398

# Reporting Summary

## Statistics

For all statistical analyses, confirm that the following items are present in the figure legend, table legend, main text, or Methods section.

| n/a | Confirmed | |
|---|---|---|
| ☐ | ☒ | The exact sample size (*n*) for each experimental group/condition, given as a discrete number and unit of measurement |
| ☐ | ☒ | A statement on whether measurements were taken from distinct samples or whether the same sample was measured repeatedly |
| ☐ | ☒ | The statistical test(s) used AND whether they are one- or two-sided<br>*Only common tests should be described solely by name; describe more complex techniques in the Methods section.* |
| ☒ | ☐ | A description of all covariates tested |
| ☐ | ☒ | A description of any assumptions or corrections, such as tests of normality and adjustment for multiple comparisons |
| ☐ | ☒ | A full description of the statistical parameters including central tendency (e.g. means) or other basic estimates (e.g. regression coefficient) AND variation (e.g. standard deviation) or associated estimates of uncertainty (e.g. confidence intervals) |
| ☐ | ☒ | For null hypothesis testing, the test statistic (e.g. *F*, *t*, *r*) with confidence intervals, effect sizes, degrees of freedom and *P* value noted<br>*Give P values as exact values whenever suitable.* |
| ☒ | ☐ | For Bayesian analysis, information on the choice of priors and Markov chain Monte Carlo settings |
| ☒ | ☐ | For hierarchical and complex designs, identification of the appropriate level for tests and full reporting of outcomes |
| ☒ | ☐ | Estimates of effect sizes (e.g. Cohen's *d*, Pearson's *r*), indicating how they were calculated |

*Our web collection on statistics for biologists contains articles on many of the points above.*

## Software and code

Policy information about availability of computer code

| Data collection | SerialEM 4.1.0-beta9, PACEtomo-v1.4.3, MAPS v3.23, AutoTEM v2.4, BD Biosciences LSR-II, BD FACSDiva 8.0.2, Nikon Elements 5.02.02 software, Molecular Devices SpectraMax i3x, Aveole Leonardo |
|---|---|
| Data analysis | Warp1.09, Warp1.10, RELION-3.1.4, RELION-4.0, IMOD 4.11, Isonet/0.2.1, AreTomo2, ChimeraX (UCSF) w/ ArtiaX v1.7, Prism v 10.3.1, Dynamo v11509, cryolo v1.8.4, DeepFinder, MATLAB version 2019b, ImageJ/FIJI v2.1.0, Imaris v10.2.2, Githubs: (1) ExoSloNano https://github.com/villa-lab/exoslonano , (2) Rosen lab/CATM github https://git.biohpc.swmed.edu/rosen-lab/catm, (3) Script to average neighboring slices of a tomogram: https://github.com/dgvjay/EM_Scripts/average_neighboringSlices.py |

For manuscripts utilizing custom algorithms or software that are central to the research but not yet described in published literature, software must be made available to editors and reviewers. We strongly encourage code deposition in a community repository (e.g. GitHub). See the Nature Portfolio guidelines for submitting code & software for further information.

## Data

Policy information about availability of data

All manuscripts must include a data availability statement. This statement should provide the following information, where applicable:

- Accession codes, unique identifiers, or web links for publicly available datasets
- A description of any restrictions on data availability
- For clinical datasets or third party data, please ensure that the statement adheres to our policy

Subtomogram averages have been deposited in EMDB (71205, 71211, 71113, 71202). Code and scripts available from the Villa lab and Rosen lab github repositories. Cell lines are available upon request to the corresponding author(s).

## Human research participants

Policy information about studies involving human research participants and Sex and Gender in Research.

| Reporting on sex and gender | Not applicable |
|---|---|
| Population characteristics | Not applicable |
| Recruitment | Not applicable |
| Ethics oversight | Not applicable |

Note that full information on the approval of the study protocol must also be provided in the manuscript.

# Field-specific reporting

Please select the one below that is the best fit for your research. If you are not sure, read the appropriate sections before making your selection.

☒ Life sciences ☐ Behavioural & social sciences ☐ Ecological, evolutionary & environmental sciences

For a reference copy of the document with all sections, see nature.com/documents/nr-reporting-summary-flat.pdf

# Life sciences study design

All studies must disclose on these points even when the disclosure is negative.

| Sample size | For protein abundance experiments (Flow Cytometry experiments), the sample size for each experiment following the established protocol of capturing 10,000 events for each experiment.<br>For room tempature and cryo-ET, sample size is limited to field of of view and available collection area. |
|---|---|
| Data exclusions | Poorly aligned tilt series were excluded from analysis, as well as micrographs with a poor CTF-fit. |
| Replication | Quantification of 1.4nm-HAN-594 enhancement in HEK cells, HEK RPL29-Halo cells with and without treatment was analyzed from nine cells for each indicated condition. Quantification of cellular growth following SLO treatment was performed in triplicate. Quantification of cellular volume from HEK RPL29 cells was determined from nine cells. Quantification of ribosomal abundance from HEK RPL29-Halo cells was performed from three independent experiments with two technical replicates. The quantification of mH2A from RPE1 Halo-mH2A cells was performed from four independent experiments. The internalization of 1.4 nm HAN-594 into RPE1 Halo-mH2A was performed from three independent experiments with two technical replicates. Line profile analysis of DAPI and mH2A-Halo intensity was analyzed from five nuclei. Chromatin intensity from ChromEMT experiments were analyzed from three tomograms. CryoFIB/ET was performed independently three times with reproducible results. |
| Randomization | Half-sets during subtomogram averaging were assigned randomly automatically by RELION-v3.1/RELION-v4 during 3D-auto refinements. Other experiments were not subject to randomization. |
| Blinding | Investigators were not blinded during data collection. |

# Reporting for specific materials, systems and methods

We require information from authors about some types of materials, experimental systems and methods used in many studies. Here, indicate whether each material, system or method listed is relevant to your study. If you are not sure if a list item applies to your research, read the appropriate section before selecting a response.

## Materials & experimental systems

| n/a | Involved in the study |
|-----|----------------------|
| ☐ | ☒ Antibodies |
| ☐ | ☒ Eukaryotic cell lines |
| ☒ | ☐ Palaeontology and archaeology |
| ☒ | ☐ Animals and other organisms |
| ☒ | ☐ Clinical data |
| ☒ | ☐ Dual use research of concern |

## Methods

| n/a | Involved in the study |
|-----|----------------------|
| ☒ | ☐ ChIP-seq |
| ☐ | ☒ Flow cytometry |
| ☒ | ☐ MRI-based neuroimaging |

# Antibodies

| | |
|---|---|
| Antibodies used | Anti-HaloTag® Monoclonal Antibody (Promega #G921A) diluted 1:5000, Anit-MAP2 antibody (Abcam catalog #EPR19691) diluted to 0.5 nM. |
| Validation | Promega provided validation of this antibody. This antibody was used for western blot analysis to verify expression of the HaloTag in HEK 293T RPL29-Halo cells. |

# Eukaryotic cell lines

Policy information about cell lines and Sex and Gender in Research

| | |
|---|---|
| Cell line source(s) | CRISPR knock-in HEK 293T RPL29-Halo was a gift from An et al. (2020), WT HEK 293T was obtained from ATCC (CRL-3216), WT RPE1 was obtained from ATCC (CRL-4000) from which RPE1 Halo-mH2A was generated in this study. CRISPR knock-in U2OS CTCF-Halo was a gift from Cattoglio et al. (2019). |
| Authentication | CRISPR knock-in HEK 293T L29-Halo were validated through Sanger sequencing of the endogenous RPL29 gene. The gene product was verified by a Western blot against HaloTag, and fluorescence microscopy. CRISPR knock-in RPE1 was validated through Sanger sequencing. |
| Mycoplasma contamination | Cell lines were not tested for mycoplasma. |
| Commonly misidentified lines (See ICLAC register) | WT HEK 293T was obtained from ATCC (CRL-3216), WT RPE1 was obtained from ATCC (CRL-4000) and from which RPE1-Halo-mHA was generated. |

# Flow Cytometry

## Plots

Confirm that:

☒ The axis labels state the marker and fluorochrome used (e.g. CD4-FITC).

☒ The axis scales are clearly visible. Include numbers along axes only for bottom left plot of group (a 'group' is an analysis of identical markers).

☒ All plots are contour plots with outliers or pseudocolor plots.

☒ A numerical value for number of cells or percentage (with statistics) is provided.

## Methodology

| | |
|---|---|
| Sample preparation | The absolute protein abundance of H2AFY was determined through flow cytometry when compared to a known standard, U2OS C32 Halo-CTCF cells which were cultured in DMEM supplemented with 1 g/L Glucose and 110 mg/L sodium pyruvate (Gibco, 10567-014) [44]. U2OS C32 HaloTag-CTCF cells were a gift from Robert Tijan's lab (UC Berkeley). U2OS C32 Halo-CTCF cells, HEK293T RPL29-Halo cells, and RPE1 H2AFY-Halo cells were cultured in 6 well plates to 90% confluency, to which 1 uM HaloTag TMR ligand (Promega, G825A) was supplied for 30 minutes. Cells were washed twice with 1x DPBS, trypsinized with 0.5% Trypsin (Gibco 15400054), gently resuspended in HBSS (Gibco 14025134), spun down and gently resuspended in HBSS with 50 ug/ml deoxyribonuclease I from bovine pancreas (Sigma D4263). For each sample, >10,000 events/rates were analyzed on a BD Biosciences LSR-II flow cytometer using a 561 nm excitation laser line and a 582 / 15 band pass filter cube. Absolute protein abundance was determined using an established protocol, Cattoglio et al. (2019). |
| Instrument | BD Biosciences LSR-II |
| Software | BD FACSDiva 8.0.2 |
| Cell population abundance | >10,000 TMR positive events |

Gating strategy | Gating was performed in order to isolate single cells based on fluorescence for TMR.

☒ Tick this box to confirm that a figure exemplifying the gating strategy is provided in the Supplementary Information.

