## [Peer Review File · Nature Methods]

ExoSloNano: Multi-Modal Nanogold Labels for identification of Macromolecules in Live Cells and Cryo-Electron Tomograms

Corresponding Author: Professor Elizabeth Villa

Version 0:

Decision Letter:

28th Nov 2024

Dear Elizabeth,

Your Article, "ExoSloNano: Multi-Modal Nanogold Tags for identification of Macromolecules in Live Cells and Cryo-Electron Tomograms", has now been seen by three reviewers. As you will see from their comments below, although the reviewers find your work of considerable potential interest, they have raised a number of concerns. We are interested in the possibility of publishing your paper in Nature Methods, but would like to consider your response to these concerns before we reach a final decision on publication. We therefore invite you to revise your manuscript to address these concerns.

We think the reviewer concerns were constructive and addressing them will strengthen the paper. We ask that you focus on clarifying how the probe can be used (a tag vs a label as discussed by ref 2), addressing the concerns about specificity, and addressing the concerns of how the gold electron density impacts reconstructions.

Link Redacted

We hope to receive your revised paper within three months. If you cannot send it within this time, please let us know. In this event, we will still be happy to reconsider your paper at a later date so long as nothing similar has been accepted for publication at Nature Methods or published elsewhere.

OPEN SCIENCE REQUIREMENTS

REPORTING SUMMARY AND EDITORIAL POLICY CHECKLISTS

OK TO DELETE SECTION IF NO GELS OR BLOTS

IMAGE INTEGRITY

DATA AVAILABILITY

All novel DNA and RNA sequencing data, protein sequences, genetic polymorphisms, linked genotype and phenotype data, gene expression data, macromolecular structures, and proteomics data must be deposited in a publicly accessible database, and accession codes and associated hyperlinks must be provided in the "Data Availability" section.

MATERIALS AVAILABILITY

SUPPLEMENTARY PROTOCOL

To help facilitate reproducibility and uptake of your method, we ask you to prepare a step-by-step Supplementary Protocol for the method described in this paper. We [encourage authors to share their step-by-step experimental protocols](https://www.nature.com/nature-research/editorial-policies/reporting-standards#protocols) on a protocol sharing platform of their choice and report the protocol DOI in the reference list. Nature Portfolio's protocols.io is a free-to-use and open resource for protocols; protocols deposited onto protocols.io are citable and can be linked from the published article. More details can be found at [protocols.io](https://www.protocols.io/help/publish-articles).

ORCID

Sincerely,
Rita

Rita Strack, Ph.D.
Senior Editor
Nature Methods

Reviewers' Comments:

Reviewer #1 (Remarks to the Author):

In this study, Young et al. describe the application of nanogold probes to detect proteins with a Halo tag. These gold particles enter cells via pores formed by bacterial toxins. Once inside, the Halo ligand on the nanogold particles mediates specific binding to the Halo-tagged protein of interest. The probes are also conjugated with Alexa Fluors, enabling their use in CLEM applications. This probe design holds broad potential for cellular tomography, allowing the detection of protein complexes that lack a defined structural signature, which is a significant need in the field.

However, while the manuscript includes some analysis of labeling efficiency, it lacks experiments or quantitative analysis on labeling specificity, which is arguably the most critical consideration in probe development.

Specific comments:

1) Fig 1e compares the fluorescence intensity and distribution of internalized nanogold particles with those of a cell-permeant Janelia fluorophore with a HaloLigand, to evaluate labeling efficiency and potential structural disruption. Due to the high abundance of ribosomes, these images do not confirm whether nanogolds are attached to ribosomes or accurately reflect ribosome distribution. This limitation is also seen in the EM images in Figure 1f, which lack sufficient resolution for definitive nanogold-ribosome colocalization analysis. While these images verify successful nanogold internalization, labeling assessments require higher resolution cryoEM or cryoET studies.

2) Figure 2c shows that subtomogram averaging of ribosomes results in a bright dot corresponding to the size of the 5nm nanogold particle. While we could say this is expected due to the significantly higher density of gold compared to proteins, it is also surprising given the large size difference between ribosomes and nanogold particles. This outcome might suggest that most ribosomes in the dataset are successfully labeled, but it also highlights a potential limitation: the probe may hinder the alignment of the targeted protein complex for subtomogram averaging. Have the authors attempted filtering, masking, or gold density removal to obtain a real ribosome subtomogram average? It's possible that the nanogold density remains with such ribosome subtomogram average, depending on the linker's flexibility.

3) A major concern is the lack of analysis on labeling specificity. For instance, in the tomograms shown in Fig. 2d, what percentage of nanoparticles lack a ribosome within the 5–6 nm range (the expected distance if a nanogold particle is specifically attached via the Halo tag)?

4) In applying 1.4-nm-HAN-594 for chromatin labeling, it would be helpful to examine whether there is a nanogold size limit for nuclear entry and effective labeling of nuclear targets. What would happen if 5nm nanogold probes were used instead? Are there differences in the efficiency of cellular or nuclear entry between the two sizes? While the authors have shown that 1.4 and 5nm nanogold particles can be distinguished in vitro, images showing both sizes within cells would be valuable to assess their potential for multiplex labeling.

Reviewer #2 (Remarks to the Author):

The authors present a new method, termed “ExoSloNano” to transport nanogold particles into live cells and tether them to proteins of interest using the HaloTag system. To carry the probes across the plasma membrane, cells are temporarily permeabilized with Streptolysin O, which allows intracellular delivery with little observable changes to cellular morphology – at least by room temperature electron microscopy. After pore repair, cells are subjected to a standard cryo-FIB pipeline and targets visualized by cryo-electron tomography (cryo-ET).

This represents a significant advance towards applying “classical” gold labeling techniques to cryo-samples, offering a promising new approach for localizing targets in live cells, which are currently too small for the available detection methods. The examples provided – ribosomes and nucleosomes containing mH2AFY – highlight the potential of ExoSloNano to expand the toolkit for labeling in cellular cryo-ET studies. The authors’ progress towards achieving “immunogold labeling in living cells” is encouraging and addresses a long-standing challenge and need in the field. This method could undoubtedly inspire future applications and is something I would consider for my own research.

However, I have three major objections that need to be addressed:

1) Confusion of label and tag. The authors describe the nanogold probe as enabling the detection of specific proteins using cryo-ET. However, the method relies on template matching (TM) for protein detection, with the nanogold only used post hoc to confirm the presence of specific components via subtomogram averaging (STA). This distinction should be made clearer in the manuscript. While ExoSloNano has merits, in particular for probing sub stoichiometric complexes and larger assemblies by STA, its applicability is constrained by prior knowledge of the target and the ability to generate averages via STA. For example, detecting the proximity of the gold label to the protein of interest provides localization within the linker’s range but does not directly detect the protein. If one were extremely purist, one could argue that – given enough resolution – even a simple GFP tag can be visualized by cryo-ET and STA and hence report on the presence of certain proteins in a larger complex. This approach, in fact, has already been done by Trinkaus & Fernández-Busnadiego, NatCom 2021. ExoSloNano of course can still provide valuable information when GFP tagging may not be possible, or averaging might yield too low-resolution averages only. However, the labeling efficiency of less than 100% paired with the fact that the gold particles cannot be used to detect the protein of interest directly is a clear limitation and needs to be addressed more clearly in the text.

2) Incomplete description of procedures and methods. To fully evaluate the validity of the claims, more detailed descriptions of the experimental procedures are necessary. For example, the ribosome STA section lacks sufficient detail to understand how the conclusions were reached. Are the methods used analogous to those described for nucleosomes? Additionally, referencing unpublished methods (e.g., “under review” or “in preparation” on lines 764 and 766) is not acceptable, as reviewers cannot assess their validity. Preprint repositories such as BioRxiv provide a solution for making such methods accessible and citable. Please give the reviewers access to these methods and provide citable references.

3) General Applicability. In my opinion, two aspects limit the general applicability of the method:

i) Sensitivity of Cell Types. While the authors use HEK and HeLa cells, it is not conceivable that their approach is viable in more sensitive cells (e.g. neurons), for which pore formation could be harmful. The authors should just clearly state the boundary conditions and for which cells their method is/may be useful.

ii) Labeling Efficiency and Specificity. The challenges of distinguishing bound versus free-floating nanogold particles remain unresolved. While the authors show ~70% of ribosomes labeled within the 5.1 nm cutoff, it is unclear whether this represents the true labeling efficiency or if some gold particles are simply farther away. This raises concerns about the method’s ability to achieve complete coverage for low-abundance targets. Addressing these points, perhaps with additional quantification (e.g. on purpose exceeding the expected copy number 5 to 10-fold; or only adding 50%) or simulations, would strengthen confidence in the method’s applicability to complex proteomic studies.

For the nucleosomes, for example, masks are applied to restrict the area that is analyzed. I am sure there are a lot of gold particles in the cytoplasm that are not bound to mH2AFY. Also vice versa: how many mH2AFY nucleosomes did not get labeled? Just to illustrate how important these considerations are: Using a binomial distribution and 100 ribosomes, if we assume a 70% labeling efficiency, there’s only a ~10% chance we find EXACTLY 70 ribosomes labeled and ~45% of ≥ 70 labeled.

Again, this does not mean that the method cannot be useful. ExoSloNano can provide useful lower bounds to many particle numbers in situ. But the boundary conditions and limitations must be specified more clearly and openly.

Furthermore, there are several smaller aspects that could improve the quality of the manuscript:

1) Quality of Text and Figures. With no offence intended, the manuscript would benefit from additional editing for clarity and consistency. Having read other papers of the same group, I am sure they can improve text and figures to make for as exciting a read as their other publications.

Here are some examples in no particular order to help the 2nd generation of the MS (not to shame or complain ☺):

{ } = delete

• line 52: “have been resolved in situ, or IN their native environment”

- line 59: “ExoSloNano” not defined
- line 94: “As of yet, the matching of high resolution signatures (2D template matching) is still {yet} restricted to molecules ...”
- line 95: “Machine learning (ML) programs for particle identification include TomoTwin and DeepFinder [21,22].” -> add cryolo, ... (probably this sentence fits better with the next paragraph?)
- line 101: “Template matching, machine learning, (COMMA) and artificial intelligence...”
- line 107: “Highly homologous structures often have small differences {in structural features} that are difficult to discern ...”
- line 118 and following (just to highlight occasional repetitiveness): “colloidal gold” 2x in two lines, “nanogold” 7x in seven lines; line 136 “challenging targets”, line 140: “chromatin organization” ... the list continues. I am sure the authors can check by themselves. ☺
- line 132: “ExoSloNano” is a typo AND still not explained what it stands for or what it does. Either introduce key concepts HERE or leave just the description of the ideal method.
- line 163: “detected in both fluorescence and electron microscop[ies]y, and recognize...”
- line 173: “Following pore formation, the plasma membrane repairs {after pore formation} through”
- line 214, 216 (and more): e6 ribosomes. Is this the proper notation for NM?
- Figure 1: c) label SLO pore; e) RPL29 in color or label which is which in merge; scale bars have different thickness; f) (top) there is an additional scale bar in the red-boxed region; in the caption (line 231): 150 nm (bottom).{(double period)}
- Figure 2: r HAT in figure, r (without HAT) in caption.
- line 256: “69.5% {percent}”
- Figure 3: b) same as above for FLM labels (color or describe merge); c) I have an inkling what the blue boxes and red crosses are but they are not described in text or caption. Caption: same r/r HAT; d) I guess the yellow box with arrow is the same as shown in e) (not described)
- line 313: “... exogenous, e.g., antibodies or endogenous, e.g., small genetically ...” not sure what is going on with the “e.g.s” here.
- line 328: “bp” (clear but not defined; maybe just spell out)
- line 376: “...of AB polymers on...” (either spell out aminobenzidine or use DAB)
- Figure 4: “???” ?; c) lower left quadrant: FM + FIB? – and why different from lower right quadrant?; e) what us the docked structure?
- Figure 5: not described what layovers are shown; why are ALL the particles below the (presumably) central slice?
- line 414: “In vitro conditions, this results in ...”; I guess “In vitro conditions/Under in vitro conditions”?
- line 422: “consistent with the size of THE 1.4 nm nanogold probe”
- line 424: “Notably, either zero or one nanogold”; “only a single copy”
- line 678: I guess carbon on gold? Or UltraAu?
- line 679: “Ten ul” probably OK to write “10 ul”

- ExFig 1: quantification plots not defined
- ExFig 2: d) error bars not defined; y-axis label “percent” -> really just 1%? Missing significance test between treated and untreated; e) just “untreated” (as also later); boxes = zoomed-in regions
- ExFig 3: c,d,e) error bars, outliers not defined;
- ExFig 5: there are two captions!; “knockin” = knock-in
- ExFig 6: error bars and outliers not defined; no outliers for b)?
- ExFig 7: d) typo “Chromatin density”; “+Nanogold”; e) typo “Chr{r}omatin”; “+ nanogold”

2) Statistical quantification. The authors should back up their claims by performing (at least) the customary statistical analyses. Examples:

- Histograms (e.g. Figs. 2&3) have different bins for the same type of data. Why?
- No FSCs are provided for the subtomogram averages. Not that the resolution matters, but it is convention for STA. Furthermore, FSC plots and resolutions could help to understand the effect of the nanogold on the STA process and help build confidence in the method.
- Distribution of cell growth (\pm SLO Ext. Fig. 2c) and chromatin density (Ext. Fig. 7 d,e) look different. Are they statistically indistinguishable (statistically insignificantly different/the same)? It is fine if the distributions are different as long as the cells finally recover. Just don't ignore this ...
- See also the considerations regarding the labeling efficiency above.

3) Distance Measurements. The manuscript would benefit from a more detailed explanation of distance measurements. For example, how was the offset of 12.6 nm for L29 calculated? Wouldn't a proper 3D Euclidian distance formula, i.e. $d = \sqrt{(x1-x2)^2+(y1-y2)^2+(z1-z2)^2}$, be needed to calculate the real distance of L29 and the gold particle? Furthermore, is it plausible that there is a 5.22 nm distance with the 5 nm probes and also 5.1 nm with 1.4 nm probe? I guess the authors subtract the gold radii, too? This is not described in the methods or text! Also, how are gold positions refined/defined per particle?

4) Unsubstantiated Claims. It is mentioned that motion correction and tomogram alignment were improved by the beads (line 250) and that by using differently sized beads multiplexing (lines 34, 285, 447) will be possible. All of this is mentioned, however, never shown. Either back up these claims or move them to the “outlook” in the last paragraph. For example, I am doubtful that multiplexing will be this easy, especially – see my first point – since the nanogold is never detected directly. This can easily be fixed by moving and tuning down things a notch.

Reviewer #3 (Remarks to the Author):

This research article presents a new method called ExoSloNano, which uses nanogold particles to label specific proteins within live cells for visualization by electron microscopy. The authors combine three techniques to achieve this labeling: (1)

SLO for delivery, (2) nanogold-HaLo ligand/tag, and (3) cryo-FIB-SEM/cryo-ET for observation. SLO has been used to deliver probes (fluorescent reagents and protein labeled for in-cell NMR) into cells, but it is unclear whether nanogold-label is delivered. Therefore, the main originality of this article is the successful delivery of nanogold-Halo ligand by SLO and its validation by cryo-FIB-SEM/cryo-ET.

The method was first validated by labeling ribosomes, which are the "standard" target of in-cell cryo-ET.

In the second application example, the authors label the nucleosomal protein macroH2A. This application is significant in two ways: It demonstrates that their method can deliver probes to the cytoplasm and the nucleus, and it clearly labels the very crowded environment of the nucleus, demonstrating the advantage of gold labels.

Overall, the experiments were carefully designed, and the authors especially paid particular attention to estimating the copy number of protein to be labeled.

Therefore, the reviewer thinks that the method will significantly contribute to the cellular structural biology field and highly recommends this article for Nature Methods.

Major points:

(1) One of the major developments of this article is multi-modal probes. 1.4 nm-Halo-Alexa-Nanogold-488 and 1.4 nm-Halo-Alexa-Nanogold-594. The methods section describes them as Line 651: "1.4-nm Halo-Alexa-Nanogold-594, 1.4-nm Halo-Alexa-Nanogold-488, and 5-nm Halo-Nanogold conjugates were custom ordered from Nanoprobes, Inc."

However, the exact molecular structures of these probes are not written in the article.

For example, in Figure 1, what is the white circle between Alexa and nanogold? Is the Halo ligand directly attached to gold atoms?

Because these multi-modal probes are the key to this study, the author should clearly describe them so that other researchers can follow.

(2) In Figure 2, "Cellular cryo-ET of 5 nm-HN labeled ribosome"

Although the authors showed the distance between ribosome and nanogold, separating the two linker-dependent distances would be ideal. (a) the linker between RPL29 and HaLo tag, and (b) the linker between HaLo tag and nanogold. To roughly estimate the first one, the reviewer suggests an additional figure showing the subtomographic averaged structure of the ribosome, focusing on the position of RPL29. It should show the additional density corresponding to the HaLo tag.

By the way, is the density of the HaLo tag "hidden" because of the high density of the nanogold?

(3) Related to point (2), the authors should deposit the 3D map of subtomographic averaged ribosome maps (aligned using entire volume and volume without gold) to EMDB.

(4) Similarly, please deposit the subtomographic averaged structure of nucleosome shown in Figure 5 to EMDB.

Minor points:

(4) Line: "nanogold enhancement and detection was negligible (Fig 1h)."

It is clear that the number of particles in negative controls is significantly smaller than the labeled ones. However, the authors should show this by numbers. Please provide a supplementary table (probably as raw data).

(5) Line 245: "Unsurprisingly, this indicated that the high-intensity region in the subtomogram corresponding to the high-mass density of the nanogold particle drives the alignment."

The meaning of "drives the alignment" is unclear, although it becomes clear from the next paragraph.

So, the reviewer suggests adding a few words to make the meaning of alignment. e.g., "the alignment of the nanogold-labeled ribosome"

(6) Figure 4e: It is difficult to see the 18 Å reconstruction of nucleosome with the model inside. Please also show the opaque surface rendering of the 3D reconstruction of the nucleosome.

Very minor point:

(7) Many "Fig." are written as "Fig" (without ".").

Reviewer #3 (Remarks on code availability):

Checked the following codes:

https://github.com/dgvjay/EM_Scripts

Version 1:

Decision Letter:

Our ref: NMETH-A58398A

29th Jul 2025

Dear Elizabeth,

Thank you for submitting your revised manuscript "ExoSloNano: Multi-Modal Nanogold Tags for identification of Macromolecules in Live Cells and Cryo-Electron Tomograms" (NMEMH-A58398A). It has now been seen by the original referees and their comments are below. The reviewers find that the paper has improved in revision, and therefore we'll be happy in principle to publish it in Nature Methods, pending minor revisions to satisfy the referees' final requests and to comply with our editorial and formatting guidelines.

Regarding revisions to satisfy the remaining concerns. We ask that you use the term label consistently throughout, though we agree that tag and label are used interchangeably in bioimaging. We also ask that you clarify the main uses of the tool and to what extent it can presently be used for subtomogram averaging/in situ structural biology. Regarding multiplexed labeling, please be careful not to claim you've shown multiplexed labeling. You can certainly mention that the approach can be used for multiplexing in instances where the probes can clearly be distinguished in EM.

TRANSPARENT PEER REVIEW

ORCID

Thank you again for your interest in Nature Methods. Please do not hesitate to contact us if you have any questions. We will be in touch again soon.

Sincerely,
Rita

Rita Strack, Ph.D.
Senior Editor
Nature Methods

P.S. Just so you know, Thursday is my last day at Nature Methods. I am not going far, but am not at liberty to announce my change until Friday. Allison will be happy to take over the final steps of your paper and to answer any questions that come up in the interim. It's been a pleasure working with you.

Reviewer #1 (Remarks to the Author):

I appreciate the authors' thorough and thoughtful response to my original comments. The revisions and new data have significantly strengthened the manuscript.

One of my primary concerns with the initial submission was the lack of quantitative analysis on labeling specificity and efficiency. In the revised manuscript, the authors have addressed this by incorporating a detailed assessment of labeling efficiency using both in situ cryo-ET datasets and purified ribosome samples. These analyses showed the proportion of the labeled targets carry the nanogold probes, providing evidence of labeling specificity and probe-target conjugation. These additions effectively resolve my concerns and make the method more quantitative, reproducible, and broadly applicable.

Another concern was the potential interference of high-contrast gold particles with subtomogram alignment. The authors have now implemented a strategy to "mask" the gold signal, allowing proper alignment based on ribosomal density. The resulting subtomogram average in Extended Data Figure 6, though low in resolution, is reasonable and demonstrates the feasibility of this approach. The inclusion of this analysis enhances the applicability of the method for structural determination of probe-labeled targets.

In addition, the manuscript has been improved in clarity and overall flow. These editorial revisions have further strengthened the presentation of the work.

Reviewer #2 (Remarks to the Author):

In their revised manuscript, Young et al. describe their ExoSloNano nanogold probes to detect proteins tagged with a HaloTag marker. These probes penetrate cells through pores created by bacterial toxins and specifically bind to the targeted Halo-tagged proteins via their Halo ligand. Additionally, the probes are conjugated with small organic fluorophores, facilitating their use in correlative light-electron microscopy (CLEM), which is particularly useful for identifying the location of protein complexes without a defined structural signature.

I find the use of ExoSloNano for the labeling of sub-stoichiometric nucleosome components particularly interesting, as indeed it is an unsolved problem on the small end of the complex spectrum to faithfully distinguish very similar proteins in cryo-ET and subtomogram averaging.

Furthermore, I really appreciate the authors' efforts to incorporate both mine and the other reviewers' comments; in particular, the efforts to better quantify the probe's behavior and providing more experimental details!

However, two of my main criticisms remain, which are in fact closely connected.
(But maybe this is all a big misunderstanding ... ☺)

• Q1 Tag vs label & Workflow Description: This requires an important clarification and clearer writing of the methods so I can either agree or disagree with the authors.

The authors say that

"we performed additional analysis and were able to detect the 1.4 nm-HAN-488 nanogold probes directly from the whole tomogram level (see Extended Data Fig. 7). This suggests that this method can work as a tag" (from the response to reviewers) also

"We also demonstrated the ability to directly detect nanogold to find targets." (page 22, line 558)

Based on the current description and methods section I do not agree. Yes, the authors can directly detect the gold. However, they always find the targets by template matching / crYOLO. Or not?

For the ribosomes, the methods section only say "Particles were picked on Warp generated deconvolved tomograms using cryolo version" (page 48, line 1067).

What do the authors mean exactly? Do they pick the gold signal or the ribosomes? Please clarify this, because it is of great importance to not create a misunderstanding!

Are the gold positions used to extract the (probably very large) box around the AuNP, delete the gold signal (as now well described), and by allowing for enough shift, a ribosome average is recovered without locking on to another ribosome in the box?

This would be quite different from the approach where gold and ribosome positions are determined SEPARATELY and then compared to one another to determine if they are bound or not. At least this is done for the nucleosomes where DeepFinder is used for the detection in conjunction with CATM (thanks for making this available now).

If indeed for the ribosomes the AuNP position is picked by crYOLO and used to also detect the ribosomes, a) cool, and b) please explain the procedure more clearly. Also please describe how box size and shifts are set to not lock on to potentially other ribosomes in the box.

However, if crYOLO is used to pick ribosomes and then by deleting the Au signal either Au or ribosome positions are obtained, please refrain from claiming that ExoSloNano can be used as a tag. None of the experiments then support that.

Just to clarify: all symmetrical labels have this issue! They do not point to their target and hence can only provide a distance cutoff in which to find the protein of interest. Based on the gold position alone, it may be hard or impossible to infer the correct particle location and orientation especially for smaller targets! Hence, if the gold signal cannot be used to recover "easy" ribosome positions independently of STA, we are looking at a clear limitation of the method as a "tag" that cannot be ignored, and it should please not be suggested in the main text!

That said, I think the authors' approach has great potential in probing the composition of larger complexes.

• Q2 Multiplexing: As also pointed out by reviewer #1, detecting two types of gold beads separately, and actually showing them together are two very different things. Like detecting eGFP and mClover. Yes, they work well individually, but they do not work together. Since the AuNP location cannot be used to pick particle positions (at least not shown here), the multiplexing idea will be limited to complexes where subtomogram averages can be obtained. While delivery of differently sized probes has been shown, they have not been detected together in cells (neither FLM nor EM).

To clarify, I am not asking for more experiments, but more clearly stating what factually HAS BEEN DONE and what COULD BE DONE is imperative for publishing the work as is!

Hence:

Page 12 line 331: "... multiple sizes COULD be employed ..."

Page 21 line 529: "... We demonstrate the feasibility of our approach for future multiplexing ..." -> e.g. "This may enable future multiplexing given suitable orthogonal probes" [and pending the answer to my Q1 😊]

Minor aspects:

- iNeurons: I really appreciate the effort in testing the general applicability in more sensitive cell types. However, no EM is shown and "happy" Map2 labeling in iNeurons should look less fragmented as you can see in our own data attached. All I see from this is that the neurons are not fully dead, yet...
- Some experimental details are missing / require details:
 - Page 44 line 935f: SLO Treatment. Please define how much of the probes was used in the final experiments! Also for Cryo-ET. I could only find this in page 50, line 1134: 4 μ M. Does this apply to all experiments?
 - Page 47 line 1022: cryo-ET Data Collection; -3 to -5 μ m defocus at \sim 1 apix – does this not cause serious CTF aliasing? (just a curiosity)
- Small items:
 - Page 1 line 30: cryo-Electron Tomography (cryo-ET) [to be consistent with the other definitions]
 - Page 2 line 50: supramolecular -> supermolecular? supermolecular?
 - Page 2 line 56: MS/MS not defined (just spell out)
 - Page 2 line 67: miniSOG
 - Page 3 line 88: 5 orders -> five orders
 - Page 3 line 105: SNR ratio -> Signal to Noise Ratio (SNR) [not defined]
 - Page 4 line 123: ..., polydisperse,[comma?] and relies on ...
 - Page 4 line 136: imaging modalities ":", or [space] – [space]?
 - Page 5 line 150: ExoSloNano [already defined]!
 - Page 6 line 189: botox-based mechanism is VASTLY different from SLO!!! [just a comment]
 - Page 6 line 193: Extended Data Fig. 13 – is it maybe 14?
 - Pages 9 – 11 end of paragraph on line 260: Is this not where (logically) the paragraph "Nanogold Signal Randomization for Subtomogram Analysis" (line 300) must go? Otherwise it is not clear for the NN analysis where the averages are coming from.
 - Page 11 line 306: soft-low pass -> soft low-pass?
 - Page 21 line 503: euler angles -> Euler angles
 - Page 21 line 537: endogenously-tagged -> endogenously tagged?
 - Page 22 line 573: not sure if such self-citing is appropriate ...
 - Page 32 Extended Data Fig 7 legend line 737f: g,h, i missing units of the mean =
 - Page 35 Extended Data Fig 9 legend line 776: knockin -> knock-in?; same line 778 and legend 779
 - Page 38 Extended Data Fig 13 legend: N=11 cells -> missing some type of quantification that this is for? (Otherwise, I count 12 cells 😊)
 - Page 40 Extended Data Figure 15: FSC Curves. Again ... not that resolution matters, but there's something wrong with these FSCs. a) -> FSC never reaches zero; unbin! (no complaints from Relion postprocess?); b) -> Phase randomized above unmasked? Also never comes down to zero (no complaints from Relion postprocess?).
 - Page 46 line 1010: Marinsreid -> MartinsrEd
 - Page 48 line 1069: "To overcome this and ..." – Overcome what?
 - Page 49 line 1094: "...using IsoNet" (double space)

Reviewer #3 (Remarks to the Author):

No further comment.

Version 2:

Decision Letter:

22nd Oct 2025

Dear Elizabeth,

I am pleased to inform you that your Article, "ExoSloNano: Multi-Modal Nanogold Labels for identification of Macromolecules in Live Cells and Cryo-Electron Tomograms", has now been accepted for publication in Nature Methods. The received and accepted dates will be 23 October 2024 and 22 October 2025. This note is intended to let you know what to expect from us over the next month or so, and to let you know where to address any further questions.

Over the next few weeks, your paper will be copyedited to ensure that it conforms to Nature Methods style. Once your paper is typeset, you will receive an email with a link to choose the appropriate publishing options for your paper and our Author Services team will be in touch regarding any additional information that may be required. It is extremely important that you let us know now whether you will be difficult to contact over the next month. If this is the case, we ask that you send us the contact information (email, phone and fax) of someone who will be able to check the proofs and deal with any last-minute problems.

Authors may need to take specific actions to achieve compliance with funder and institutional open access mandates.

If your research is supported by a funder that requires immediate open access (e.g. according to [Plan S principles](https://www.springernature.com/gp/open-science/plan-s-compliance) or the [NIH public access policy](https://www.springernature.com/gp/open-science/us-federal-agency-compliance)) then you should select the gold OA route, and we will direct you to the compliant route where possible. Because authors warrant under our subscription licensing terms that they haven't committed to licensing any version of their article under a licence inconsistent with the terms of our agreement – including the applicable embargo period – publication under the subscription model isn't suitable for authors whose funders require no embargo.

If you are active on Twitter/X or Bluesky, please e-mail me your and your coauthors' handles so that we may tag you when the paper is published.

Best regards,
Allison

Allison Doerr, Ph.D.
Chief Editor
Nature Methods

** Visit the Springer Nature Editorial and Publishing website at http://editorial-jobs.springernature.com?utm_source=ejP_NMeth_email&utm_medium=ejP_NMeth_email&utm_campaign=ejp_Nmeth www.springernature.com/editorial-and-publishing-jobs for more information about our career opportunities. If you have any questions please click [here](mailto:editorial.publishing.jobs@springernature.com).

Reviewers' Comments:

Reviewer #1 (Remarks to the Author):

In this study, Young et al. describe the application of nanogold probes to detect proteins with a Halo tag. These gold particles enter cells via pores formed by bacterial toxins. Once inside, the Halo ligand on the nanogold particles mediates specific binding to the Halo-tagged protein of interest. The probes are also conjugated with Alexa Fluors, enabling their use in CLEM applications. This probe design holds broad potential for cellular tomography, allowing the detection of protein complexes that lack a defined structural signature, which is a significant need in the field.

However, while the manuscript includes some analysis of labeling efficiency, it lacks experiments or quantitative analysis on labeling specificity, which is arguably the most critical consideration in probe development.

We thank the reviewer for their constructive review which strengthened our manuscript. We hope that the new experiment and analysis included on the revised manuscript address their concerns.

Specific comments:

1) Fig 1e compares the fluorescence intensity and distribution of internalized nanogold particles with those of a cell-permeant Janelia fluorophore with a HaloLigand, to evaluate labeling efficiency and potential structural disruption. Due to the high abundance of ribosomes, these images do not confirm whether nanogolds are attached to ribosomes or accurately reflect ribosome distribution. This limitation is also seen in the EM images in Figure 1f, which lack sufficient resolution for definitive nanogold-ribosome colocalization analysis. While these images verify successful nanogold internalization, labeling assessments require higher resolution cryoEM or cryoET studies.

Indeed resin-embedded conventional TEM lacks sufficient resolution to assess probe binding efficiency. Our goal for this figure was to demonstrate that this method of delivering fluorescent nanogold probes into the cell is compatible with conventional TEM (which allows larger volumetric studies). However, as this method lacks single molecule resolution, we performed in-situ cryo-ET studies to visualize single molecules and address labeling efficiencies. Labeling efficiencies have been added for the cryo-ET ribosomal experiments with 5 nm NG (line 273, Fig. 2f) and with 1.4 nm HAN-488 (line 319, Fig. 3h).

2) Figure 2c shows that subtomogram averaging of ribosomes results in a bright dot corresponding to the size of the 5 nm nanogold particle. While we could say this is expected due to the significantly higher density of gold compared to proteins, it is also surprising given the large size difference between ribosomes and nanogold particles. This outcome might suggest that most ribosomes in the dataset are successfully

labeled, but it also highlights a potential limitation: the probe may hinder the alignment of the targeted protein complex for subtomogram averaging. Have the authors attempted filtering, masking, or gold density removal to obtain a real ribosome subtomogram average? It's possible that the nanogold density remains with such ribosome subtomogram average, depending on the linker's flexibility.

Yes, indeed the nanogold particle is more dense and provides more signal than the biological target (the ribosome). To overcome this and recover the ribosomal signal and to mitigate the high intensity pixels from preventing ribosomal alignment during STA, the value of the pixels corresponding to the nanogold particle were randomized on a per-particle basis using a custom MATLAB script, effectively hiding the gold particles from the image, a process that the reviewer refers to as density removal. Then, those nanogold-randomized subtomograms were aligned in RELION yielding a low resolution (but now ribosomal aligned) subtomogram average. Particles were then re-extracted and a low pass filtered mask was used subsequently during further subtomogram analysis to mask out the nanogold. This workflow has been added as Extended Data Fig. 6.

3) A major concern is the lack of analysis on labeling specificity. For instance, in the tomograms shown in Fig. 2d, what percentage of nanoparticles lack a ribosome within the 5–6 nm range (the expected distance if a nanogold particle is specifically attached via the Halo tag)?

In order to identify unbound nanogold particles, a custom MATLAB script was written which identifies the high intensity pixels from the whole tomogram level of the best aligned tomograms (tomograms in which the mean residual error for the tomographic reconstruction is <1 nm).

High intensity voxels 3 standard deviations away from the mean were isolated. To create a binary mask, isolated voxels were set equal to 1 and everything else was set equal to 0. Neighboring voxels of value 1 were connected and grouped and the new center was determined. Background signal was any unpaired voxels equal to 1 and subsequently were removed. In theory, the number of voxels for which a 5 nm sphere (radius 2.5 nm) sampled over a grid of 4 Å/pixels is 1023 voxels. Two cut-offs were determined. The first were any coordinates greater than 1023 voxels were taken to be gallium, ice, or membrane and were filtered out. To account for undersampling due to the boundary effect of sampling a spherical object over a grid (Extended Data Fig. 7), the lower bound was set to be 500 voxels. False negatives (gallium, ice crystals) were filtering out high intensity coordinates which were <35 nm from the lamellae surface (Extended Data Fig. 7). For the tomogram presented in Fig. 2d, there were high intensity pixels that were later identified to be phosphate clusters within the mitochondria. All high intensity coordinates consistent with the 5 nm nanogold were identified to be bound to a ribosome (as in all were within the determined range from earlier analysis), 67% of the ribosomes for the tomogram in Fig. 2d were identified to have a 5 nm nanogold bound.

Analysis on the percent of unlabeled nanogold particles was determined for the 1.4 nm HAN-488 in situ dataset and was determined to be 15% across 14 tomograms (Extended Data Fig. 7). It is conceivable that the higher background is due to the more challenging image analysis on the smaller nanogold particle.

4) In applying 1.4-nm-HAN-594 for chromatin labeling, it would be helpful to examine whether there is a nanogold size limit for nuclear entry and effective labeling of nuclear targets. What would happen if 5 nm nanogold probes were used instead? Are there differences in the efficiency of cellular or nuclear entry between the two sizes?

We thank the reviewer for this question. We performed SLO-mediated delivery of a 5 nm-Fluorescent Nanogold into live cells (lines 493-497) and we visualized nuclear delivery (see Extended Data Fig. 13).

We want to point out that our protocol for recovery SLO includes an overnight step. It is possible that any nanogold that cannot enter through NPC can locate to the nucleus after nuclear envelope breakdown.

While the authors have shown that 1.4 and 5nm nanogold particles can be distinguished in vitro, images showing both sizes within cells would be valuable to assess their potential for multiplex labeling.

We anticipate multiplexing will be realized in the future, but believe that is beyond the scope of this study. We will move any mention of this to the Outlook section. And we have revised the following sentence, "Notably, these data **suggest** ~~show~~ that nanogold particles of multiple sizes can be employed."

Reviewer #2 (Remarks to the Author):

The authors present a new method, termed “ExoSloNano” to transport nanogold particles into live cells and tether them to proteins of interest using the HaloTag system. To carry the probes across the plasma membrane, cells are temporarily permeabilized with Streptolysin O, which allows intracellular delivery with little observable changes to cellular morphology – at least by room temperature electron microscopy. After pore repair, cells are subjected to a standard cryo-FIB pipeline and targets visualized by cryo-electron tomography (cryo-ET).

This represents a significant advance towards applying “classical” gold labeling techniques to cryo-samples, offering a promising new approach for localizing targets in live cells, which are currently too small for the available detection methods. The examples provided – ribosomes and nucleosomes containing mH2AFY – highlight the potential of ExSloNano to expand the toolkit for labeling in cellular cryo-ET studies. The authors’ progress towards achieving “immunogold labeling in living cells” is

encouraging and addresses a long-standing challenge and need in the field. This method could undoubtedly inspire future applications and is something I would consider for my own research.

Thank you for the comments and for the general excitement in utilizing this method.

However, I have three major objections that need to be addressed:

1) Confusion of label and tag. The authors describe the nanogold probe as enabling the detection of specific proteins using cryo-ET. However, the method relies on template matching (TM) for protein detection, with the nanogold only used post hoc to confirm the presence of specific components via subtomogram averaging (STA). This distinction should be made clearer in the manuscript. While ExoSloNano has merits, in particular for probing sub stoichiometric complexes and larger assemblies by STA, its applicability is constrained by prior knowledge of the target and the ability to generate averages via STA.

Thank you for this comment.

We developed this method with the ultimate goal of identifying targets unambiguously, and thus we appreciate the push to demonstrate this to the extent that it is possible now. Accordingly, we performed additional analysis and were able to detect the 1.4 nm-HAN-488 nanogold probes directly from the whole tomogram level (see Extended Data Fig. 7). This suggests that this method can work as a tag, but we agree that it is best suited if there are some other prior constraints (a membrane surface, for instance) in order to restrict the particle search.

We have added this point to a new "Limitations" section of the Discussion (lines 532-545), where we hope we made more explicit how this proof of principle has been performed, and the suitability of this method for different goals.

For example, detecting the proximity of the gold label to the protein of interest provides localization within the linker's range but does not directly detect the protein. If one were extremely purist, one could argue that – given enough resolution – even a simple GFP tag can be visualized by cryo-ET and STA and hence report on the presence of certain proteins in a larger complex. This approach, in fact, has already been done by Trinkaus & Fernández-Busnadiego, NatCom 2021. ExoSloNano of course can still provide valuable information when GFP tagging may not be possible, or averaging might yield too low-resolution averages only. However, the labeling efficiency of less than 100% paired with the fact that the gold particles cannot be used to detect the protein of interest directly is a clear limitation and needs to be addressed more clearly in the text.

To address concerns with detecting proteins of interest directly, please refer to the previous answer, and to the "Limitations" section of the Discussion section (lines 547-552). It is possible (and will be much more feasible with future developments by us and others) to directly detect gold in the sample, something that will likely never be possible for a fluorescent protein or other small protein.

Indeed, Trinkaus & Fernández-Busnadiego, NatCom 2021 could detect an ordered array of GFP molecules, but it is likely facilitated because the GFP was bound to an alpha-synuclein filament, leading to an ordered array of "GFP-like" molecules as the authors refer to them. As far as we are aware, subtomogram averaging of a single 30 kda globular protein (the size of GFP) has not yet been demonstrated.

In general, we think it is currently unlikely that GFP on its own would be a sufficient tag for most targets, or it would currently be a more widely implemented method for detecting proteins by in situ cryo-ET. Potentially, in the future, fluorescence mediated localization via super resolution CLEM approaches might allow the direct detection of GFP in the future, given that the laser power required to resolve single molecules does not lead to sample devitrification.

As stated in the outlook, our method is an addition to a growing palette of tools to localize proteins in cells, with clear advantages and challenges that thanks to the reviewers we now delineate more clearly.

2) Incomplete description of procedures and methods. To fully evaluate the validity of the claims, more detailed descriptions of the experimental procedures are necessary. For example, the ribosome STA section lacks sufficient detail to understand how the conclusions were reached. Are the methods used analogous to those described for nucleosomes? Additionally, referencing unpublished methods (e.g., "under review" or "in preparation" on lines 764 and 766) is not acceptable, as reviewers cannot assess their validity. Preprint repositories such as BioRxiv provide a solution for making such methods accessible and citable. Please give the reviewers access to these methods and provide citable references.

The ribosomal STA section was expanded and additional figures have been added in order to explain how the nanogold signal was removed (Extended Data Fig. 6). The nucleosome STA method section has been expanded, and the method paper previously mentioned as under review is now in press. We thank the reviewer for pointing out this oversight and hope that these additions allow the claims to be assessed.

3) General Applicability. In my opinion, two aspects limit the general applicability of the method:

i) Sensitivity of Cell Types. While the authors use HEK and HeLa cells, it is not conceivable that their approach is viable in more sensitive cells (e.g. neurons), for

which pore formation could be harmful. The authors should just clearly state the boundary conditions and for which cells their method is/may be useful.

We used HEK cells, as they are widely used in in situ cryo-ET studies, as well we used a primary cell line, RPE1. SLO is a widely used method in cell biology studies, and we now highlight that in the manuscript (lines 177-179) and supply additional references in which other researchers have used SLO on immune cells, Cos-7, CHO, U2OS, HeLa cells. Other researchers have delivered cargo into the cytosol of neuronal cells (primary rat cortical neurons) and iNeuron cells using another bacterial toxin (botulinum neurotoxins) and these citations have been added (lines 188-192).

To test SLO-mediated delivery into more sensitive cells, we induced iPS cells into iNeurons and treated them with SLO in order to deliver a fluorescently-labeled MAP2 antibody. We visualized robust labeling of iNeuron dendrites demonstrating antibody delivery into the cells (see Extended Data Fig.14).

Nevertheless, an assessment of whether SLO treatment affects the process under study should be made, and we now acknowledge this in the Limitation section.

ii) Labeling Efficiency and Specificity. The challenges of distinguishing bound versus free-floating nanogold particles remain unresolved. While the authors show ~70% of ribosomes labeled within the 5.1 nm cutoff, it is unclear whether this represents the true labeling efficiency or if some gold particles are simply farther away. This raises concerns about the method's ability to achieve complete coverage for low-abundance targets. Addressing these points, perhaps with additional quantification (e.g. on purpose exceeding the expected copy number 5 to 10-fold; or only adding 50%) or simulations, would strengthen confidence in the method's applicability to complex proteomic studies.

The labeling efficiency in different experiments will vary to some degree, as this method requires exogenous probes into the cell which is a convolution of various factors (efficacy of SLO pore formation, probe delivery including access to the target and binding affinity of e.g., HaloTag-HaloLigand, ability to saturate all targets, and rate of new protein synthesis after probe delivery). Thus, a labelling efficiency will reflect a combination of these efficiencies.

In order to address the reviewer concern, we now calculate the labeling efficiency for the ribosomal experiments using the same formula as in Single Molecule Fluorescence experiments (Hellmeier et al. 2024).

$$\text{Labeling Efficiency} = \frac{N_{\text{nanogold labeled ribosomes}}}{N_{\text{nanogold labeled ribosomes}} + N_{\text{nanogold unlabeled ribosomes}}}$$

In the first experiment (Fig. 2), we used pairwise distances between the ribosome subunit L29 and the nanogold, which is on average 5.1 ± 1.9 nm (Fig. 2e, now a violin plot). Using this criterion, the labeling efficiency of ribosomes, of which 6 million copies exist inside cells, is 0.695.

In order to have a better assessment of labelling efficiency, we scaled up the experiment and treated (4) 10-cm dishes of HEK 293T L29-Halo cells with SLO and 1.4 nm-HAN-488, and then purified ribosomes by a sucrose gradient in order to determine the labeled and unlabeled classes. In this experiment, the labeling efficiency was 0.92. This makes us confident that the efficiency of labelling in this case is extremely high.

For saturation of lower abundant targets for which there is a distinctive cellular localization (ER, lysosome, mitochondria), we envision that fluorescence microscopy could be used to detect the fluorophore-nanogold in order to corroborate that the fluoro-nanogold has been delivered to the cell, and that is localized to its intended target, without having extensive off-target localization (i.e. high background fluorescence signal).

For extremely low abundant targets, there are more fundamental issues such as determining whether it's statistically likely that this target will be included in enough tilt series (Young and Villa 2023). If the low abundance makes it so that neither fluorescence localization can be used to guide (e.g., too dim of a signal in cryogenic microscopy, difficult to correlate to TEM data), and that the likelihood of capturing the target in a tilt series is extremely low, this and other methods trying to label proteins within cells for cryo-ET cannot be used to overcome those pre-existing challenges. For project design in cryo-ET, we refer the reader to our previous review, in the last sentence of the discussion (xx line 551-553).

For the nucleosomes, for example, masks are applied to restrict the area that is analyzed. I am sure there are a lot of gold particles in the cytoplasm that are not bound to mH2AFY. Also vice versa: how many mH2AFY nucleosomes did not get labeled? Just to illustrate how important these considerations are: Using a binomial distribution and 100 ribosomes, if we assume a 70% labeling efficiency, there's only a ~10% chance we find EXACTLY 70 ribosomes labeled and ~45% of ≥ 70 labeled. Again, this does not mean that the method cannot be useful. ExoSloNano can provide useful lower bounds to many particle numbers in situ. But the boundary conditions and limitations must be specified more clearly and openly.

Thank you for the comment. For the latter part of this comment, please refer above to the labeling efficiency with the ribosome purification experiment, as well as the confusion over the definition of labelling efficiently.

With regards to the area in which we did tomography, prior to cryo-ET experiments, we used fluorescence microscopy to show that the fluorescent nanogold (1.4 nm HAN-594) was strictly localized to the nucleus in Halo-mH2A cells and that there is minimal fluorescence in the

cytoplasm. As we saw minimal fluorescence in the cytoplasm, we inferred that there will be minimal nanogold in the cytoplasm because the nanogold became bound to its target.

We have added Extended Data Fig. 10 to show that the probe is strictly localized to the nucleus of RPE1 mH2A cells. This is in contrast when the 1.4 nm HAN-594 probe is delivered to RPE1 WT cells, the fluorescence signal is dispersed throughout the nucleus and the cytoplasm, indicating that without the tag, the gold can freely enter and exit the nucleus, whereas in the tagged cells, the 1.4 nm HAN-594 probe is restricted to the nucleus as the HaloTag reacts with the mH2A-Halo nucleosomes.

For the last point about quantification of unlabeled particles, the reviewer is correct that this has to be considered carefully, and we do not think that we intend to do otherwise. We stand by this method being useful in many instances. For example, as far as we are aware, there are no alternative methods for identifying macroH2A nucleosomes and that in part was the motivation to develop this method.

Furthermore, there are several smaller aspects that could improve the quality of the manuscript:

1) Quality of Text and Figures. With no offence intended, the manuscript would benefit from additional editing for clarity and consistency. Having read other papers of the same group, I am sure they can improve text and figures to make for as exciting a read as their other publications.

We thank the reviewer for not intending to offend, and for all the constructive comments (major and minor) that they provided which helped us improve the manuscript. In that spirit, I (Elizabeth) would like to recommend that in future reviews, comparisons to previous work from a lab are not made, since they can be interpreted as a direct criticism of the trainees, even though I do believe this was not the intention.

Here are some examples in no particular order to help the 2nd generation of the MS (not to shame or complain 😊):

{ } = delete

• line 52: “have been resolved in situ, or IN their native environment”

Revised, thank you.

• line 59: “ExoSloNano” not defined

Moved this definition to earlier, when the acronym was introduced. Thank you for pointing out this omission.

• line 94: “As of yet, the matching of high resolution signatures (2D template matching) is still {yet} restricted to molecules ...”

Revised, thank you.

- **line 95: “Machine learning (ML) programs for particle identification include TomoTwin and DeepFinder [21,22].” -> add cryolo, ... (probably this sentence fits better with the next paragraph?)**

Revised, thank you.

- **line 101: “Template matching, machine learning, (COMMA) and artificial intelligence...”**

A comma has been added, thank you.

- **line 107: “Highly homologous structures often have small differences {in structural features} that are difficult to discern ...”**

Revised, thank you.

- **line 118 and following (just to highlight occasional repetitiveness): “colloidal gold” 2x in two lines, “nanogold” 7x in seven lines; line 136 “challenging targets”, line 140: “chromatin organization” ... the list continues. I am sure the authors can check by themselves. 😊**

Thank you for the comment, we have revised the text to reduce repetitiveness.

- **line 132: “ExoSioNano” is a typo AND still not explained what it stands for or what it does. Either introduce key concepts HERE or leave just the description of the ideal method.**

Thank you for pointing out this typo. We amended to introduce the acronym ExoSioNano earlier in the text. Thank you for pointing out this omission.

- **line 163: “detected in both fluorescence and electron microscop{ies}y, and recognize...”**

Revised, thank you.

- **line 173: “Following pore formation, the plasma membrane repairs {after pore formation} through”**

Revised, thank you.

- **line 214, 216 (and more): e6 ribosomes. Is this the proper notation for NM?**

The notation has been revised, thank you.

- **Figure 1:**

- c) label SLO pore;**

Label has been added, thank you.

- e) RPL29 in color or label which is which in merge; scale bars have different thickness;**

Fig 1e) Labels have been added to the merged images. Fig 1e) The scale bars are now the same thickness.

f) (top) there is an additional scale bar in the red-boxed region;

Fig 1f) Thank you for noticing this, the extra scale bar has been removed.

f) in the caption (line 231): 150 nm (bottom).{.}(double period)

Done

• **Figure 2: r HAT in figure, r (without HAT) in caption.**

Caption has been updated, thank you.

• **line 256: “69.5% {percent}”**

Amended, thank you.

• **Figure 3:**

b) same as above for FLM labels (color or describe merge);

Thank you, the merged image has been described.

c) I have an inkling what the blue boxes and red crosses are but they are not described in text or caption.

A new figure without the annotations has been added. It was the acquisition target

Caption: same r/r HAT;

Done

d) I guess the yellow box with arrow is the same as shown in

Yes

e) (not described)

Figure legend has been added.

• **line 313: “... exogenous, e.g., antibodies or endogenous, e.g., small genetically ...” not sure what is going on with the “e.g.s” here.**

Done.

• **line 328: “bp” (clear but not defined; maybe just spell out)**

Done, changed to "base pairs."

• **line 376: “...of AB polymers on...” (either spell out aminobenzidine or use DAB)**

Revised, thank you.

• **Figure 4: “???” ?;**

The ???'s were to highlight that this was an open-ended question of what happens inside the cell which motivated the experiment, we have changed "???" to "unknown."

c) lower left quadrant: FM + FIB? – and why different from lower right quadrant?;

It is the same lamella, but the image in the lower left quadrant was taken before the lamellae was "polished" or underwent the final thinning step in order to be transmissible for cryo-TEM. Imaging ultrathin lamella by cryo-fluorescence can sometimes lead to devitrification and care was taken to avoid this.

e) what is the docked structure?

Thank you for pointing this out, PDB code 2CV5 was used, and this was added to the legend for Figure 4.

Figure 5: not described what layovers are shown; why are ALL the particles below the (presumably) central slice?

Figure 5c was rendered in ChimeraX/ArtiaX. The updated version of this figure shows all the nucleosomes that are above the slice shown, any nucleosomes below that slice are no longer visible.

• line 414: “In vitro conditions, this results in ... ”; I guess “In in vitro conditions/Under in vitro conditions”?

Changed to "Under in vitro conditions"

• line 422: “consistent with the size of THE 1.4 nm nanogold probe”

Thank you, added.

• line 424: “Notably, either zero or one nanogold”; “only a single copy”

Thank you for the comment, but the result is not only a single copy, it is 0 or 1 copy.

The motivating question was to identify nucleosomes that contain macroH2A and if they would be hybrid nucleosomes or homo-typic nucleosomes containing two copies. But the results were that particles either contained 0 or 1 nanogold (not all particles had 1 copy, some had 0 as they were the canonical nucleosome), but there were no particles that had 2 copies. So, some had zero, some had 1, but none had two. If we say "only a single copy," it might imply all the nucleosomes had a single copy of mH2A.

• line 678: I guess carbon on gold? Or UltraAu?

Carbon on gold, clarification was added.

• line 679: “Ten ul” probably OK to write “10 ul”

Revised.

• ExFig 1: quantification plots not defined

We added 'violin plot' in the figure legend for clarity.

- **ExFig 2:**

- d) error bars not defined; y-axis label “percent” -> really just 1%? Missing significance test between treated and untreated;**

Significance test was added.

- e) just “untreated” (as also later); boxes = zoomed-in regions**

Done; Added.

- **ExFig 3: c,d,e) error bars, outliers not defined;**

Done.

- **ExFig 5: there are two captions!; ”knockin” = knock-in**

Amended.

- **ExFig 6: error bars and outliers not defined; no outliers for b)?**

Revised.

- **ExFig 7: d) typo “Chromatin dEnsity”; “+Nanogold”; e) typo “Chr{r}omatin”; “+nanogold”**

Thank you for noticing this, the typos have been corrected.

2) Statistical quantification. The authors should back up their claims by performing (at least) the customary statistical analyses.

Examples:

- **Histograms (e.g. Figs. 2&3) have different bins for the same type of data. Why?**

The histograms were changed to violin plots for clarity and made consistent.

- **No FSCs are provided for the subtomogram averages. Not that the resolution matters, but it is a convention for STA. Furthermore, FSC plots and resolutions could help to understand the effect of the nanogold on the STA process and help build confidence in the method.**

FSCs have been added to Extended Data Fig. 15.

- **Distribution of cell growth (\pm SLO Ext. Fig. 2c) and chromatin density (Ext. Fig. 7 d,e) look different. Are they statistically indistinguishable (statistically insignificantly different/the same)? It is fine if the distributions are different as long as the cells finally recover. Just don't ignore this ...**

Thank you for your comment, we have now added statistics to the legends of both these figures. Extended Data Fig. 2c shows no significant difference between the growth rates of the four conditions. Extended Data Fig. 12 shows no significant difference between the two groups

in regard to both chromatin diameter and chromatin density. We also added an additional replicate to calculate these statistics.

• **See also the considerations regarding the labeling efficiency above.**

Labeling efficiencies have been added for the ribosomal cryo-ET datasets (line 273, Fig. 2f, and line 319, Fig. 3h).

3) Distance Measurements. The manuscript would benefit from a more detailed explanation of distance measurements. For example, how was the offset of 12.6 nm for L29 calculated? Wouldn't a proper 3D Euclidean distance formula, i.e. $d = \sqrt{(x_1-x_2)^2+(y_1-y_2)^2+(z_1-z_2)^2}$, be needed to calculate the real distance of L29 and the gold particle? Furthermore, is it plausible that there is a 5.22 nm distance with the 5 nm probes and also 5.1 nm with 1.4 nm probe? I guess the authors subtract the gold radii, too? This is not described in the methods or text! Also, how are gold positions refined/defined per particle?

We thank the reviewer for pointing out this oversight. We have now included this in the methods section (lines 992-1001). The nanogold coordinates come from the RELION subtomogram average in which the alignment converged on the nanogold particle (Fig. 2c subtomogram average panel). In order to ribosomal signal and to mitigate the high intensity pixels from preventing ribosomal alignment during STA, the value of the pixels corresponding to the nanogold particle were randomized on a per-particle basis using a custom MATLAB script, effectively hiding the gold particles from the image. Then, those nanogold-randomized subtomograms were aligned in RELION yielding a low resolution (but now ribosomal aligned) subtomogram average. Particles were then re-extracted and a low pass filtered mask was used subsequently during further subtomogram analysis to mask out the nanogold. This workflow has been added as Extended Data Fig. 6.

After randomizing the nanogold signal from each subtomogram, subtomogram averaging was performed. The ribosomal average was re-centered around the last ordered residue of RPL29 (12.9 nm away), resulting in a new set of coordinates which correspond to RPL29.

Then, the pairwise distances for each nanogold-ribosome pair was determined. In order to determine the distances between the RPL29 and the nanogold, a matrix of all pairwise distances was determined using the Euclidean distance formula between both datasets in MATLAB (i.e. the distances between all possible pairs between set 1 and set 2 were generated). Then the minimum value or the minimum pairwise distances between each ribosome and nanogold was determined and plotted. The gold radii was not subtracted as the coordinates of the 1.4 nm or 5 nm nanogold were centered during the subtomogram averaging step.

4) Unsubstantiated Claims. It is mentioned that motion correction and tomogram alignment were improved by the beads (line 250) and that by using differently sized beads multiplexing (lines 34, 285, 447) will be possible. All of this is mentioned, however, never shown. Either back up these claims or move them to the “outlook” in the last paragraph. For example, I am doubtful that multiplexing will be this easy, especially – see my first point – since the nanogold is never detected directly. This can easily be fixed by moving and tuning down things a notch.

We have removed the claim that the nanogold could act as fiducials and facilitate tilt series alignment.

Further, we performed additional analysis and were able to directly detect the 1.4 nm nanogold from the whole tomogram level (see Extended Data Fig. 7).

Reviewer #3 (Remarks to the Author):

This research article presents a new method called ExoSloNano, which uses nanogold particles to label specific proteins within live cells for visualization by electron microscopy. The authors combine three techniques to achieve this labeling: (1) SLO for delivery, (2) nanogold-Halo ligand/tag, and (3) cryo-FIB-SEM/cryo-ET for observation. SLO has been used to deliver probes (fluorescent reagents and protein labeled for in-cell NMR) into cells, but it is unclear whether nanogold-label is delivered. Therefore, the main originality of this article is the successful delivery of nanogold-Halo ligand by SLO and its validation by cryo-FIB-SEM/cryo-ET.

The method was first validated by labeling ribosomes, which are the "standard" target of in-cell cryo-ET.

In the second application example, the authors label the nucleosomal protein macroH2A. This application is significant in two ways: It demonstrates that their method can deliver probes to the cytoplasm and the nucleus, and it clearly labels the very crowded environment of the nucleus, demonstrating the advantage of gold labels. Overall, the experiments were carefully designed, and the authors especially paid particular attention to estimating the copy number of protein to be labeled.

Therefore, the reviewer thinks that the method will significantly contribute to the cellular structural biology field and highly recommends this article for Nature Methods.

Major points:

(1) One of the major developments of this article is multi-modal probes. 1.4 nm-Halo-Alexa-Nanogold-488 and 1.4 nm-Halo-Alexa-Nanogold-594. The methods section describes them as Line 651: "1.4-nm Halo-Alexa-Nanogold-594, 1.4-nm Halo-Alexa-Nanogold-488, and 5-nm Halo-Nanogold conjugates were custom ordered from Nanoprobes, Inc."

However, the exact molecular structures of these probes are not written in the article. For example, in Figure 1, what is the white circle between Alexa and nanogold? Is the Halo ligand directly attached to gold atoms?

Because these multi-modal probes are the key to this study, the author should clearly describe them so that other researchers can follow.

Thank you very much for this comment. The 1.4 nm HAN-488, 1.4 nm HAN-594, and the 5 nm nanogold probes were all purchased from Nanoprobes, Inc. as a custom synthesis, using commercially available precursors (HaloLigand from Promega, 10,000 MW Dextran 488 from Thermo, and the 1.4 nm Nanogold from Nanoprobes). A more detailed schematic was added to Extended Data Fig. 1. The catalog numbers for each reagent were added to the method section.

Our apologies that the white circle was not explicit or explained, it was intended to represent a 10,000 MW dextran moiety (Dextran, Alexa Fluor® 488; 10,000 MW, Fixable (Thermo Fisher,

Catalog # D22910) which was used to (1) allow conjugation of fluorophores onto the dextran and (2) mitigate fluorescence quenching of the fluorophore by the nanogold particle.

(2) In Figure 2, "Cellular cryo-ET of 5 nm-HN labeled ribosome"

Although the authors showed the distance between ribosome and nanogold, separating the two linker-dependent distances would be ideal.

(a) the linker between RPL29 and HaLo tag, and

(b) the linker between HaLo tag and nanogold. To roughly estimate the first one, the reviewer suggests an additional figure showing the subtomogram averaged structure of the ribosome, focusing on the position of RPL29. It should show the additional density corresponding to the HaLo tag.

(a) We used a Worm-Like Chain model to model the unordered/flexible residues between the last ordered residue of RPL29 and the HaloTag, the distance is estimated to be ~3.6 nm.

(b) The distance between the HaloTag and the Nanogold is

-HaloLigand is on the 10,000 MW Dextran (roughly 3.6 nm) in diameter.

-HaloLigand itself which is about 22 Å from end-to-end but also enters the HaloTag cavity.

By the way, is the density of the HaLo tag "hidden" because of the high density of the nanogold?

This is what we have reasoned. Please see Extended Data Fig. 6 which explains how we have removed the nanogold signal in order to perform subtomogram averaging of the ribosome.

The second issue is that the HaloTag is not fixed relative to the ribosome, there is a distribution of distances (Extended Data Fig. 5), and it is difficult to leverage the ribosomal average in order to recover the HaloTag as they are not fixed relative to each other, and the HaloTag is just 30 kDa.

(3) Related to point (2), the authors should deposit the 3D map of subtomographic averaged ribosome maps (aligned using entire volume and volume without gold) to EMDB.

Amended, EMDB codes are in the Data availability section.

(4) Similarly, please deposit the subtomographic averaged structure of nucleosome shown in Figure 5 to EMDB.

Amended, this EMDB code is in the Data availability section.

Minor points:

(4) Line: "nanogold enhancement and detection was negligible (Fig 1h)."

It is clear that the number of particles in negative controls is significantly smaller than the labeled ones. However, the authors should show this by numbers. Please provide a supplementary table (probably as raw data).

We have added Extended Table 1, which shows the raw quantification.

(5) Line 245: "Unsurprisingly, this indicated that the high-intensity region in the subtomogram corresponding to the high-mass density of the nanogold particle drives the alignment." The meaning of "drives the alignment" is unclear, although it becomes clear from the next paragraph. So, the reviewer suggests adding a few words to make the meaning of alignment. e.g., "the alignment of the nanogold-labeled ribosome"

This has been revised, thank you for the suggestion.

(6) Figure 4e: It is difficult to see the 18 Å reconstruction of nucleosome with the model inside. Please also show the opaque surface rendering of the 3D reconstruction of the nucleosome.

Thank you for the comment, the opaque surface rendering of the 18 Å reconstruction of the nucleosome has been added.

Very minor point:

(7) Many "Fig." are written as "Fig" (without ".").

We fixed this, thank you for pointing this out.

Reviewer #3 (Remarks on code availability):

Checked the following codes:

https://github.com/dgvjay/EM_Scripts

Thank you for checking!

Reviewer #1:

Remarks to the Author:

I appreciate the authors' thorough and thoughtful response to my original comments. The revisions and new data have significantly strengthened the manuscript.

One of my primary concerns with the initial submission was the lack of quantitative analysis on labeling specificity and efficiency. In the revised manuscript, the authors have addressed this by incorporating a detailed assessment of labeling efficiency using both in situ cryo-ET datasets and purified ribosome samples. These analyses showed the proportion of the labeled targets carry the nanogold probes, providing evidence of labeling specificity and probe-target conjugation. These additions effectively resolve my concerns and make the method more quantitative, reproducible, and broadly applicable.

Another concern was the potential interference of high-contrast gold particles with subtomogram alignment. The authors have now implemented a strategy to “mask” the gold signal, allowing proper alignment based on ribosomal density. The resulting subtomogram average in Extended Data Figure 6, though low in resolution, is reasonable and demonstrates the feasibility of this approach. The inclusion of this analysis enhances the applicability of the method for structural determination of probe-labeled targets.

In addition, the manuscript has been improved in clarity and overall flow. These editorial revisions have further strengthened the presentation of the work.

We thank the reviewer for the insightful suggestions that improved the manuscript.

Reviewer #2:

Remarks to the Author:

In their revised manuscript, Young et al. describe their ExoSloNano nanogold probes to detect proteins tagged with a HaloTag marker. These probes penetrate cells through pores created by bacterial toxins and specifically bind to the targeted Halo-tagged proteins via their Halo ligand. Additionally, the probes are conjugated with small organic fluorophores, facilitating their use in correlative light-electron microscopy (CLEM), which is particularly useful for identifying the location of protein complexes without a defined structural signature.

I find the use of ExoSloNano for the labeling of sub-stoichiometric nucleosome components particularly interesting, as indeed it is an unsolved problem on the small

end of the complex spectrum to faithfully distinguish very similar proteins in cryo-ET and subtomogram averaging.

Furthermore, I really appreciate the authors' efforts to incorporate both mine and the other reviewers' comments; in particular, the efforts to better quantify the probe's behavior and providing more experimental details!

However, two of my main criticisms remain, which are in fact closely connected. (But maybe this is all a big misunderstanding ... 😊)

- Q1 Tag vs label & Workflow Description: This requires an important clarification and clearer writing of the methods so I can either agree or disagree with the authors.

The authors say that

“we performed additional analysis and were able to detect the 1.4 nm-HAN-488 nanogold probes directly from the whole tomogram level (see Extended Data Fig. 7). This suggests that this method can work as a tag” (from the response to reviewers)

also

“We also demonstrated the ability to directly detect nanogold to find targets.” (page 22, line 558)

Based on the current description and methods section I do not agree. Yes, the authors can directly detect the gold. However, they always find the targets by template matching / crYOLO. Or not?

For the ribosomes, the methods section only say “Particles were picked on Warp generated deconvolved tomograms using cryolo version” (page 48, line 1067).

What do the authors mean exactly? Do they pick the gold signal or the ribosomes? Please clarify this, because it is of great importance to not create a misunderstanding!

Are the gold positions used to extract the (probably very large) box around the AuNP, delete the gold signal (as now well described), and by allowing for enough shift, a ribosome average is recovered without locking on to another ribosome in the box?

This would be quite different from the approach where gold and ribosome positions are determined SEPARATELY and then compared to one another to determine if they are bound or not. At least this is done for the nucleosomes where DeepFinder is used for the detection in conjunction with CATM (thanks for making this available now).

If indeed for the ribosomes the AuNP position is picked by crYOLO and used to also detect the ribosomes, a) cool , and b) please explain the procedure more clearly. Also

please describe how box size and shifts are set to not lock on to potentially other ribosomes in the box.

However, if crYOLO is used to pick ribosomes and then by deleting the Au signal either Au or ribosome positions are obtained, please refrain from claiming that ExoSloNano can be used as a tag. None of the experiments then support that.

Just to clarify: all symmetrical labels have this issue! They do not point to their target and hence can only provide a distance cutoff in which to find the protein of interest. Based on the gold position alone, it may be hard or impossible to infer the correct particle location and orientation especially for smaller targets! Hence, if the gold signal cannot be used to recover “easy” ribosome positions independently of STA, we are looking at a clear limitation of the method as a “tag” that cannot be ignored, and it should please not be suggested in the main text!

That said, I think the authors’ approach has great potential in probing the composition of larger complexes.

We thank the reviewer for noting the usefulness of our method and for recognizing that we use it to address an unsolved biological problem in a methods paper. Before replying to the specific questions raised, we would like to point out that we could not find any reference in the literature differentiating tag and label in the way this reviewer does. Often, tags and labels are used indistinctly where we found them to be used distinctively (e.g., Liss, et al., 2016, DOI: 10.1038/srep17740; De Luis et al., 2025, DOI:10.1016/j.jbc.2025.110229), tags are considered a genetically encoded peptide added to the target during protein synthesis, e.g, GFP, Halo, whereas labels are chemical molecules that can be conjugated to the protein of interest or the tag after protein biosynthesis. In this definition, our method uses both a tag (Halo) and a label (Halo ligand + fluorophore + gold nanoparticle). Thus, the nomenclature that the reviewer attempts to establish does not make much sense to us. That said, we think that we understand their concerns and try to address them, and ***are now using the term label throughout our manuscript.***

First, we clarify that the goal of our labeling strategy is not first and foremost to enable in-situ structural biology, i.e., perform subtomogram analysis after finding the particles, although in some cases this method can be used toward this goal. In our work, and we know it to be true for others, it is often impossible to locate a protein within a network. There are a plethora of problems in cell biology where finding these proteins in the context of their environment, uniquely available through cryo-ET, would answer many impactful questions.

Further, as the reviewer notes, this method is readily useful to differentiate between complex A from complex B,C.. We showed one example for macroH2A, where we solved a longstanding question on stoichiometry, and paved the way to future studies investigating how their location with respect to nuclear lamina determines their function in lamina-associated domains. We hope that this method will be readily used for distinguishing between different complexes with similar structures, such as kinesin variants, different microtubule associated proteins, nucleosome variants, compositional changes conferring specialized functions to ribosomes, among others.

At this point it is worth noting, as we do in the introduction, that in-situ cryo-EM requires careful experimental design, and is a long way from the button-clicking automation that has been achieved for single particle analysis, and will remain so until other technical advances like phase plates and correctors deliver the potential of directly recognizing proteins down to ~40 kDa from the data (Russo et al., 2020, DOI: 10.1039/D2FD00076H). When that is the case, labeling methods will be more powerful and widely applicable.

To our knowledge, gold nanoparticles are the highest-resolution label available in biophysics that also show the molecular context (vs. fluorescence microscopy). **This and other labeling methods in cryo-ET bring the molecular specificity of the mature field of fluorescence microscopy to the vast contextual information of cryo-ET. We now discuss in the Limitations section the challenges of subtomogram analysis where the target is small or or low abundance.**

The analysis of ribosomes in the revised text is as in the first submission. Initially, we first picked ribosomes using cryOLO and recovered the gold in the 1.4 nm and 5 nm NG-labeled dataset. In the revised text, **we show the ability to directly localize the nanogold in the tomogram.** Finding the gold is key to show that this method can effectively find labeled proteins within their environment, and we thank the reviewers for this important suggestion. Our analysis determining the location of the target and the label separately for targets that can be identified combined with subtomogram analysis is meant to validate that the nanogold label is present in the location that is expected, to measure labeling efficiency, to show that the label can be extracted from the data for analysis, and to locate sub-stoichiometric nucleosome components, as the reviewer kindly appreciates in their comments.

With respect to the reviewer comment about the ability to extract the particles based on the gold locations, as the reviewer points out, this is a general problem with labels that we consider beyond the scope of this manuscript. With the current state of the art hardware and software, the primary utility of this method is already realized. Achieving

what the reviewer envisions for ribosomes would expand its application but would require significant computational processing and may or may not work with current data quality and processing software (subtomogram analysis currently requires well center particles, the majority of which are true positives). As this is a general issue, it is in itself an independent study worth pursuing. ***We have added a paragraph to the Limitations noting the challenge of subtomogram analysis for particles that are labelled (by this or any other labelling method) but cannot be directly picked with template matching or by considering other priors (lines 520-534).*** Another area in which gold labels will be immediately useful is in the new area of 2D template matching, where the gold would dramatically reduce the search space, making the method computationally tenable for more projects and laboratories (personal communication, Prof. Bronwyn Lucas, UC Berkeley). 2-D template matching requires a high-resolution template, but it's much more effective than 3-D template matching used in subtomogram analysis.

Another way to achieve the location of the particle in cases where subtomogram analysis is the goal could be to use two or more same or different labels per particle to hone in on the center location of the target of interest. We are currently doing this for Piezo channels on plasma membranes, with very encouraging preliminary data. We consider this beyond the scope of this manuscript, but want to illustrate that the method's versatility allows for its adaptation to different goals.

- Q2 Multiplexing: As also pointed out by reviewer #1, detecting two types of gold beads separately, and actually showing them together are two very different things. Like detecting eGFP and mClover. Yes, they work well individually, but they do not work together. Since the AuNP location cannot be used to pick particle positions (at least not shown here), the multiplexing idea will be limited to complexes where subtomogram averages can be obtained. While delivery of differently sized probes has been shown, they have not been detected together in cells (neither FLM nor EM).

To clarify, I am not asking for more experiments, but more clearly stating what factually HAS BEEN DONE and what COULD BE DONE is imperative for publishing the work as is!

The reviewer's example of eGFP and mClover refers to the wavelengths of excitation and emission of the fluorophores, for which there is no equivalent in nanoparticles of different sizes. There is no reason to think that these different nanoparticles conjugated to different tags (Halo, SNAP, etc) would not work. For fluorophores included in the labels, orthogonal dyes can be selected that work together. ***In the manuscript we show that the gold- nanoparticle sizes are easily distinguishable, and while this is in different data sets, we have no reason to think this would be different***

otherwise and see no point in carrying out an experiment (in agreement with the reviewer).

Hence:

Page 12 line 331: "... multiple sizes COULD be employed ..."

We have implemented this change.

Page 21 line 529: "... We demonstrate the feasibility of our approach for future multiplexing ..." -> e.g. "This may enable future multiplexing given suitable orthogonal probes" [and pending the answer to my Q1 😊]

We have revised this sentence to: "This can enable future multiplexing using orthogonal probes"

Minor aspects:

We recognize the reviewer's attention to detail. We have implemented relevant changes as suggested in his minor aspects section.

• iNeurons: I really appreciate the effort in testing the general applicability in more sensitive cell types. However, no EM is shown and "happy" Map2 labeling in iNeurons should look less fragmented as you can see in our own data attached. All I see from this is that the neurons are not fully dead, yet...

We leave it to future users to troubleshoot their specified cell type. The experiments we showed were done in collaboration with Michael Ward (NINDS, NIH) who is an expert in iNeurons.

• Some experimental details are missing / require details:

- Page 44 line 935f: SLO Treatment. Please define how much of the probes was used in the final experiments! Also for Cryo-ET. I could only find this in page 50, line 1134: 4 μM . Does this apply to all experiments?

- Page 47 line 1022: cryo-ET Data Collection; -3 to -5 μm defocus at ~ 1 apix – does this not cause serious CTF aliasing? (just a curiosity) **CTF estimations report resolution of individual images at $\sim 5-6 \text{ \AA}$, generally due to the reduced exposure of 3-5 e/ \AA per tilt image. If CTF estimations were $\sim 1-2 \text{ \AA}$ at -3 to -5 μm defocus, yes, there likely would be aliasing. Just a reminder that we are not interpreting EM maps at that resolution.**

• Small items:

- Page 1 line 30: cryo-Electron Tomography (cryo-ET) [to be consistent with the other definitions) **Revised**

- Page 2 line 50: supremolecular -> supramolecular? supermolecular? **Revised to supramolecular**

- Page 2 line 56: MS/MS not defined (just spell out) **Revised**
- Page 2 line 67: miniSOG **Revised**
- Page 3 line 88: 5 orders -> five orders **Revised**
- Page 3 line 105: SNR ratio -> Signal to Noise Ratio (SNR) [not defined] **Revised**
- Page 4 line 123: ..., polydisperse,[comma?] and relies on ... **Revised**
- Page 4 line 136: imaging modalities “:” or [space] – [space]? **Revised**
- Page 5 line 150: ExoSloNano [already defined]! **Revised**
- Page 6 line 189: botox-based mechanism is VASTLY different from SLO!!! [just a comment] **Okay**
- Page 6 line 193: Extended Data Fig. 13 – is it maybe 14? **Correct it was 14, this is now Extended Data Fig 10 after condensed for ED Fig limitations.**
- Pages 9 – 11 end of paragraph on line 260: Is this not where (logically) the paragraph “Nanogold Signal Randomization for Subtomogram Analysis” (line 300) must go? Otherwise it is not clear for the NN analysis where the averages are coming from.
- Page 11 line 306: soft-low pass -> soft low-pass? **Revised**
- Page 21 line 503: euler angles -> Euler angles **Revised**
- Page 21 line 537: endogenously-tagged -> endogenously tagged? **Revised**
- Page 22 line 573: not sure if such self-citing is appropriate ... **Back-of-the-envelope estimations of protein abundance is an extremely important consideration in tomography studies, and we find that it’s rarely addressed, which is the failure cause of many projects. We wrote that review to help new and existing tomography**
- Page 32 Extended Data Fig 7 legend line 737f: g,h, i missing units of the mean = **Revised**
- Page 35 Extended Data Fig 9 legend line 776: knockin -> knock-in?; same line 778 and legend 779 **Revised**
- Page 38 Extended Data Fig 13 legend: N=11 cells -> missing some type of quantification that this is for? (Otherwise, I count 12 cells 😊)
Revised
- Page 40 Extended Data Figure 15: FSC Curves. Again ... not that resolution matters, but there’s something wrong with these FSCs. a) -> FSC never reaches zero; unbin! (no complaints from Relion postprocess?); b) -> Phase randomized above unmasked? Also never comes down to zero (no complaints from Relion postprocess?).
Different masks for processing have been used and new FSC's have been generated.
- Page 46 line 1010: Marinsreid -> MartinsrIEd **Revised**
- Page 48 line 1069: “To overcome this and ...” – Overcome what? **The nanogold signal which dominates the alignment.**
- Page 49 line 1094: “...using IsoNet” (double space) **Revised**

Reviewer #3:

Remarks to the Author:

No further comment.

Thank you.